# Extensive folding variability between homologous chromosomes in mammalian cells

Ibai Irastorza-Azcarate [1,19 ✉], Alexander Kukalev [1,19], Rieke Kempfer [1,2,12,19], Christoph J Thieme [1], Guido Mastrobuoni [3], Julia Markowski [1,2,4,13], Gesa Loof [1,2,14], Thomas M Sparks [1], Emily Brookes [5,15], Kedar Nath Natarajan [1,5,16], Stephan Sauer [5,17], Amanda G Fisher [5,18], Mario Nicodemi [6], Bing Ren [7], Roland F Schwarz [4,8,9], Stefan Kempa [3] & Ana Pombo [1,2,5,10,11,19 ✉]

## Abstract

**Genetic variation and 3D chromatin structure have major roles in gene regulation. Due to challenges in mapping chromatin conformation with haplotype-specific resolution, the effects of genetic sequence variation on 3D genome structure and gene expression imbalance remain understudied. Here, we applied Genome Architecture Mapping (GAM) to a hybrid mouse embryonic stem cell (mESC) line with high density of single-nucleotide polymorphisms (SNPs). GAM resolved haplotype-specific 3D genome structures with high sensitivity, revealing extensive allelic differences in chromatin compartments, topologically associating domains (TADs), long-range enhancer–promoter contacts, and CTCF loops. Architectural differences often coincide with allele-specific differences in gene expression, and with Polycomb occupancy. We show that histone genes are expressed with allelic imbalance in mESCs, and are involved in haplotype-specific chromatin contacts marked by H3K27me3. Conditional knockouts of Polycomb enzymatic subunits, Ezh2 or Ring1, show that one-third of ASE genes, including histone genes, is regulated through Polycomb repression. Our work reveals highly distinct 3D folding structures between homologous chromosomes, and highlights their intricate connections with allelic gene expression.**

**Keywords** Genome Structure; Gene Regulation; Allele-specific Expression; Polycomb; Histone Locus
**Subject Category** Chromatin, Transcription & Genomics

## Introduction

Mammalian cells contain two parental chromosome copies, each with extensive heterozygous sequence variations. Genetic diversity of parental alleles confers advantages over repressive mutations, and is associated with longer lifespan (Xu et al, 2019) and reduced risk of aging-related diseases (Belloy et al, 2020). Many repressive heterozygous variants, with broad cell functions, are also found in healthy individuals (Schmenger et al, 2022), and loss of heterozygosity and single allele amplifications are features of many cancers (LaFramboise et al, 2005; Nichols et al, 2020). Skewed allelic gene expression has been reported to affect 6 to 80% of genes, depending on the species and tissue (Dixon et al, 2015; Crowley et al, 2015; Murata et al, 2012; Pinter et al, 2015; Chen et al, 2016; Savol et al, 2017; Cleary and Seoighe, 2021). DNA methylation at gene promoters or transcription factor binding sites has been implicated in allele-specific expression of imprinted genes in mouse and human (Noordermeer and Feil, 2020). More recently, genome-wide analyses of monoallelic expression in the murine zygote, morula and blastocyst, revealed a more prominent role of Polycomb repression than DNA methylation in allelic imbalance of gene expression (Santini et al, 2021; Inoue et al, 2017). The relative contributions of genetic and epigenetic mechanisms to random allelic expression imbalance are less well understood, but suggested to be highly gene-specific (Crowley et al, 2015; Marion-Poll et al, 2021).

Little is known about haplotype-specific differences in 3D genome structure and their contributions to allelic asymmetries in gene expression. The sparsity of genetic variation between haplotypes makes it technically challenging to map 3D genome structure with haplotype specificity by either sequencing or imaging technologies. In ligation-based methods, such as Hi-C, the unequivocal assignment of ligation events to

[1]Max-Delbrück-Center for Molecular Medicine in the Helmholtz Association (MDC), Berlin Institute for Medical Systems Biology (BIMSB), Epigenetic Regulation and Chromatin Architecture Group, 10115 Berlin, Germany. [2]Humboldt-Universität zu Berlin, Berlin, Germany. [3]Max-Delbrück Centre for Molecular Medicine, Berlin Institute for Medical Systems Biology, Proteomics and Metabolomic Platform, 10115 Berlin, Germany. [4]Max-Delbrück Centre for Molecular Medicine, Berlin Institute for Medical Systems Biology, Evolutionary and Cancer Genomics Group, 10115 Berlin, Germany. [5]MRC Laboratory of Medical Sciences, Imperial College London, London W12 0NN, UK. [6]Dipartimento di Fisica, Università di Napoli "Federico II", and INFN, Napoli, Italy. [7]Center for Epigenomics and Department of Cellular and Molecular Medicine, University of California, San Diego School of Medicine, La Jolla, CA, USA. [8]Institute for Computational Cancer Biology (ICCB), Center for Integrated Oncology (CIO), Cancer Research Center Cologne Essen (CCCE), Cologne, Germany. [9]BIFOLD—Berlin Institute for the Foundations of Learning and Data, Berlin, Germany. [10]Department of Biology, Johns Hopkins University, Baltimore, MD, USA. [11]Department of Molecular Biology and Genetics, Johns Hopkins University School of Medicine, Baltimore, MD, USA. [12]Present address: Sophia Genetics SA, A-One Park, Rolle 1180, Switzerland. [13]Present address: Department of Biomedical Informatics, Harvard Medical School, Boston, MA, USA. [14]Present address: Aix Marseille Univ, CNRS, IBDM (UMR 7288), Turing Centre for Living Systems, Marseille, France. [15]Present address: School of Biological Sciences, University of Southampton, Southampton, UK. [16]Present address: DTU Bioengineering, Technical University of Denmark, Kongens Lyngby, Denmark. [17]Present address: Regeneron Ireland DAC, Dublin 2 D02 HH27, Ireland. [18]Present address: Department of Biochemistry, University of Oxford, Oxford OX1 3QU, UK. [19]These authors contributed equally: Ibai Irastorza-Azcarate, Alexander Kukalev, Rieke Kempfer, Ana Pombo. ✉E-mail: ibai.irastorzaazcarate@mdc-berlin.de; ana.pombo@mdc-berlin.de

the correct haplotype (phasing) requires the presence of at least one SNP on either side of the ligation product and therefore has inherently low sensitivity (reviewed in Li et al, 2021). Nevertheless, phased Hi-C data has revealed intrinsic parental variability for the mammalian female X chromosomes, upon random inactivation, and in the timing of chromatin folding during meiosis, but few structural differences have been reported between the two parental copies of somatic chromosomes, except at a small number of imprinted genes (Ferguson-Smith, 2011; Rao et al, 2014b; Reinius and Sandberg, 2015; Giorgetti et al, 2016; Tan et al, 2018; Han et al, 2020; Tan et al, 2021; He et al, 2023).

GAM is a ligation-free technology which captures long-range chromatin interactions spanning whole chromosomes and has revealed extensive specificity in the 3D chromatin structure of specific cell types (Beagrie et al, 2017; Beagrie et al, 2023; Winick-Ng et al, 2021; Fiorillo et al, 2021). GAM measures 3D genome topology by sequencing the DNA content from a collection of thin (~200 nm) nuclear cryosections, and infers 3D chromatin contacts from the probability of co-segregation of genomic regions across the collection of nuclear slices. As whole genomic regions (typically 20–50 kb long) are called positive in GAM data from the accumulation of many sequencing reads, including many SNP-containing reads, we reasoned that the phasing of GAM data should be highly efficient. Local haplotype fidelity of GAM data has been previously shown (Markowski et al, 2021), supporting our efforts to generate haplotype-specific insights into chromatin folding from GAM data.

To investigate differences in the 3D genome folding of homologous chromosomes, we applied GAM to a hybrid mESC line with high SNP density. We developed novel computational pipelines to phase GAM data, and discovered extensive 3D structural differences between the two parental chromosomes across all length scales, including in A/B compartments, topologically associating domains (TADs), and at the contact level. We also collected total RNA-seq data and found that 15% of expressed genes have allele-specific expression (ASE) bias in mESCs, including some imprinted genes, but also many housekeeping, ribosomal, and histone genes. ASE genes were often located in regions with haplotype-specific structural differences, which coincided with H3K27me3 occupancy, haplotype-specific enhancer–promoter contacts, or CTCF loops. We also inferred chromatin compaction from GAM data, and found that the most active alleles are consistently more decondensed than the least active ones. We discovered that many histone genes are ASE genes in mESCs, and that histone genes are involved in allele-specific long-range chromatin contacts marked by H3K27me3 occupancy. Finally, we used conditional knockouts of Polycomb enzymatic subunits and showed that the expression of many, but not all ASE genes, including histone genes, is under Polycomb regulation.

# Results

## Overview of datasets collected

To investigate haplotype-specific differences in 3D genome structure using GAM, we collected data from the F123 mESC line (Gribnau et al, 2003). The F123 line was originally derived from F1 hybrid embryos from a CAST/S129 cross and its genotype has high SNP density (average 1 SNP/124 nucleotides across autosomes; Fig. 1A). GAM data was produced in multiplex mode which combines three independent nuclear profiles (3NP) in each GAM sample (Beagrie et al, 2023; Winick-Ng et al, 2021), and collected

from two biological replicates. After quality control, the replicate datasets were merged, resulting in the largest GAM dataset to date, obtained from approximately 3,700 single mESCs (Fig. EV1A).

To address the impact of haplotype-specific 3D genome structure on gene expression and chromatin regulation, we also mapped gene expression using total RNA-seq, chromatin occupancy using ChIP-seq of RNA polymerase II phosphorylated on Serine-5 (Pol2-S5p) or Serine-7 (Pol2-S7p) residues of its C-terminal domain, and the Polycomb mark H3K27me3 (Fig. 1B). We collected and remapped published datasets produced in F123 mESCs for ChIP-seq of CTCF, cohesin (RAD21), H3K4me3 and H3K27ac (Huang et al, 2021; Data ref: Hui and Ren 2020), chromatin accessibility (ATAC-seq; Juric et al, 2019b; Data ref: Juric et al, 2019a), and DNA methylation (whole-genome bisulfite sequencing; Li et al, 2019; Data ref: Li et al, 2019). Finally, we also considered published annotations of lamina-associated domains (LADs) obtained by Lamin B1 DamID from the mESCs clone E14Tg2A (Peric-Hupkes et al, 2010b; Data ref: Peric-Hupkes et al, 2010a). The datasets produced, publicly available, and the processed data resources are summarized in Dataset EV1.

## GAM-phaser: a pipeline to phase GAM data

GAM contact maps are produced by measuring the frequency of co-segregation of genomic windows of a given length across the collection of nuclear profiles (NPs; Beagrie et al, 2017). The reads sequenced in each GAM sample are used to identify, in a binary fashion, the presence or absence of genomic windows of a given resolution in that sample (Fig. 1C). As each GAM sample is obtained from thin nuclear slices and contains only 5–15% of the genome (Beagrie et al, 2023), the sequencing depth required to detect positive windows is promptly saturated in each GAM library (approx. 2–3 million reads; see Methods section "GAM library preparation and high-throughput sequencing"). With enough sequencing depth, the effective resolution of GAM contact matrices depends on the number of NPs collected to enable sampling all possible window co-segregation events up to a given genomic distance or across each chromosome (Beagrie et al, 2023).

We developed the GAM-Phaser pipeline to phase GAM data to the CAST and S129 haplotypes (Fig. EV1B). Briefly, the positive genomic windows in each GAM sample are first defined in unphased GAM datasets (Fig. 1C), using a sample-specific threshold of nucleotide coverage, as previously described (Winick-Ng et al, 2021). Next, SNP-containing reads are phased to CAST and S129 haplotypes. For a conservative detection of haplotype-specific windows in each GAM sample, we applied the read detection threshold defined for unphased windows to the SNP-containing reads. After phasing genomic windows, it becomes possible to distinguish whether a given GAM sample contains DNA from one or both chromosome copies (Sample 1 and 2, respectively; Fig. 1D).

The overall phasing efficiency achieved across all F123 GAM datasets was a total of 37% of all sequencing reads assigned, with a similar proportion to CAST or S129 haplotypes (Fig. EV1C). However, it was possible to phase 70–75% genomic windows at 10–100 kb resolutions, respectively, which were evenly detected between CAST and S129 windows (Fig. EV1D). The high efficiency of GAM data phasing results from the presence of multiple nucleotide polymorphisms, which collect many SNP-containing reads, in each positive genomic window. In comparison, phasing of

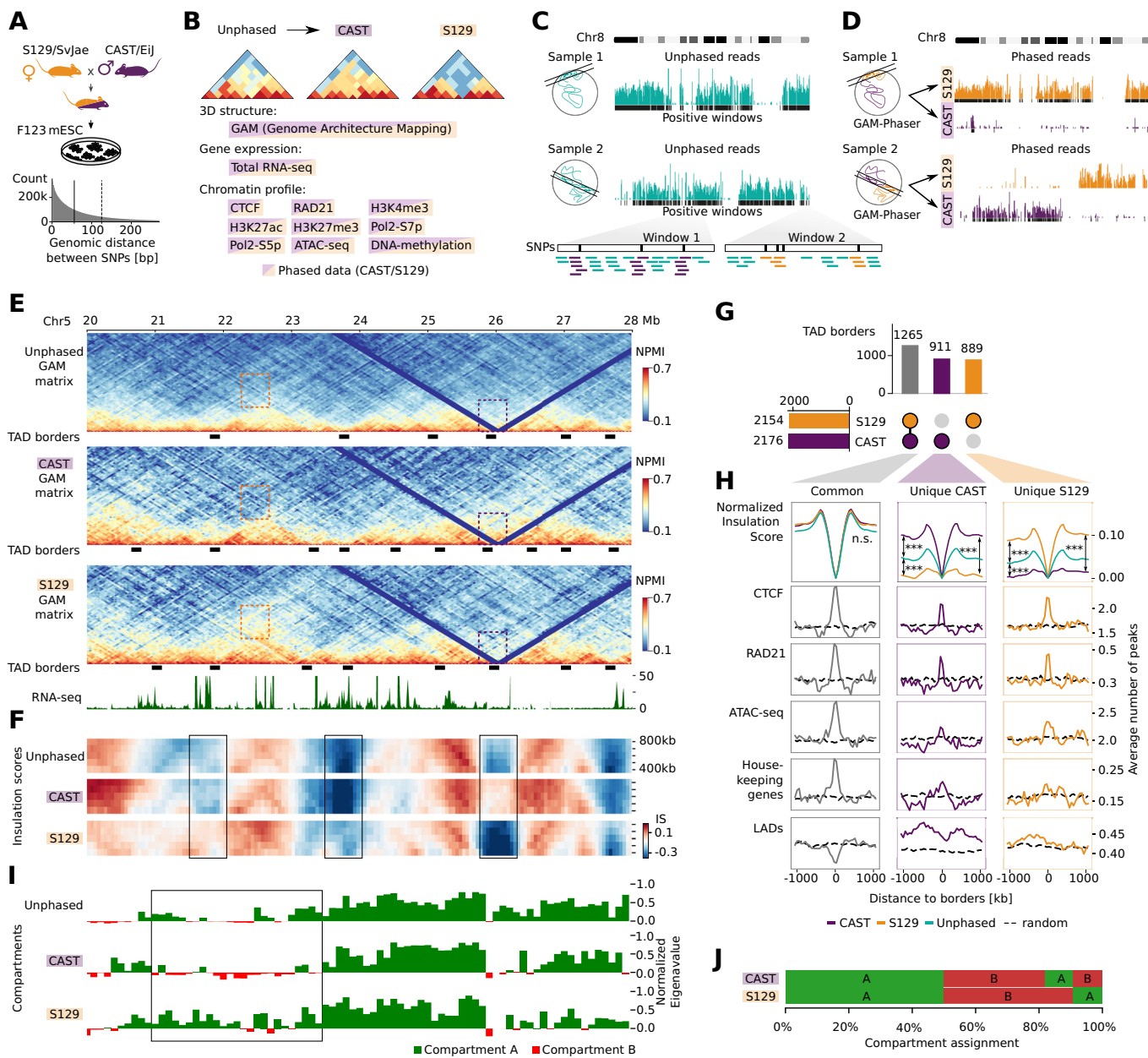

Hi-C data from equivalent hybrid mESC lines has achieved only 26 or 35% phasing efficiency of ligation events (Fig. EV1E; Giorgetti et al, 2016; Bonora et al, 2021). Comparison of informative contact entries in phased GAM and phased Hi-C matrices from human GM12878 B-lymphoblastoid cells of lower SNP density (Rao et al, 2014b; Data ref: Rao et al, 2014a) shows the detection of 79–93% of all possible intrachromosomal contacts in GAM data, at 50 kb for all genomic distances, compared with only 20–51% in Hi-C data at the same resolution (Fig. EV1F). Genomic windows were rarely phased to both CAST and S129 haplotypes in the same nuclear profile (Fig. EV1D, dual phasing). To exemplify the low expected co-detection of allelic windows in the same nuclear slices, we took published imaging data using fluorescence in situ hybridization in thin cryosections (cryo-FISH) performed in a different ESC line (clone 46C), which targeted the genomic regions containing *Hoxb1*

or *Hoxb13* with fosmid probes covering ~40 kb (Barbieri et al, 2017; Dataset EV2). Cryo-FISH data analyses confirmed that a minority of nuclear sections contain both copies of each locus (13% and 11%, respectively; Fig. EV1G).

To determine a suitable resolution of the phased GAM data, we calculated the detectability of window co-segregation events at different genomic resolutions, as previously (Beagrie et al, 2017, Winick-Ng et al, 2021). For a robust analysis of allele-specific chromatin structure, we chose a window resolution of 50 kb for downstream analyses, which gives detection of >97% co-segregation events across all genomic distances (Fig. EV1H). Higher resolutions down to 10 kb also gave good co-segregation frequencies for unphased data; for example, 99% of all possible pairs of 10 kb genomic windows within 10 Mb were co-detected in at least one GAM sample.

◄   **Figure 1.   GAM shows structural differences between alleles.**

(**A**) F123 mESCs are derived from the cross between S129 and CAST mice. In the F123 genome, the median SNP distance is 55 bp (solid line), and the mean is 124 bp (dashed line). (**B**) Overview of the data used in this study. Two biological replicates were collected or available for all datasets, and one replicate for Pol2-S5p, Pol2-S7p and H3K27me3 ChIP-seq. (**C**) Schematics showing phasing from two GAM samples. Reads mapped to chromosome 8 are shown above positive windows. Each window can contain a different number of SNPs, and reads mapped to these regions are used for GAM phasing. (**D**) Phasing shows that most of the reads belong to one of the haplotypes; black bars below the phased reads represent phased positive genomic windows. Large sections of the chromosome are phased in GAM data. (**E**) Unphased and phased GAM maps, TAD borders and the total RNA-seq track are shown for chr5: 20–28 Mb. Colored rectangles mark differences in chromatin contacts between the CAST and S129 haplotypes, with orange and purple corresponding to increased number of contacts for S129 or CAST, respectively. (**F**) Heatmap of insulation scores calculated with square sizes that range from 400 to 800 kb. The insulation score heatmaps are represented for the same region as in (**E**). Boxes highlight regions with structural differences between CAST and S129. (**G**) UpSet plot shows the number of common and unique TADs to each haplotype. (**H**) Normalized insulation score in TADs categorized according to (**G**) is represented in a genomic window centered on the TSS ± 1,000 kb. Significance of insulation differences was determined with Mann–Whitney test (*** represents $P < 10^{-13}$ for all comparisons). $P$ values are as follows: common borders, CAST against S129: 0.83; CAST against unphased: 0.41; S129 against unphased: 0.28; CAST unique borders, CAST against S129: $8.9 \times 10^{-46}$; CAST against unphased: $2.7 \times 10^{-14}$; S129 against unphased: $1.7 \times 10^{-18}$; S129 unique borders, CAST against S129: $1.3 \times 10^{-47}$; CAST against unphased: $1.4 \times 10^{-17}$; S129 against unphased: $5.6 \times 10^{-17}$. Number of CAST, S129 and common borders are 911, 889, and 1265, respectively. Other plots show the average number of CTCF, RAD21 and ATAC-seq peaks, Housekeeping genes and LADs around the TSS. Dashed lines depict the expected number of features using circular permutations and averaging the score from 10 iterations. (**I**) A and B compartment annotations and normalized eigenvector values for unphased and phased matrices for the region shown in (**E**). Box highlights a region with notable differences between CAST and S129 matrices. (**J**) Compartment assignments show 18% different annotations between CAST and S129.

## GAM detects extensive haplotype-specific differences in chromatin contacts

To begin assessing the extent of haplotype-specific differences in chromatin contacts captured in GAM data, we compared unphased and phased contact matrices (Fig. 1E). We found extensive structural variability between the CAST- and S129-phased matrices, and noticed that both local and long-range contacts are stronger and more obvious in the haplotype-specific matrices than in the unphased, average matrices (Fig. 1E, orange and purple rectangles, respectively for strong contacts in S129 or CAST haplotypes). Structural variability between haplotypes becomes even more prominent when plotting whole chromosome matrices, where clusters of increased long-range contacts are clearly visible across large genomic distances (Fig. EV2A, orange and purple arrows, respectively). Contact distance decay and momentum curves showed similar frequency of contacts between haplotypes within <5 Mb of genomic distance, but became visibly distinct at long-range distances with different haplotype preferences depending on the chromosome (Fig. EV2B), suggesting that larger-scale properties contribute to allelic chromatin structures.

## Most TAD borders are haplotype-specific

To quantify haplotype-specific differences at the level of TAD organization, we calculated insulation scores at different length scales, using 400–800 kb square sizes (Crane et al, 2015; Winick-Ng et al, 2021), and found clear differences in insulation between parental genomes (Fig. 1F; see boxes and Fig. EV2C for an additional example; for insulation score data see permanent data repository Irastorza-Azcarate et al, 2024). Consistent with the unphased matrices being an average of the CAST- and S129-specific matrices, we confirmed that the CAST and S129 insulation scores correlated less with each other than with unphased insulation scores (400 kb insulation square sizes; Fig. EV2D).

Next, we computed TAD borders in unphased and phased matrices using the 400 kb insulation square size, as previously described (Winick-Ng et al, 2021). More than 40% of all TAD borders detected are haplotype-specific (911 and 889 unique to CAST and S129) compared with 1265 common borders detected in both haplotypes (Fig. 1G; Dataset EV3; for all combinations see

Fig. EV2E). The distinct insulation between haplotype-specific TAD borders was confirmed by comparing average insulation plots (Mann–Whitney test, ***$P < 10^{-13}$ for all comparisons; Fig. 1H). Many borders common to both haplotypes were also detected in unphased matrices (1061), as expected, and some CAST- and S129-specific borders could also be detected in the unphased matrices but not in the other haplotype (373 and 351, respectively; Fig. EV2E), likely reflecting their strong prevalence in one haplotype chromosome across the cell population. However, many CAST- and S129-specific borders were not captured in the unphased matrices (514 and 518, respectively), highlighting the specificity of chromatin topology in the two haplotypes.

We asked whether haplotype-specific TAD borders were enriched for CTCF, cohesin or housekeeping (HK) genes, as previously shown for unphased TAD borders (Dixon et al, 2012). CTCF and cohesin were found highly enriched in both common and haplotype-specific borders, whereas housekeeping genes and chromatin accessibility (ATAC-seq) are more strongly enriched in common borders (Fig. EV2E). We also noted a preference for common borders to more likely correspond to LAD/interLAD transitions (22.9%) than CAST- and S129 unique borders (14.4% or 11.4%, respectively; Fig. EV2F).

## Haplotype-specific compartments account for 20% of the genome

Previous work in mouse T cells from B6xCAST hybrid mice detected only 4% of compartment changes between haplotypes (Han et al, 2020), and region-specific examples of compartment changes have also been reported at specific loci in hybrid mESCs (Rivera-Mulia et al, 2018). To quantify the extent of haplotype differences in compartment A/B annotation genome-wide in GAM data, we computed eigenvector values from principal component analysis (PCA) from unphased and haplotype-specific GAM matrices (Fig. 1I, box; see also whole chromosome regions in Fig. EV2G; Dataset EV4). Genome-wide analyses showed compartment changes between the two alleles in 18% of the genome, in contrast with 49% and 32% of the genome being annotated A–A or B–B, respectively (Fig. 1J). We observed that the distributions of eigenvector values of allele-specific matrices are more symmetrical compared to the unphased values and cover a wider range, suggesting that compartmentalization states are better captured

in the phased data than in the haplotype-averaged unphased data (Fig. EV2H).

To investigate the functional consequences of extensive haplotype-specific differences in chromatin structure on gene expression and their relationship with chromatin-based mechanisms of gene regulation, we quantified the haplotype-specific differences in gene expression and chromatin features in F123 mESCs, and characterized their co-occurrence across the linear genome. Subsequently, we integrated allele-specific 3D genome structure with the linear distribution of allele-specific gene expression and chromatin occupancy.

## Allele-specific expressed genes are enriched in housekeeping, ribosomal, and histone gene groups

To understand the extent of allele-specific gene expression in F123 mESCs, we measured gene expression from total RNA-seq data for protein-coding and long noncoding genes, after selecting the most expressed transcript isoform (based on the levels of Pol2-S5p and Pol2-S7p at annotated transcription start sites; see "Methods"). We calculated differential allelic expression as previously described (Castel et al, 2015), considering both exonic and intronic regions (for gene expression levels see permanent data repository Irastorza-Azcarate et al, 2024). Out of 17,956 expressed genes, we detected 13,713 genes similarly expressed from both alleles, 2222 genes with ASE imbalance ( | log2 fold change | ≥1, adjusted $P$ value ≤ 0.05 and TPM ≥ 1), of which 1308 and 914 genes were more expressed from the CAST or S129 genomes, respectively, and 2,021 genes were expressed without SNP (Fig. 2A). ASE genes are all genes exhibiting expression imbalance, while monoallelic genes are a subgroup of ASE genes which contain only reads from one allele. Among the ASE genes, we found 193 monoallelic genes, including several histone genes, such as *Hist2h2ac*, 15 imprinted genes, including *Lin28a* and *Peg13*, all more expressed from the CAST allele, and *Cdkn1c*, more expressed from the S129 allele (Fig. 2B). The paternal allele more frequently exhibited higher expression, as previously reported in murine embryonic fibroblasts and adult tissues (Crowley et al, 2015; Savol et al, 2017).

Gene Ontology (GO) enrichment analysis showed that ASE genes are involved in metabolic processes, immunity response and encode ribosomal proteins (Fig. 2C; Dataset EV5). Separate GO enrichment analysis of CAST and S129 ASE genes showed no haplotype-specific enrichment of biological functions. Amongst the ASE genes, we found many ribosomal protein genes (32%, 30/94; Fig. 2D), a group of genes which were previously reported to be expressed with allelic imbalance in other mouse tissues or cell types, and in Medaka and catfish tissues (Murata et al, 2012; Pinter et al, 2015; Crowley et al, 2015; Chen et al, 2016). The ASE gene list also contained 38% of all histone genes (26/69) and 8% of housekeeping genes (Dataset EV6). The ASE imbalance of histone genes has been reported in mouse embryonic fibroblasts, and can also be observed by mining publicly available resources from mouse tissues (Crowley et al, 2015; Pinter et al, 2015; Savol et al, 2017), but has so far not been investigated.

## H3K27me3 occupies a third of ASE gene promoters

ASE imbalance is thought to be achieved by repression mechanisms acting on one allele (Garcia et al, 2014), and some studies report a major role of Polycomb repression in monoallelic expression,

predominantly at the maternal allele (Inoue et al, 2017; Santini et al, 2021). To explore whether Polycomb repression mechanisms are also important more generally in expression imbalance, we mapped H3K27me3, Pol2-S5p, and Pol2-S7p occupancy in F123 mESCs.

Previous genome-wide analyses in mESCs showed that the promoters of signaling or metabolic genes are often occupied by H3K27me3, Pol2-S5p and Pol2-S7p, in a mixed Polycomb-Active (PRCa) promoter state, thought to result from allele-specific deposition of Polycomb or fluctuations between active and Polycomb repression in different cells (Brookes et al, 2012; Ferrai et al, 2017). To investigate whether ASE imbalance relates with direct Polycomb occupancy on the promoters of ASE genes, we classified all non-overlapping gene promoters in F123 mESCs according to their H3K27me3, Pol2-S5p or Pol2-S7p occupancy. One-third of ASE genes have PRCa promoter states (H3K27me3 + S5p + S7p + , 477 genes; Fig. 2E), including signaling and metabolic genes, such as *Mapk13* and *Apoe*, respectively (gene promoter classification tables are available in the permanent data repository Irastorza-Azcarate et al, 2024). Consistent with the repressive effects of Polycomb, the expression of ASE genes marked by H3K27me3, S5p and S7p is approximately half of the expression levels of ASE genes occupied by S5p and S7p only (Fig. EV3A).

To further explore a functional role of Polycomb in the repression of ASE genes, we took advantage of published RNA-seq data for AID-mediated acute depletion of the catalytic subunit of PRC1 (RING1A[KO] RING1B[AID] in E14-tg2a ES cells; Dobrinić and Klose, 2021b; Data ref: Dobrinić et al, 2021a). We found that 290 ASE genes are both marked by H3K27me3 in F123-ESCs, and are upregulated upon AID-induced RING1B depletion in the AID-E14-ESCs (Fig. 2F, permanent data repository Irastorza-Azcarate et al, 2024). Other ASE genes may also be under Polycomb influence, as they are characterized by H3K27me3 occupancy in F123-ESCs (219 genes) or by being upregulated in the AID-induced RING1B depletion in E14 ESCs. These analyses also confirm that many ASE genes (395 genes) are not associated with Polycomb repression or occupancy, and are likely regulated by other mechanisms.

## Chromatin features at ASE gene promoters are mostly biallelic

To further investigate other chromatin-mediated mechanisms that might contribute to ASE imbalance, we applied peak finders to ATAC-seq, H3K4me3, H3K27ac, CTCF, cohesin (RAD21), H3K27me3, Pol2-S5p and Pol2-S7p data (peaks coordinates are provided in the permanent data repository Irastorza-Azcarate et al, 2024). Occupancy peaks were relatively short (on average 305–1507 base pairs; Fig. EV3B), except for H3K27me3 and Pol2 modifications (on average 2000–3400). Most peaks could be phased (41–86%), but they were not allele-specific (biallelic peaks; Fig. 2G). A minority of phased peaks were classified as CAST (2.4–4.6%) or S129 (0.6–3.1%) specific, and were slightly more abundant in the CAST haplotype, a preference also observed in the number of CAST ASE genes.

We measured the overlap between ASE gene promoters and haplotype-specific peaks, and found that most ASE gene promoters coincide with biallelic peaks of ATAC, H3K4me3, H3K27ac, and Pol2-S5p/S7p, less frequently with CTCF and H3K27me3, and rarely with RAD21 (Fig. 2H). CAST-specific peaks that overlap with ASE gene promoters are preferentially found at promoters of genes more highly expressed from the CAST haplotype (1–16%

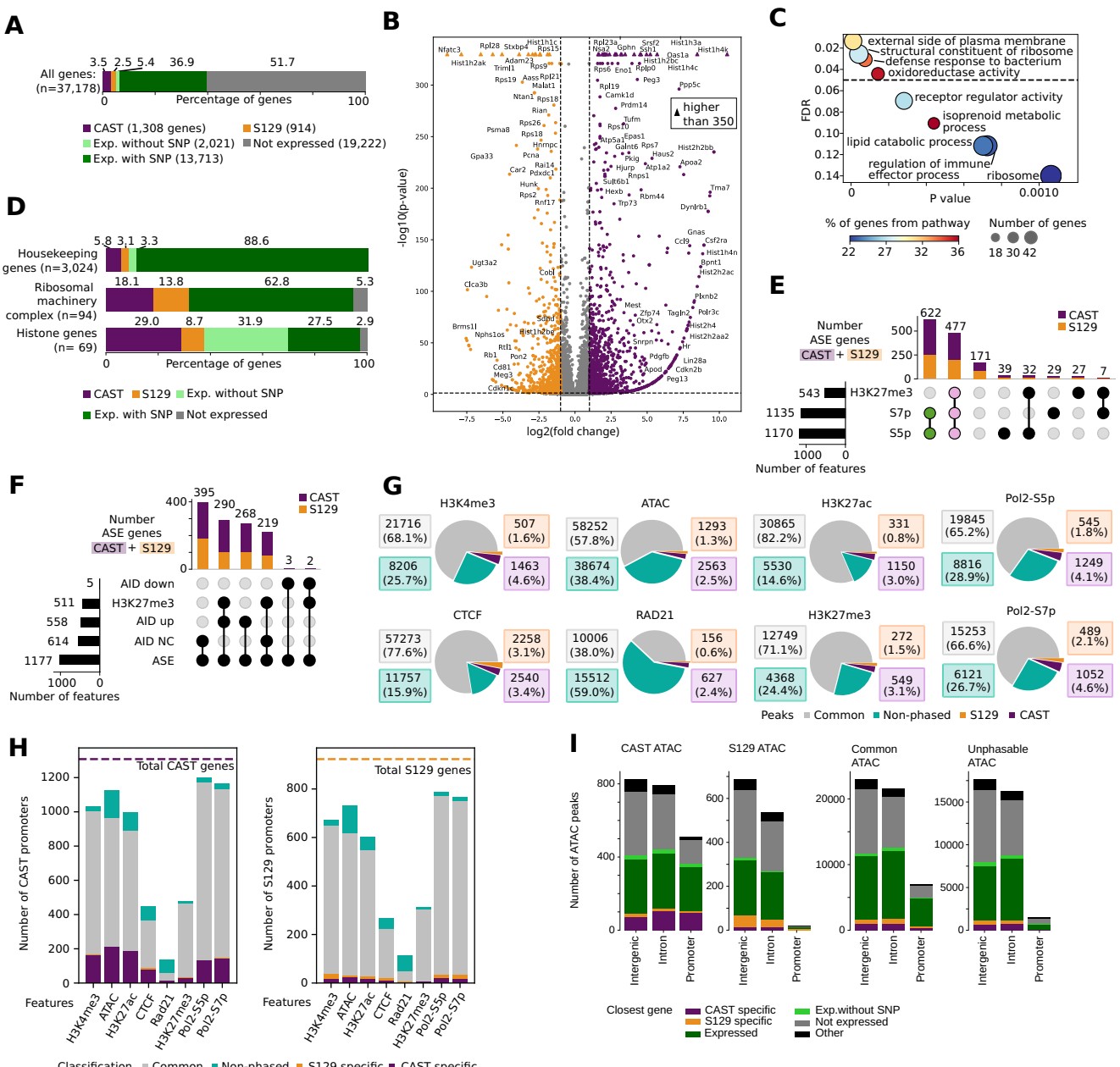

**Figure 2. Allele-specific expressed (ASE) genes are enriched for housekeeping, ribosomal protein and histone genes, and many contain Polycomb.**

(**A**) Number and percentage of CAST and S129 ASE genes expressed with no SNP, genes that are biallelic, and genes that are not expressed. (**B**) Volcano plot of all expressed genes containing SNPs. Genes with a |log2 fold change| ≥1 and an adjusted P value of ≤0.05 were classified as ASE. Number of genes: 37,178. (**C**) Significant Gene Ontology (GO) terms of ASE genes. (**D**) Bar plot showing percentage of genes classified as housekeeping, ribosomal machinery complex and histone proteins. (**E**) Overlap of CAST and S129 ASE genes with H3K27me3, Pol2-S5p, and Pol2-S7p. (**F**) Overlap of CAST and S129 genes with H3K27me3 peaks and upregulated genes (AID up), downregulated genes (AID down) or genes with no change (AID NC), for AID-mediated acute depletion of the catalytic subunit of PRC1 (Dobrinić and Klose, 2021b). (**G**) Number and percentage of H3K4me3, H3K27ac, H3K27me3, Pol2-S5p, Pol2-S7p, CTCF and RAD21 ChIP-seq and ATAC-seq peaks that are CAST or S129-specific, common or could not be phased. (**H**) Number of CAST (top) and S129 gene promoters (bottom) that overlap with different features. (**I**) The number of CAST, S129, common and non-phased ATAC-seq peaks in intergenic, intron, and promoters. The color indicates the type (gray: not expressed; dark green: biallelic expressed; green: expressed with no SNP; orange: S129-specific; purple: CAST-specific; black: unknown) of the closest gene to the ATAC peak.

depending on chromatin feature), in contrast with S129-specific peaks which rarely coincide with S129-specific promoters (0.4–2%). This tenfold haplotype imbalance is unlikely to be technical, as, for example, the detection of CAST and S129 ATAC peaks is almost even (2.5% and 1.3%, respectively). Amongst the ASE gene

promoters marked by H3K27me3 (511 genes, Fig. 2F), only a minor fraction have allele-specific occupancy of H3K27me3 (43 ASE gene promoters; Fig. 2H). The presence of H3K27me3 at ASE gene promoters, the upregulation of ASE genes following acute depletion of Polycomb catalytic subunits, and the limited allele-specificity at the

most repressed allele, suggest that Polycomb repression may contribute to ASE imbalance through other mechanisms, possibly via its presence in intergenic regions or gene bodies, or through structural folding effects.

## Allele-specific intergenic regulatory regions are often close to ASE genes expressed in the same haplotype

To explore allele-specific long-range effects in ASE imbalance, we considered ATAC peaks in proximity to ASE gene promoters, in intergenic and intronic regions. We found that CAST ATAC peaks are preferentially nearest to CAST ASE genes, while S129 ATAC peaks are closer to S129 ASE genes (Fig. 2I). The observation that both CAST- and S129-specific ATAC peaks have a preference for proximity to ASE genes more expressed in the same haplotype suggests a role for enhancer–promoter (E–P) chromatin contacts in ASE imbalance. We searched for transcription factor motif enrichment at CAST or S129 ATAC peaks present at promoters, intergenic or genic regions, and found a single transcription factor, ZFP57, enriched in CAST-specific peaks at CAST gene promoters, and not in other promoters or genomic regions (Fig. EV3C; for list of motifs in ATAC-seq peaks see permanent data repository Irastorza-Azcarate et al, 2024). ZFP57 is a zinc finger protein involved in the maintenance of imprinted genes through binding of DNA methylated regions (Mackay et al, 2008, Shi et al, 2019). We also explored the association of ASE imbalance with differential methylation in F123 mESCs. After identifying differentially methylated genes from published phased whole bisulfite sequencing data in F123 mESCs (Li et al, 2019; Data ref: Li et al, 2019), we found that only 61 ASE gene promoters were found associated with allele-specific DNA methylation (2.7% of all ASE genes; Fig. EV3D; Dataset EV7), including three imprinted genes, *Mest, Snrpn* and *Peg13* (Dataset EV6). *Mest* is a CAST ASE gene which shows stronger chromatin contacts at the maternal than paternal *Mest* locus (Fig. EV3E), in line with previous reports in neonatal and adult neurons, using Dip-C (Tan et al, 2021).

To complete the exploration of linear chromatin features and their association with ASE imbalance, we considered CTCF and RAD21 peaks. Most CTCF peaks are biallelic (57,273), and only 2540 and 2258 are CAST or S129-specific, respectively (Fig. 2G). In contrast, most Rad21 peaks could not be phased (15,512), and only 627 or 156 peaks were assigned to the CAST or S129 alleles, respectively. Although CTCF peaks rarely overlap ASE gene promoters (Fig. 2H), some allele-specific CTCF peaks overlap promoters of monoallelic expressed genes, for example, *Cdkn2b* and *Hist2h4* (see permanent data repository Irastorza-Azcarate et al, 2024). These results suggest that CTCF and RAD21 are not a major feature of ASE imbalance. We also noticed that haplotype-specific CTCF peaks are present at approximately one quarter of TAD borders, but they always co-occur with biallelic CTCF peaks, and show no preference for borders of the matching haplotype (Fig. EV3F,G), suggesting that CTCF-mediated mechanisms are not general drivers of haplotype-specific TAD formation.

## CAST-specific ASE genes and chromatin features co-occur in the linear genome

Previous studies in mouse and medaka tissues reported ASE gene clustering in the linear genome (Garcia et al, 2014; Crowley et al, 2015).

We inspected the position of ASE genes in F123 mESCs across whole chromosomes and confirmed a tendency for ASE gene clustering in mESCs (Figs. 3A and EV4A). Genome-wide analyses showed that CAST and S129 genes are present in all autosomal chromosomes, tend to be clustered, and are often intermingled with each other (circular Permutation test, *P* values = 0.0001, 0.0145, 0.0001 for CAST, S129, and CAST + S129; Fig. EV4B). ASE genes are located in genomic regions with high density of expressed genes compared with regions without ASE genes (*t* test: *P* value = $2.7 \times 10^{-71}$; Fig. EV4C).

Next, we measured the genomic overlap and clustering of ASE genes and active chromatin features, and found that CAST genes and chromatin features often co-occur with each other, in contrast with S129 features and genes which rarely co-occur (Fig. EV4D,E), potentially due to parental effects as shown in different tissues (liver, brain, lung and kidney) or cells, including in F123-derived fibroblasts, where expression tends to be more abundant from the paternal allele (Crowley et al, 2015; Savol et al, 2017). CAST and S129 features are generally segregated along the linear genome, as shown by the minor co-occurrence of CAST features or CAST genes with S129 features or S129 genes (Fig. EV4E).

Taken together, our exploration of the linear organization of ASE genes and chromatin accessibility and occupancy suggests that different mechanisms may control ASE gene expression, of which Polycomb occupancy and repression were most often associated with ASE imbalance. In the next sections, we investigated how these linear genome features relate with haplotype differences in 3D genome structure.

## ASE clustering occurs preferentially within compartment A

We asked whether ASE gene clustering was reflected in haplotype-specific compartment transitions, and found 107 CAST and 79 S129 ASE genes present in genomic regions with A–B or B–A (CAST-S129) compartment assignments, with a preference for ASE genes to be more expressed in the compartment A (euchromatic) annotation of the corresponding haplotype (Fig. 3B; Chi-squared test, *P* = 0.015). Most other ASE genes are present in compartment A annotations (Fig. EV5A; circular Permutation test, 10,000 permutations, *P* value = 0.0001), a tendency that is likely driven by their preferred co-occurrence with biallelic expressed genes.

Haplotype-specific ATAC, CTCF, H3K4me3 and H3K27ac peaks were also found preferentially associated with haplotype-specific compartment differences, with CAST-specific peaks being slightly more abundant in A–B (CAST-S129) compartments, and S129-specific peaks in B–A regions, with the exception of S129-specific CTCF peaks which are equally distributed in A–B and B–A regions (Fig. EV5B).

## ASE genes are clustered within TADs enriched for H3K27me3 occupancy

We then asked whether ASE gene clustering in the linear genome reflects the TAD organization. ASE genes are present in 45% of TADs, in all cases together with biallelically expressed genes (Fig. 3C). Approximately one-third of TADs contain CAST ASE genes, another third S129 ASE genes, and the last third contain both (Fig. 3C for CAST TAD annotations, Fig. EV5C for S129 TAD

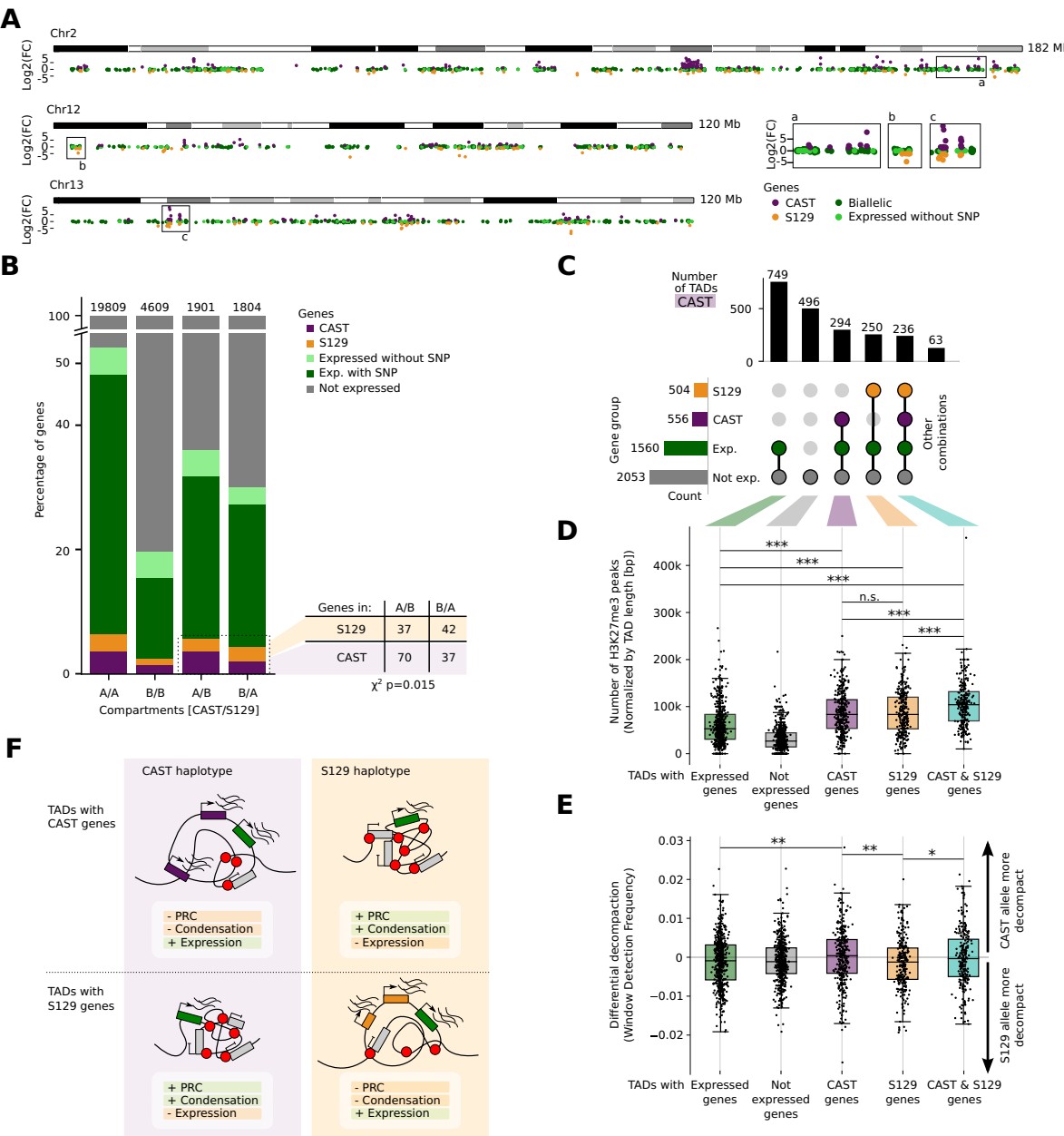

**Figure 3. ASE genes are clustered in TADs enriched for H3K27me3 occupancy.**

(A) Manhattan plot showing log2 fold change of all expressed genes for chromosomes 2, 12, and 13. Genes are colored according to their expression as CAST or S129-specific, biallelic or do not contain SNPs. Boxes show regions that were zoomed: a region with the majority of CAST ASE genes, majority of S129 ASE genes or a mix of CAST and S129 ASE genes. (B) Bar plot showing the percentage of genes that overlap with A or B compartments in both haplotypes, or have different compartment annotations in CAST and S129. The preferred tendency for CAST genes to be in CAST compartment A, and S129 genes to be in S129 compartment A is statistically significant (Chi-square test= 0.015. Number of A/B with S129 genes: 37, and with CAST genes: 70. Number of B/A with S129 genes: 42, and with CAST genes: 27). (C) UpSet plots showing, for the CAST allele (S129 allele in Fig. EV5C), groups of TADs containing different sets of types of genes and their number. (D) For each group in (C), the number of H3K27me3 peaks normalized by TAD length (two-sided $t$ test: *$P < 0.05$, **$P < 0.01$, ***$P < 0.001$; $P$ values from top to bottom in CAST TADs: $1.0 \times 10^{-16}$, $7.0 \times 10^{-15}$, $1.9 \times 10^{-32}$, $4.4 \times 10^{-5}$, n.s: 0.95, $7.6 \times 10^{-5}$). Number of TADs with: expressed genes, 749; not expressed genes, 496; CAST genes, 294; S129 genes, 250; and TADs with CAST and S129 genes, 236. The center of each box plot represents the median, the box boundaries correspond to the Q1 and Q3 quartiles, and the whiskers extend from the box to the farthest data point lying within 1.5× the interquartile range (IQR) from the box (Q1-1.5 IQR and Q3 + 1.5 IQR, respectively). (E) The differential (CAST-S129) window detection frequency is represented for each group in (C). Negative values indicate decompaction in the S129 haplotype, while positive values indicate decompaction in CAST (two-sided $t$ test: *$P < 0.05$, **$P < 0.01$, ***$P < 0.001$; $P$ values from top to bottom for CAST TADs: 0.008, 0.003, 0.018). Number of TADs with: expressed genes, 749; not expressed genes, 496; CAST genes, 294; S129 genes, 250; and TADs with CAST and S129 genes, 236. The center of each box plot represents the median, the box boundaries correspond to the Q1 and Q3 quartiles, and the whiskers extend from the box to the farthest data point lying within 1.5× the interquartile range (IQR) from the box (Q1-1.5 IQR and Q3 + 1.5 IQR, respectively). (F) Summary model displaying differences in chromatin compaction of TADs containing CAST or S129 ASE genes.

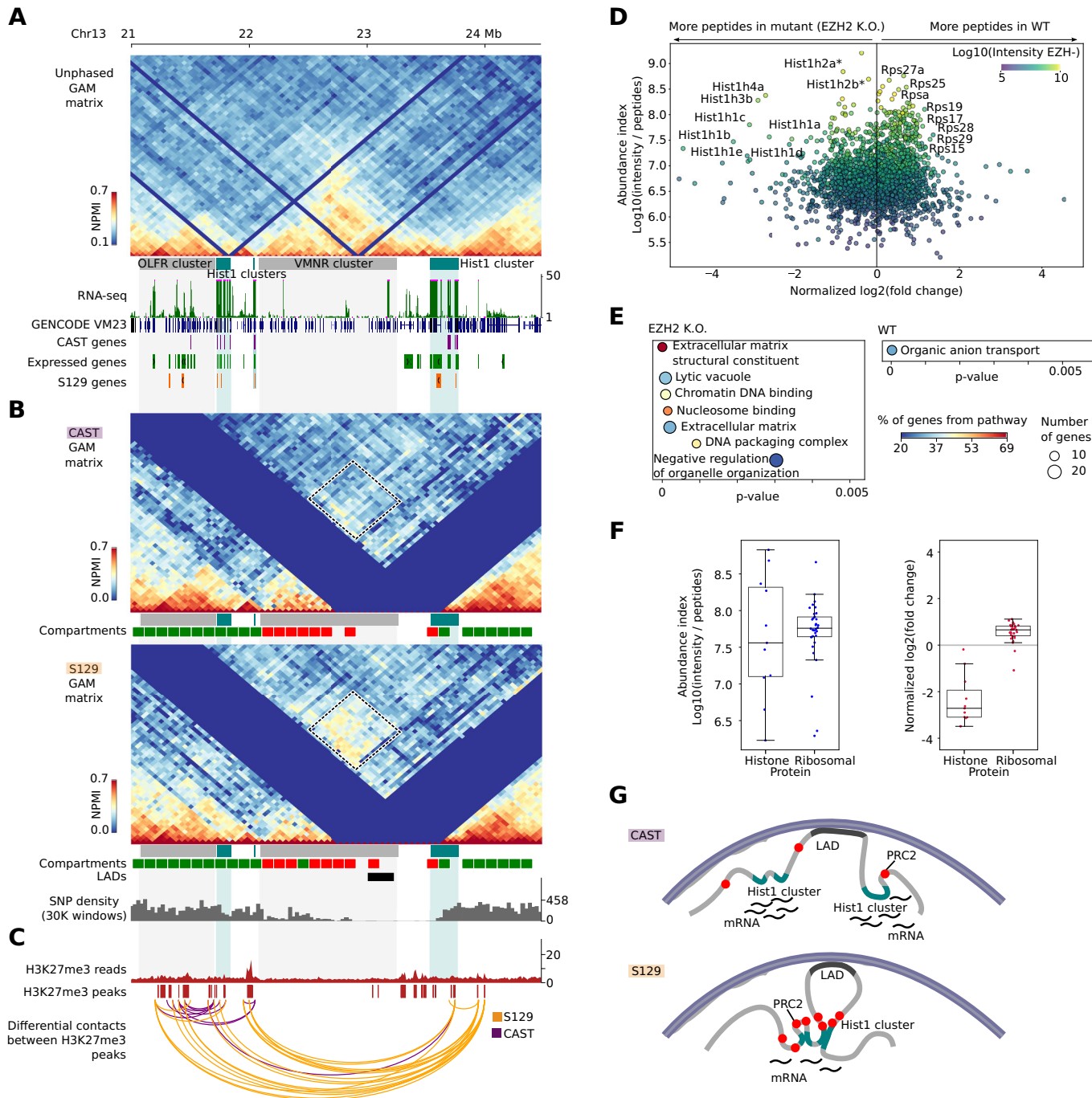

annotations). ASE gene clustering within TADs is statistically significant for CAST, S129 or combined CAST/S129 ASE genes (Permutation test, 10,000 permutations, all *P* values ≤ 0.0001; Fig. EV5D,E).

We asked whether TADs containing ASE genes are also enriched for H3K27me3 peaks, irrespective of whether they were mono- or biallelic, and found a statistically significant enrichment compared with TADs containing only biallelic expressed genes or silent genes (*t* test, *P* values < 0.001; Figs. 3D and EV5F). The H3K27me3 enrichment is especially strong in TAD annotations containing both CAST and S129 ASE genes, compared with TADs

containing ASE genes of only one haplotype, suggesting allele-specific local contributions of H3K27me3 to ASE imbalance. Since chromatin compaction is a feature of Polycomb activity in vitro and in vivo across short and long genomic regions (Nichols et al, 2020; Barbieri et al, 2017; Schoenfelder et al, 2015), we first asked whether the differential presence of CAST or S129 ASE genes within specific TADs correlated with increased chromatin decompaction in the haplotype with the larger number of expressed genes. We took advantage of the fact that the GAM technology inherently detects relative differences in chromatin compaction (Beagrie et al, 2017), based on the fact that genomic windows with the same DNA

**Figure 4. Histone genes in the Hist1 cluster establish S129-specific contacts that coincide with H3K27me3 occupancy, and are regulated by Polycomb repression mechanisms.**

(A) Unphased GAM map of the Hist1 locus (chr13: 21.0–24.5 Mb). Below, tracks showing the position of each Hist1 cluster, olfactory receptor cluster and the VMNR cluster; total RNA-seq data, position of all genes, expressed genes, and genes specific to CAST and S129 alleles. (B) Phased maps of the same region to the CAST and S129 allele. Below, SNP density track at 30 kb windows, showing a region which contains part of the VMNR cluster and the rightmost Hist1 cluster, devoid of SNPs. The rectangle highlights contacts between the Hist1 clusters which are strong in the S129 allele, and weak in the CAST allele. (C) Tracks for H3K27me3 reads and peaks. Below, allele-specific contacts for each allele extracted from the phased GAM maps that coincide with H3K27me3 peaks. (D) Mass spectrometry SILAC experiments carried out in ESC-Ezh2-1.3 cells grown in the absence or presence of tamoxifen to induce conditional knockout of *Ezh2*, in three biological replicates, each with two technical replicates. *Ezh2* knockout results in upregulation of histone proteins. *Abundance* was calculated as intensity divided by number of peptides, while *normalized log2fc* was calculated applying the z-score normalization to the log2 of heavy/light (H/L) ratio of the WT experiment divided by the H/L ratio of the conditional knockout. Data points labeled with an asterisk represent peptides common to several histone genes: Hist1h2a* represents Hist1h2ah, H2afj, Hist1h2ak, Hist1h2af, Hist3h2a, Hist2h2a and Hist2h2aa1; while Hist1h2b* represents Hist1h2bk, Hist1h2bf, Hist1h2bp and Hist1h2bb. (E) Gene Ontology terms for the top 5% upregulated genes for each condition. (F) Boxplots showing the abundance index and the log2 fold change for histone proteins and ribosomal proteins related to (D). Numbers of data points are 11, 34, 11, and 34, respectively from left to right. The center of each box plot represents the median, the box boundaries correspond to the Q1 and Q3 quartiles, and the whiskers extend from the box to the farthest data point lying within 1.5× the interquartile range (IQR) from the box (Q1-1.5 IQR and Q3 + 1.5 IQR, respectively). (G) Proposed model for the Hist1 locus folding and gene regulation. Haplotype-resolved GAM data shows that the Hist1 clusters come together preferentially in the S129 allele. These contacts may be mediated by Polycomb which establishes a repressive environment and thus results in lower overall expression. The Hist1 clusters in the CAST allele are spatially separated which coincides with increased gene expression.

content but different compaction are detected across the collection of GAM nuclear slices (NPs) proportionally to their physical volume (Fig. EV5G). We measured the window detection frequency (WDF) of genomic windows (Dataset EV8), and found that genomic regions within TADs containing only CAST or only S129 ASE genes have on average higher WDF, i.e., are more decondensed, in the most expressed allele irrespective of haplotype (Figs. 3E and EV5H). Increased WDF is also observed at the gene level, as genomic windows containing the most expressed allele are also more decondensed (Fisher's exact test $P = 5.3 \times 10^{-5}$; Fig. EV5I). The observation that TADs with ASE imbalance are associated with Polycomb occupancy and increased compaction of the repressed allele, provides orthogonal support for a role of Polycomb repression in chromatin condensation genome-wide, which is shown here in the context of haplotype-specific chromatin regulation (Fig. 3F).

## Long-range interactions in the Hist1 gene cluster are allele-specific

To further explore how the linear clustering of ASE genes relates to allelic differences in higher-order chromatin contacts, we considered the Hist1 locus which contains 19 ASE histone genes. The Hist1 locus is the largest and densest of the four histone loci, and contains three Hist1 subclusters (~200, ~10, and ~500 kb) in a 2 Mb region, harboring a total of 55 histone genes interspersed with two silent clusters of sensory receptor genes, Olfr and Vmnr (Fig. 4A). The Vmnr cluster is annotated as B compartment in unphased GAM data, and a LAD region, flanked by active histone genes in compartment A. Most histone genes in the Hist1 locus are expressed in F123 mESCs (52 out of 55 genes) of which 35 contain SNPs. Of the 19 ASE genes in the locus, 14 and 5 genes are more highly expressed from the CAST or S129 allele, respectively (Fig. EV6A), indicating that the Hist1 cluster is more transcriptionally active in the CAST than the S129 chromosome copy.

Unphased GAM data shows that the three Hist1 locus subclusters interact with each other (Fig. 4A), establishing long-range contacts that resemble those found at the human Hist1 locus in ESCs by SPRITE (Quinodoz et al, 2018). As the Hist1 locus has a robust density of SNPs, except across the *Vmnr* gene cluster, it was possible to phase most of the region. In the haplotype-specific

GAM contact matrices, we found that the Hist1 locus shows extensive structural differences between CAST and S129 haplotypes (Fig. 4B), in particular a large S129-specific patch of strong contacts between the most distant Hist1 subclusters, separated by 1.5 Mb. As the S129 locus expresses fewer genes than the CAST locus, we hypothesized that the long-range contacts might relate to histone gene repression. H3K27me3 occupancy was detected at 11 out of 19 histone ASE genes, and their promoters are classified as PRCa (Fig. EV6B; see classification table in permanent data repository Irastorza-Azcarate et al, 2024). Although histone genes have not previously been reported as targets of Polycomb repression, evidence for the presence of H3K27me3 or mono-ubiquitinylated H2A (H2Aub1) at the promoters of histone genes can be traced in published mESC datasets for *Hist3h2ba* (Brookes et al, 2012), and *Hist2h3c1*, *Hist2h4*, *Hist3h2ba* genes (Ferrai et al, 2017) in different mESC lines.

## Histone genes are upregulated upon conditional Polycomb knockout

To explore potential roles of Polycomb repression in Hist1 gene regulation, we calculated differential contacts between CAST and S129 matrices (Fig. EV6C), and extracted all allele-specific contacts in the region involving windows containing H3K27me3 peaks (Figs. 4C and EV6D). We found that these long-range contacts connect all 3 clusters in the S129 allele, suggesting that the repression of a larger number of histone genes in the S129 haplotype may relate to local and long-range effects of Polycomb repression.

To directly address a functional role for Polycomb repression in the dampening of histone gene expression, we took advantage of two previously characterized conditional tamoxifen-inducible knockout cells of *Ring1b* (murine ESC-ERT2 clone; Stock et al, 2007) or *Ezh2* (murine ESC-Ezh2-1.3 clone; Pereira et al, 2010), which encode the major enzymatic activities of Polycomb repressor complex 1 (PRC1) or 2 (PRC2), respectively. Upon addition of tamoxifen, ESC-ERT2 and ESC-Ezh2-1.3 lose H2Aub1 or H3K27me3, respectively, within 24/48 h or 96 h (Stock et al, 2007; Pereira et al, 2010). We performed quantitative SILAC mass spectrometry analysis in the two cell lines, before and after knockout induction (Dataset EV9). We discovered that histone proteins were highly upregulated after knockout of either *Ring1b* or

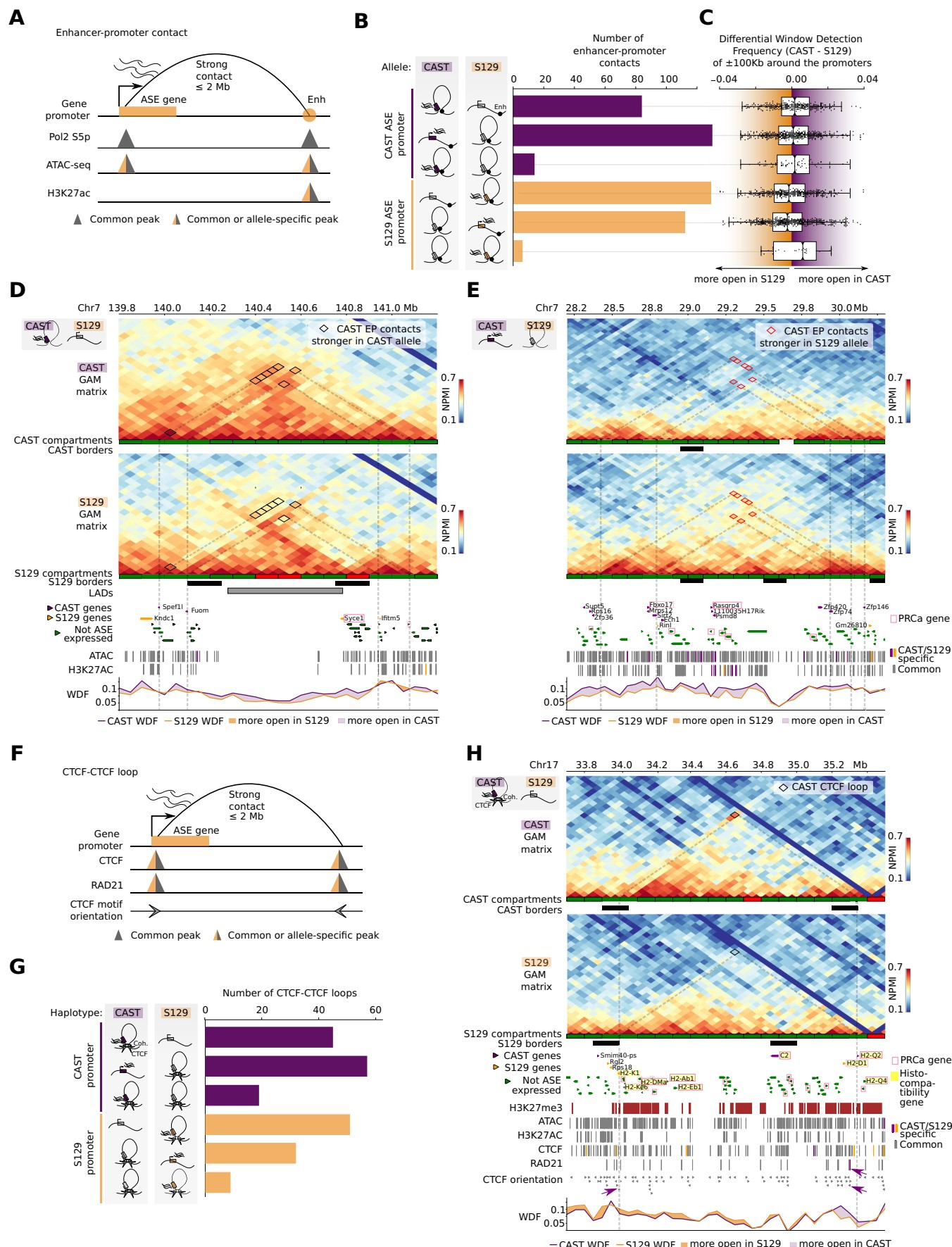

**Figure 5. Enhancer–promoter contacts and CTCF loops coincide with ASE genes.**

(A) Features and conditions used to define enhancer-promoter contacts (Enh, Enhancer). (B) Quantification of enhancer–promoter contacts, depending on the different configurations in each allele. (C) Box plot showing the normalized window detection frequency (WDF) for each configuration. Numbers from top to bottom are: 85, 131, 14, 130, 113, and 6, respectively. The center of each box plot represents the median, the box boundaries correspond to the Q1 and Q3 quartiles, and the whiskers extend from the box to the farthest data point lying within 1.5× the interquartile range (IQR) from the box (Q1-1.5 IQR and Q3 + 1.5 IQR, respectively). (D) Example region with contact differences on chr7 with allele-specific enhancer–promoter contacts. (E) Example for decondensation in the CAST allele with allele-specific enhancer–promoter (E–P) contacts. (F) Features and conditions used to define CTCF loops. (G) Quantification of CTCF loops with Cohesin (Coh.), depending on the different configurations in each allele. (H) Contact map illustrating an allele-specific CTCF loop on chr17. The track for CTCF orientation shows the directionality of CTCF motifs. Purple arrows point to CTCF motifs with convergent orientation involved in the CTCF loop with CAST-specific Rad21 peak in one of the anchors.

*Ezh2*, and were among the proteins with the highest fold change upregulation (Figs. 4D and EV6E). In fact, GO enrichment analysis on proteins with 5% highest fold change shows enrichment for terms associated with DNA packing complex and nucleosome binding proteins (Fig. 4E; Dataset EV5). In contrast, ribosomal proteins, also abundantly expressed and characterized by allelic expression imbalance, are not upregulated upon conditional Polycomb knockout, supporting the view that ASE imbalance is not exclusively regulated by Polycomb repression (Figs. 4F and EV6F, respectively, for *Ezh2* and *Ring1b* knockouts).

Our observations show that many histone genes are ASE genes regulated by Polycomb repression mechanisms, with promoters occupied by Pol2-S5p, -S7p and H3K27me3. We also show that histone genes within the Hist1 locus establish long-range chromatin contacts, often occupied by H3K27me3, which bridge a gene-silent LAD, and occur especially in the S129 haplotype that expresses fewer Hist1 genes (see schematics in Fig. 4G).

## Allele-specific contacts between ASE genes and enhancers, and CTCF

Next, we were curious about allele-specific contacts between ASE gene promoters and putative regulatory regions (enhancers; E), and whether E-ASE gene contacts would be predominant in the most or least expressed allele. To define a stringent list of E-ASE gene contacts, we selected the strongest contacts in each allele (z-scores >2.0; Fig. EV7a) that connect ASE genes also marked by Pol2-S5p and ATAC peaks, with putative enhancers also marked by Pol2-S5p, ATAC, and H3K27ac, within 2 Mb genomic distances (Fig. 5A; the table of differential contacts and features is available in GSE254717).

We first asked whether the selected E-ASE gene contacts are preferentially established from the most- or least-expressing allele. Similar numbers of E-ASE gene contacts were found in the most or least expressed allele, but rarely in both alleles, suggesting that allele-specific E-ASE gene contacts can alternatively coincide with the expression of the active allele or with the repression of the least expressed allele (Figs. 5B and EV7B). The differences in strength of contacts between the two haplotypes based on NPMI values were statistically significant (two sample *t* test; *P* values < 1e−9 for all comparisons; Fig. EV7C,D). Regardless of whether the strong E-ASE gene contact occurs in the haplotype where the ASE gene is most or least expressed, we found increased decompaction of the most expressed allele involved in a strong E-ASE gene contact, by comparing the WDF of the 150 kb genomic regions centered on the ASE gene promoters (Fig. 5C). These results show that enhancers can contact their putative target genes independently of their compaction or expression state, and confirm that allele-specific expression coincides with increased local decompaction of the expressed genomic region.

Strong E-ASE gene contacts that coincide with expression of the active allele are in line with models of increased gene expression driven by increased E–P contacts (Carter et al, 2002; Simonis et al, 2006; Noordermeer et al, 2011; Bartman et al, 2016; Barshad et al, 2023). For example, the genes *Fuom* and *Spef1l* are two CAST ASE genes which establish strong CAST contacts between themselves and enhancer-containing windows spanning a > 1 Mb genomic region which is contained within the same compartment A (Fig. 5D). In contrast, the S129 haplotype is characterized by fewer strong contacts across the whole region, and the presence of a S129-specific compartment B and a LAD interspersing the two CAST ASE genes. WDF measurements show the higher decompaction of the whole region in CAST than S129 haplotypes.

We also found examples of loss of strong E-ASE gene contacts in the most active allele, in line with enhancer mechanisms where increased transcriptional activity coincides with loss of E–P contacts (Benabdallah et al, 2019). For example, the gene *Zfp146* is a CAST ASE gene which establishes a strong E-ASE gene contact in the S129 allele, spanning 1.7 Mb. Other CAST genes that form strong E-ASE gene contacts in S129 are *Sirt2* and *Zfp74*, which contact each other (Fig. 5E). As previously, we find lower WDF in the silent S129 haplotype than CAST haplotype indicating that ASE gene expression is associated with increased decompaction in the expressing allele.

These results show that allele-specific expression can coincide alternatively with strong allele-specific E-ASE gene contacts or with loss of strong E-ASE gene contacts. Irrespective of whether the proximity to putative regulatory regions occurs in the active or repressed state, the most expressed allele is characterized by increased local chromatin decondensation, which may relate to the formation of transcriptional condensates (Cramer, 2019).

Finally, we searched for strong ASE gene contacts anchored by CTCF and RAD21 occupancy, which contained CTCF motifs in convergent orientation and were less than 2 Mb apart (Figs. 5F and EV7E). We found a small number of CTCF loops involving ASE genes (Fig. 5G), for example, for *Camk1d*, *Gnas* and *H2-Q2* genes (the table of differential contacts and features is available in GSE254717). *H2-Q2* is a CAST ASE gene within the Major Histocompatibility region which is involved in a strong CAST-specific CTCF loop with S129 ASE gene *H2-K1* (Fig. 5H). In total, four histocompatibility genes are ASE genes: *C2* and *H2-Q2* are CAST and *H2-K1* and *H2-D1* are S129 ASE genes. The contact is mediated by two common CTCF peaks with convergent orientation and two cohesin peaks, in which the peak in the right anchor is CAST-specific (purple arrows). The CTCF-mediated loop in the CAST allele may favor the expression of *H2-Q2* but not of *H2-K1* in the CAST allele, while the absence of CTCF loop in the S129 correlates with S129 expression of *H2-K1* but not *H2-Q2*. These results suggest that some CTCF loops may be involved in ASE imbalance. However, we observe that these genomic regions are also under Polycomb regulation, for example, in specialized cells such as in

oligodendroglia (Meijer et al, 2022). Histocompatibility genes are also marked by Polycomb histone marks in mESCs and throughout different stages of differentiation of mESCs to neuronal lineages (Ferrai et al, 2017). Taken together, these examples showcase the complex interplay between different mechanisms of chromatin and gene regulation and the challenges in the interpretation of the extensive allele-specific differences in 3D genome structure.

## Discussion

Many genes are expressed with allelic imbalance due to a combination of genetic differences between the two chromosome copies, and parental-specific epigenetic mechanisms often attributed to Polycomb repression or DNA methylation (Ohishi et al, 2019; Savol et al, 2017; Lappalainen et al, 2013; Crowley et al, 2015; Marion-Poll et al, 2021; Inoue et al, 2017; Santini et al, 2021). While extensive folding differences between the active and inactive chromosome X copies have been reported using Hi-C (Giorgetti et al, 2016; Tan et al, 2018), few differences in 3D chromatin structure have been reported in autosomes based on ligation-dependent methods (Llères et al, 2019; Rao et al, 2014b; Han et al, 2020), likely due to the sparsity of SNPs in the genome and the requirement for SNP presence on both sides of ligation events (Rivera-Mulia et al, 2018). In high SNP density mouse crosses, the maximum fraction of phased ligation events are capped at one-third of all sequenced ligation events (Giorgetti et al, 2016). These difficulties have been discussed and currently motivate the development of imputation or machine learning approaches that extrapolate unphased events (Miller and Adjeroh, 2024), but these methods require independent validation.

In GAM technology, chromatin contacts are inferred by spatial sampling of chromosome structure through slicing nuclei in thin slices, and sequencing of the genomic content of each slice (Beagrie et al, 2017). Chromatin contacts are measured from the co-segregation of genomic windows across the collection of nuclear slices. As the length of each window is typically 10–50 kb, each window contains many nucleotide polymorphisms, such that the phasing can be done with high sensitivity. For example, in hybrid F123 mESCs, 50-kb windows contain an average of 385 SNPs. The GAM sampling process therefore makes the phasing of genomic windows highly efficient, with successful phasing of about 75% of all detected genomic windows in F123 mESC GAM datasets.

In this study, we collected the largest GAM dataset to date from the F123 hybrid mESC line, and developed a novel pipeline termed GAM-Phaser to phase GAM data. Phased GAM data revealed an unprecedented level of structural differences between autosomes, at all scales of 3D genome structure and across all autosomes, demonstrating the power of window-based approaches to map haplotype-specific differences in chromatin structure.

Mapping ASE imbalance using total RNA-seq detected approximately 2,222 ASE genes, of which 193 are monoallelically expressed. Many ASE genes are housekeeping, with roles in metabolism and signaling, and enriched for genes encoding for ribosomal subunits and histone genes. By mapping the occupancy of Pol2-S5p, Pol2-S7p and H3K27me3, we found that ASE genes often have features of bivalent chromatin and mixed Polycomb-Active promoter states, previously reported in mESC lines and throughout neuronal differentiation (Brookes et al, 2012; Ferrai

et al, 2017). Many, but not all, ASEs gene promoters are marked by H3K27me3, and are upregulated upon acute degradation of Polycomb enzymatic subunits. Among the ASE genes with Polycomb-Active promoter states, we found 25 ASE histone genes, 19 of them located within the Hist1 cluster. ASE genes are present in gene-dense regions, intermingled with, or close to, biallelic genes, suggesting that the repression of ASE genes in one allele is likely specific to gene and genomic neighborhood, and not related with 3D chromatin structure in a trivial manner. Allele-specific DNA methylation is not a major feature of ASE imbalance as it occurs at a minority of ASE genes, as suggested previously (Kerkel et al, 2008), especially at monoallelic genes. Nevertheless, we discovered that ZFP57, a transcription factor involved in the maintenance of imprinted genes through binding of DNA methylated regions, is specifically enriched at the promoters of CAST ASE genes, suggesting a parental contribution of DNA methylation to ASE imbalance.

The extensive differences in 3D genome structure between the two copies of each chromosome were observed at all levels of 3D genome organization, both locally and spanning large genomic distances. Haplotype-specific GAM data detected differences in compartment A/B annotation in 18% of the genome, a much larger proportion than the 4% previously reported using Hi-C in cells with similar SNP density (Han et al, 2020). We also found extensive differences in chromatin insulation at the level of topologically associating domains (TADs). The majority (59%) of all TAD borders detected in CAST and S129 alleles are allele-specific, and characterized by CTCF and cohesin enrichment, albeit to a lower extent than TAD borders present in both haplotypes. Allele-specific CTCF occupancy on chromatin is generally rare (6.5% of all CTCF peaks), and mostly occurs inside TADs, suggesting that haplotype-specific TAD border formation is not simply based on haplotype-specific CTCF occupancy. We found that TAD organization is related to ASE gene clustering, with ASE genes being present in only half of all TADs, and their presence coinciding with increased H3K27me3 occupancy. By assessing chromatin compaction directly from GAM data, we found that Polycomb occupancy coincides genome-wide with increased compaction of the least expressed allele, adding to previous in vitro and in vivo observations (Nichols et al, 2020; Barbieri et al, 2017).

We explored in more detail the Hist1 locus, which contains 19 ASE histone genes in F123 mESCs. Chromatin contacts within the Hist1 cluster are highly haplotype-specific and most prominent in the S129 allele characterized by decreased histone gene expression. The Hist1 cluster is abundantly covered by H3K27me3-marked chromatin (25% of 50 kb windows are positive for H3K27me3 peaks), and many of the haplotype-specific contacts in the S129 genome occur between genomic windows marked by Polycomb occupancy. To functionally test a role for Polycomb repression in histone gene downregulation, we performed mass spectrometry after tamoxifen-induced knockouts of the two major enzymatic subunits of Polycomb Repressor Complexes, PRC1 (*Ring1b*) and PRC2 (*Ezh2*). Histone protein levels were found highly upregulated upon Polycomb knockout, in contrast with ribosomal proteins, showing that histone genes are targets of Polycomb repression mechanisms. Further work will be necessary to investigate how the increased S129-specific contacts at Hist1 locus relate to the lower expression of specific histone genes in each haplotype, and how the ASE imbalance of specific histone genes relates to the cell cycle, the

histone locus body or Polycomb bodies (Ghule et al, 2008; Nizami et al, 2010; Quinodoz et al, 2018).

Finally, we found that allele-specific expression can coincide with strong allele-specific E-ASE gene contacts or with loss of strong E-ASE gene contacts, but not both, in the same gene. Allele-specific CTCF loops were also rare but occasionally associated with ASE genes. Immune system genes were found at haplotype-specific CTCF loops, and are ASE genes in other biological systems, including in F1 crosses between goats and Ibex, or between modern humans and Neanderthals, and associated with disease (Yang et al, 2022, McCoy et al, 2017). Moreover, histocompatibility genes are susceptible to cis-regulation variants (Gutierrez-Arcelus et al, 2020). The observation that histocompatibility genes form highly haplotype-specific contacts in a haplotype- or parental-specific manner indicates a role for 3D genome structure in the diversity of major histocompatibility complexes and the capacity of the immune system evolution, which requires further work in relevant biological systems (Sommer, 2005).

Overall, the variety of chromatin regulatory mechanisms connected with ASE imbalance suggests that it is tuned by combinations of different mechanisms and is highly gene-specific (Crowley et al, 2015; Marion-Poll et al, 2021). These findings also demonstrate the value of haplotype-specific 3D genome structure to help address mechanisms of disease due to genetic variation or epigenetic deregulation of genes. Future questions and limitations of the present study are the contribution of parental versus genetic sequence effects, which can be addressed by mapping allele-specific differences in the alternative cross (S129xCAST) and using other genotype crosses. Further efforts are required to understand the stability and evolution of allele-specific chromatin structures in differentiation and in different cell lineages. The detected differences between CAST/paternal and S129/maternal phasing of local features open new questions about parental-specific epigenetic mechanisms acting on ASE imbalance, which require further in-depth study. Further work is also necessary to enable the phasing of GAM from human samples, which are characterized by ten times lower SNP densities than F123 mESCs. Finally, it is still an open question to what extent phasing the allele-specific topology of the two chromosome copies can help to interpret the effects of genetic variation on gene (de)regulation, towards a deeper understanding of genome biology and gene regulation mechanisms.

# Methods

### Reagents and tools table

| Reagent/resource | Reference or source | Identifier or catalog number |
| --- | --- | --- |
| **Experimental models** | | |
| F123 mESC cells (hybrid cell line, derived from a F1 *M. musculus* S129/Jae and *M. castaneous* mouse cross) | Gribnau et al, 2003 | |
| mESC-ERT2 Ring1A-/- cells | Stock et al, 2007 | |
| mESC-Ezh2-1.3 cells | Pereira et al, 2010 | |
| **Antibodies** | | |
| Mouse anti-RNAP2 S5p (clone CTD4H8) | BioLegend | 904001 |
| Rat anti-RNAP2 S7p (clone 4E12) | Chapman et al, 2007 | Prof Dr Dirk Eick, Helmholtz-Zentrum-München, Germany |
| Rabbit anti-H3K37me3 | Millipore | 07-449 |
| **Oligonucleotides and other sequence-based reagents** | | |
| GAT-7N | Biomers | 5'- GTG AGT GAT GGT TGA GGT AGT GTG GAG NNN NNN N |
| GAT-COM | Biomers | GTG AGT GAT GGT TGA GGT AGT GTG GAG |
| **Chemicals, enzymes, and other reagents** | | |
| DMEM | Invitrogen | 11995065 |
| KnockOut™ DMEM | Invitrogen | 10829-018 |
| KnockOut Serum Replacement | Invitrogen | 10828028 |
| L-glutamine, 200 mM Solution | Invitrogen | 25030-024 |
| MEM Non-Essential Amino Acids Solution, 100X | Invitrogen | 11140-035 |
| 2-Mercaptoethanol | Invitrogen | 31350-010 |
| ESGRO® (LIF) | Millipore | ESG1107 |
| Gelatin | Sigma-Aldrich | G1393 |
| CF-1 IRR | Global Stem | GSC-6201G |
| PCR Mycoplasma Test Kit | AppliChem | A3744,0020 |
| 4-hydroxytamoxifen | Sigma-Aldrich | H7904 |
| TRIzol Reagent | Invitrogen | 15596026 |
| Agilent RNA 6000 Nano Kit | Agilent | 5067-1511 |
| Turbo DNase I | Ambion | AM1907 |
| TruSeq Stranded total RNA library preparation kit | Illumina | 15031048 |
| Paraformaldehyde, 16% W/V | VWR | 43368.9M |
| Sucrose | Sigma-Aldrich | S0389 |
| PBS tablets | Sigma-Aldrich | P4417 |
| PEN membrane steel frame slides 4.0 µm | Leica Microsystems | 11600289 |

| Reagent/resource | Reference or source | Identifier or catalog number |
|---|---|---|
| Cresyl violet | Sigma-Aldrich | C5042 |
| PCR Cap Strip filled with opaque adhesive material | Carl Zeiss Microscopy | 415190-9161-000 |
| Guanidinium-HCl 8 M, pH 8.5 | Sigma-Aldrich | G7294 |
| Triton X-100 | Sigma-Aldrich | T9284 |
| Tween-20 | AppliChem | A4974 |
| EDTA 0.5 M, pH 8.0 | AppliChem | A4892 |
| Qiagen protease | Qiagen | 19157 |
| DeepVent® (exo-) DNA Polymerase | NEB | M0259L |
| Deoxynucleotide (dNTP) Solution Mix | NEB | N0447L |
| Quant-iT® PicoGreen dsDNA assay kit | Invitrogen | P7589 |
| Illumina Nextera XT library preparation kit | Illumina | FC-131-1096 |
| TruSeq ChIP Library Preparation Kit | Illumina | IP-202-1012 |
| High Sensitivity DNA analysis kit | Agilent | 5067-4626 |
| L-lysine, +8 Da | Cambridge Isotope Laboratories | CNLM291H |
| L-arginine +10 Da | Cambridge Isotope Laboratories | CNLM-539H |
| **Software** | | |
| Cutadapt | https://cutadapt.readthedocs.io/en/stable/, Martin, 2011 | |
| Burrows-Wheeler Aligner | https://bio-bwa.sourceforge.net, Li and Durbin, 2010 | |
| bcftools | http://samtools.github.io/bcftools/bcftools.html | |
| samtools | https://www.htslib.org/doc/samtools.html | |
| bedtools | https://bedtools.readthedocs.io/en/latest/, Quinlan and Hall, 2010 | |
| SNPsplit | https://www.bioinformatics.babraham.ac.uk/projects/SNPsplit/, Krueger and Andrews, 2016 | |
| Bismark software package | https://www.bioinformatics.babraham.ac.uk/projects/bismark/, Krueger and Andrews, 2011 | |
| bowtie2 (v 2.3.4.3) | https://bowtie-bio.sourceforge.net/bowtie2/index.shtml, Langmead and Salzberg, 2012 | |
| GEM-Tools suite | Marco-Sola et al, 2012 | |
| TAR (v 2.7.2c) | Dobin et al, 2012 | |
| HTSeq-count | Anders et al, 2014 | |
| DESeq2 | Love et al, 2014 | |

| Reagent/resource | Reference or source | Identifier or catalog number |
|---|---|---|
| The Genome Analysis Toolkit (GATK) 4.1.3.0 | | |
| ChromA | https://github.com/marianogabitto/ChromA, Gabitto et al, 2020 | |
| Bayesian Change-point Model (BCP) peak-finder | Xing et al, 2012 | |
| **Equipment** | | |
| Ultracryomicrotome | Leica Biosystems | EM UC7 |
| Laser microdissection microscope | Leica Microsystems | LMD7000 |
| NGS sequencer | Illumina | NextSeq500/550 |
| Liquid handling system | TTP | Mosquito HV |
| Bioanalyzer electrophoresis | Agilent | 2100 Bioanalyzer |
| PCR cycler | BioRad | C1000 |
| HPLC system | Eksigent | Eksigent |
| Mass spectrometer | Thermo | Orbitrap Velos |
| Bioruptor sonicator | Diagenode | Bioruptor Plus |

## Cell culture

F123 mESCs (a male, hybrid cell line) was, derived from a F1 S129/Jae and Cast mouse cross (Gribnau et al, 2003). Cells were cultured in a layer of mitotically inactivated feeder murine embryonic fibroblasts under standard conditions (DMEM, supplemented with 15% KSR, 1× Glutamax, 10 mM non-essential amino acids, 50 μM beta-mercaptoethanol, 1000 U/ml leukemia inhibitory factor, LIF). Before harvesting, mESCs were passaged onto feeder-free 0.1% gelatin-coated plates for at least 2 passages to remove feeder cells. As feeder removal results in reduced levels of LIF in the culture, the LIF concentration in the media was doubled when the cells were in feeder-free culture conditions. Cells were harvested after ~48 h at 70–80% confluency. F123 mESC batches all tested negative for Mycoplasma infection, performed according to the manufacturer's instructions (AppliChem, Cat#A3744,0020). F123 mESCs were obtained from the 4DN consortium (https://data.4dnucleome.org/biosources/4DNSRTNKUDSA). Cell line authentication was initially performed by the consortium and independently confirmed in this study using a set of SNPs specific to the F123 line.

ESC-ERT2 Ring1A$^{-/-}$ cells (Stock et al, 2007) were maintained in an undifferentiated state by co-culture on mitomycin-inactivated mouse embryonic fibroblasts on 0.1% gelatin-coated flasks in DMEM supplemented with non-essential amino acids, 2 mM L-glutamine, 0.1 mM 2-mercaptoethanol (all from Gibco), 20% FCS (Autogen Bioclear, Calne, UK) and 1000 U/ml of leukemia inhibitory factor (ESGRO-LIF, Chemicon/Millipore). For the Ring1B conditional deletion, ESC-ERT2 cells were plated feeder-free on gelatin-coated plates 12 h before supplementing the medium with 800 nM 4-hydroxytamoxifen (H7904, Sigma, Poole, UK), and grown for 48 h. ESC-ERT2 Ring1A$^{-/-}$ cells were regularly

tested for Mycoplasma infection as a service provided by MRC Laboratory of Medical Sciences. ESC-ERT2 Ring1A$^{-/-}$ cells were from the lab of Haruhiko Koseki, where the cell line was originally generated (Stock et al, 2007).

ESC-Ezh2-1.3 cells (Pereira et al, 2010) were maintained in an undifferentiated state by co-culture on mitomycin-inactivated mouse embryonic fibroblasts on 0.1% gelatin-coated flasks in Knockout DMEM supplemented with non-essential amino acids, 2 mM L-glutamine, 0.1 mM 2-mercaptoethanol (all from Gibco), 20% FCS (Autogen Bioclear, Calne, UK), 5% Knockout Serum Replacement (Invitrogen), and 1000 U/ml of leukemia inhibitory factor (ESGRO-LIF, Chemicon/Millipore). For the *Ezh2* conditional deletion, ESC-Ezh2-1.3 cells were plated feeder-free on gelatin-coated plates 12 h before supplementing the medium with 800 nM 4-hydroxytamoxifen (H7904, Sigma, Poole, UK), and grown for 96 h, including replating at 48 h. ESC-Ezh2-1.3 cells were regularly tested for Mycoplasma infection as a service provided by MRC Laboratory of Medical Sciences. ESC-Ezh2-1.3 cells obtained from the lab of Amanda Fisher, where the cell line was originally generated (Pereira et al, 2010).

## Total RNA sequencing

Total RNA was extracted from F123 mESCs using TRIzol Reagent (Invitrogen, Cat# 15596026) following the manufacturer's instructions. Total RNA was analyzed on the Bioanalyzer using the Agilent RNA 6000 Nano Kit to ensure intact, non-degraded RNA presence and was subsequently treated with TURBO DNase I (Ambion, Cat# AM1907). Total RNA-seq libraries were generated from 1 μg of DNase-treated RNA using the TruSeq Stranded total RNA library preparation kit (Illumina, Cat# 15031048) according to the manufacturer's instructions. Samples were pooled and paired-end (75 bp) sequenced using an Illumina NextSeq500/550 sequencer, following the manufacturer's instructions.

## Genome architecture mapping (GAM)

Fixation of F123 mESCs was performed as described previously (Beagrie et al, 2017). Briefly, mESCs were grown to 70% confluency, media was removed, and cells were fixed in 4% and 8% paraformaldehyde in 250 mM HEPES-NaOH (pH 7.6; 10 min and 2 h, respectively), gently scrapped, and softly pelleted, before embedding (>2 h) in saturated 2.1 M sucrose in PBS, and frozen in liquid nitrogen on copper sample holders. Frozen mESC samples are stored indefinitely in liquid nitrogen. Two independent biological replicates were collected.

Ultrathin cryosections were cut with a glass knife using an ultracryomicrotome (Leica Biosystems, EM UC7) at ~230 nm thickness, and transferred to UV-irradiated PEN membrane steel frame slides 4.0 μm (Leica Microsystems, 11600289) for laser microdissection. Before laser microdissection, cryosections were washed in sterile-filtered, molecular biology grade, 1× PBS (3 times, 5 min each) to remove the sucrose, sterile-filtered water (3 times, 5 min each), and stained with sterile-filtered 1% (w/v) cresyl violet (Sigma-Aldrich, C5042) in water, for 10 min, followed by two washes with water (30 s each). Individual nuclear profiles (NPs) were isolated using a laser microdissection microscope (Leica Microsystems, LMD7000). NPs were collected in a PCR Cap Strip filled with opaque adhesive material (Carl Zeiss Microscopy,

415190-9161-000). For each collection day, 1 or 2 caps were left empty and taken through the whole-genome amplification (WGA) and sequencing process as a negative control for quality control purposes (labeled as "0NP" samples in Dataset EV10).

Whole-Genome Amplification (WGA) was performed as described previously (Winick-Ng et al, 2021) with minor modifications. Briefly, DNA was extracted from NPs at 60 °C in the lysis buffer (20 mM Tris-HCl pH 8.0, 1.4 mM EDTA, 560 mM guanidinium-HCl, 3.5% Tween-20, 0.35% Triton X-100) containing 0.75 units/ml Qiagen protease (Qiagen, 19155). After 24 h of DNA extraction, the protease was heat-inactivated at 75 °C for 30 min and the extracted DNA was amplified via two rounds of PCR. At-first quasi-linear amplification was performed with random hexamer GAT-7N primers with an adapter sequence. The lysis buffer containing the extracted genomic DNA was mixed with 2× DeepVent mix buffer (2× Thermo polymerase buffer (10x), 400 μm dNTPs, 4 mM MgSO$_4$ in ultrapure DNase free water), 0.5 μM GAT-7N primers (5′-GTG AGT GAT GGT TGA GGT AGT GTG GAG NNN NNN N) and 2 units/μl DeepVent® (exo-) DNA polymerase (New England Biolabs, M0259L), and incubated for 11 cycles in the BioRad thermocycler. The second exponential PCR amplification was performed in the presence of 1x DeepVent mix, 10 mM dNTPs, 0.4 μM GAT-COM primers (5′-GTG AGT GAT GGT TGA GGT AGT GTG GAG) and 2 units/μl DeepVent (exo-) DNA polymerase in the programmable thermal cycler for 26 cycles.

## GAM library preparation and high-throughput sequencing

After whole-genome amplification, the DNA WGA product was purified with SPRI beads (1.7x) The DNA concentration of each sample was quantified using the Quant-iT® PicoGreen dsDNA assay kit (Invitrogen #P7589). Genomic sequencing library was prepared from 1 ng of purified DNA using the Illumina Nextera XT library preparation kit (Illumina #FC-131-1096), following the manufacturer's instructions or with a reduced volume of reagents to 20%. The library preparation step was done either manually or using TTP Mosquito HV liquid handling system, as specified in Dataset EV10. After the library preparation, DNA was again purified with in-house SPRI beads (1.7×) and equal amounts of DNA from each sample was pooled together (up to 196 samples) for the sequencing. The final pool of libraries was purified two more times with SPRI beads (1.7×) and analyzed using DNA High Sensitivity on-chip electrophoresis on an Agilent 2100 Bioanalyzer. The samples were sequenced on an Illumina NextSeq500/550 sequencer as single-end 75 bp reads, according to the manufacturer's instructions.

## SNP calling in F123 hybrid (N-masked genome generation)

The high SNP density of the F123 genome was used to phase the reads from sequenced GAM libraries to the maternal and paternal haplotypes. For generating haplotype-specific calls for the hybrid F123 (CAST×S129) cells, the parental genome sequencing data from publicly available databases was used. The genome sequence of *Mus musculus castaneus* was downloaded from the European Nucleotide Archive (accession number ERP000042). *Mus musculus musculus S129/SvJae* genome sequence data was downloaded from

the Sequence Read Archive (accession number SRX037820). Read trimming was performed using Cutadapt (https://cutadapt.readthedocs.io/en/stable/, Martin, 2011) and mapped the reads to the mm10 genome assembly using the Burrows-Wheeler Aligner (https://bio-bwa.sourceforge.net, Li and Durbin, 2010). SNP location and sequence were identified using bcftools (http://samtools.github.io/bcftools/bcftools.html). SNPs that were detected in less than 5 reads, and quality below 30 were excluded from the analysis.

## Phasing of GAM data with the GAM-Phaser pipeline

GAM-Phaser is a pipeline developed here for GAM data phasing, summarized in Fig. EV1B. GAM-Phaser takes as input a VCF file (file containing position of SNPs) and raw GAM sequencing data (fastq files) and outputs haplotype-specific GAM window segregation tables for each haplotype considered. GAM-phaser takes advantage of the existing SNPsplit package (Krueger and Andrews, 2016) to mask high-quality paternal and maternal SNPs with N-character in a genome. The mm10 reference genome assembly was used (Dec. 2011, GRCm38/mm10).

GAM-Phaser generates an N-masked genome using the information about the genomic coordinates of SNPs and the reference genome assembly via the SNPsplit package. At the next step, raw GAM sequencing data are mapped to the N-masked genome using default parameters of bowtie2 (version 2.3.4.3; Langmead and Salzberg, 2012). The reads mapped to the N-masked genome are checked for the presence or absence of a SNP, and sorted to the haplotype-specific bam-files with SNPsplit package. Next, the genome is partitioned into equal-sized windows, and the coverage of all reads, CAST-phased reads and S129-phased reads is computed using bedtools for all collected F123 GAM libraries (Quinlan and Hall, 2010). Afterwards, the optimal threshold between the sequencing noise and the signal is determined separately for each GAM sample of the total dataset. The optimal threshold of nucleotide coverage for calling positive windows is calculated as the lowest coverage per bin that gives the highest percent of windows that have at least one neighboring positive window on at least one side. Windows are phased to the CAST haplotype when the number of nucleotides covered by the reads containing CAST SNPs is higher or equal to the optimal threshold, and to the S129 allele when the number of nucleotides covered by the reads containing S129 SNPs is higher or equal to the optimal threshold.

## Quality control of GAM samples

After read mapping and positive window calling, the quality of each GAM sample in the dataset collected was assessed to ensure that the laser microdissection, DNA extraction and subsequent experimental steps were successful. Quality control metrics calculated for each GAM sample include the number of uniquely mapped reads to the mouse genome, the percentage of orphan windows (windows without at least one neighbor) and the percent of total genome coverage. To exclude potentially cross-contaminated samples, Jaccard similarity index was calculated between the sequences of positive and negative windows from all GAM samples that were processed together on the same 96-well plate, as previously reported (Winick-Ng et al, 2021). Samples with a Jaccard similarity index >0.4 were excluded from the data analysis as potentially cross-contaminated. A sample was considered to be of good quality if it had <60% orphan windows, >50,000 uniquely

mapped reads and did not appear as cross-contaminated (Fig. EV8A). The detailed quality metrics for all samples including sequencing depth are provided in Dataset EV10. Out of 2234 GAM samples collected, 1986 (88.9%) passed quality control, according to the sample quality criteria.

The final GAM dataset was composed of 3707 high-quality nuclear profiles (NPs), and sampled from two biological replicates: 863 NPs were collected in 3NP mode (549 from replicate 1 and 314 from replicate 2), 8 NPs in 2NP mode (replicate 2), while 1,122 NPs were collected in 1NP mode (from replicate 1) and combined to 3NP in silico (see Datasets EV11 and EV12), as described previously (Winick-Ng et al, 2021, Beagrie et al, 2023) (Fig. EV1A).

## Randomization, blinding, and sample size

Randomization and blinding were not relevant for the current study. The experiments and the subsequent analyses were performed on the F123 mESC line, for which no clinical trial, treatment or disease comparison was performed.

The appropriate number of samples for a GAM dataset varies and depends on multiple parameters such as nuclear volume, level of chromatin compaction, and quality of DNA extraction (Beagrie et al, 2017; Winick-Ng et al, 2021; Beagrie et al, 2023). In previous work, we have explored mathematically how the number of GAM samples affects different variables (Beagrie et al, 2023; Extended data Fig. 3B,C,F, therein); for example, a GAM dataset collected in 3NP multiplex mode can detect contacts that occur with probabilities of at least 20% across the cell population with 1600 nuclear slices, for all intrachromosomal genomic distances. To take into account technical variations in the efficiency of DNA extraction from each GAM sample, the optimal resolution of a given GAM dataset is calculated upon data collection. To determine optimal resolution for each GAM dataset, we use nonparametric Kendall rank correlation coefficient to ensure good detection of all possible intrachromosomal co-segregation events (further details below). The F123 GAM dataset in the present study has median intrachromosomal co-segregation frequency of 11–16 for 50 kb windows, and 4–6 for 20 kb windows, depending on chromosome, with a distance cutoff of 10 Mb. When higher resolution data is required, depending on the goals of the project, further GAM samples can be collected from the same frozen cells, which are kept indefinitely in liquid nitrogen.

For total RNA-seq experiments in F123 mESCs, libraries were generated from two biological replicates, to account for experimental variability. No statistical method was used to predetermine sample size. The information about the read length and the sequencing depth is provided in Dataset EV1.

For ChIP-seq experiments, no statistical method was used to predetermine sample size. The information about the read length and the sequencing depth is provided in Dataset EV1.

## Determining the resolution of pairwise co-segregation matrices

The quality of chromatin contact maps from GAM data can be defined by two main metrics: resolution (genomic length) of genomic bins, and contact detectability (number of entries in the contact matrix which were observed at least once). The effective resolution of GAM datasets generally depends on the number of NPs collected (Beagrie et al, 2023), as each GAM sample contains only 5–15% of the genome, and enough

NPs are necessary to sample the co-segregation of all possible genomic windows in each chromosome. The reads sequenced in each GAM sample are used to identify the presence or absence of genomic bins in that sample in a binary fashion that does not directly affect the sensitivity to detect contacts. The chromatin contacts are defined as normalized co-segregation frequencies between genomic bins, and their sensitivity depends on how many events are counted (i.e., how many GAM samples were collected). Since each GAM sample has so little DNA (5–10% of the DNA of a single cell), the sequencing depth required to detect positive windows in each sample is promptly saturated with a low sequencing depth of 2–3 million reads per sample NP in the present data. This is approximately double the depth used in the first GAM manuscript, of 1–2 million (Beagrie et al, 2017; Beagrie et al, 2023). To assess the quality of genome sampling in the F123 datasets, the distribution of raw co-segregation events for all intrachromosomal pairs of genomic windows was compared to the standard Poisson distribution, at different resolution(s) and genomic distance(s) using a nonparametric Kendall rank correlation coefficient. The calculation of raw co-segregation events was followed by Yeo–Johnson power transformation. Standard Poisson distribution was computed using the mean and the standard deviation derived from the distribution of the real co-segregation events at each tested resolution(s) and genomic distance(s). Kendall's $\tau$ correlation coefficient $\geq 0.95$ was considered as the indication of good quality of genome sampling at the specified resolution and genomic distance.

## GAM data normalization

Raw co-segregation GAM matrices were normalized using normalized pointwise mutual information (NPMI) for all pairs of windows genome-wide, as previously described (Winick-Ng et al, 2021). NPMI describes the difference between the probability of a pair of genomic windows being found in the same NP given both their joint distribution and their individual distributions across all NPs. For visualization purposes, scale bars were adjusted to a range 0 and the NPMI value corresponding to the 99th percentile of all NPMI values for each genomic region displayed.

## Window detection frequency calculation

Window detection frequency (WDF) is a GAM specific parameter representing the relative of times each genomic window is captured in the whole GAM dataset (Beagrie et al, 2017). If we consider two loci with equal genomic length but different compactions, the actual volume of the more compact locus is relatively smaller than the least compact locus, even though they have the same DNA content. Through the slicing process of GAM data collection, loci with larger volume are captured more frequently than loci with smaller volume.

The WDF of each genomic window is calculated from the GAM segregation tables, by dividing the total number of samples in which this window is called positive by the total number of samples in the dataset.

$$\text{WDF(genomic window)} = \text{Number of samples with positive window} / \text{Total number of samples}$$

WDF was calculated from the combined 3DN-GAM segregation tables at 50 kb resolution, separately for CAST and S129 alleles (Dataset EV8).

## Identification of undersampled regions in GAM contact matrices

The WDF of genomic windows was also used to exclude from further analyses the genomic windows which are insufficiently sampled in the GAM process. WDF scores from across all windows in the genome follow a normal distribution. To detect outliers, a smoothing algorithm was applied to the WDF values per chromosome in stretches of eleven consecutive 50 kb genomic windows. Next, normalized delta (ND) was calculated for each window, according to

$$\text{ND} = (\text{raw\_Signal} - \text{smoothed\_Signal}) / \text{smoothed\_Signal}$$

If the ND is larger than a fold change of 5, the window is excluded from the curated dataset.

Next, the four adjacent windows (2 upstream and 2 downstream) to the window being removed were also removed, to ensure good quality of sampling in the final GAM data used for further analyses. Finally, genomic bins with an average mappability score below 0.2 are removed. Genome mappability for mm10 mouse genome assembly was computed using GEM-Tools suite (Marco-Sola et al, 2012) setting read length to 75 nucleotides. The mean mappability score was computed for each genomic bin with bigWigAverageOverBed utility from Encode.

## Selection of a non-redundant gene list

The most expressed isoform for each gene was identified using the same strategy as in (Ferrai et al 2017) with some modifications. Briefly, a complete expression analysis table containing 39,261 unique genes and 88,437 isoforms was considered. Almost 20,000 genes ($n = 19,003$) had a single annotated isoform. For the remaining 20,258 genes, a single isoform was selected based on the following criteria: (i) gene isoform with the highest amount of reads for Pol2-S5p in the 2-kb window centered on the TSS (14,051 genes); (ii) if ambiguity was still present, the gene isoform with the highest amount of reads for Pol2-S7p in the 2-kb window centered on the TSS was selected (1623 genes); (iii) if ambiguity was still present, the longest gene isoform was selected (3827 genes); (iv) if ambiguity was still present, a random annotated isoform was selected (757 genes).

## Promoter state classification

To classify gene promoter states, we followed the same strategy as in (Ferrai et al, 2017) with some modifications. Briefly, gene promoters were considered positive for Pol2-S5p, Pol2-S7p, or H3K27me3 when: (i) the 2 kb windows centered on the TSS overlapped with a region enriched for the mark, and (ii) the amount of reads in the TSS window was above a threshold. The threshold was defined as the 5th percentile of the distribution of reads in the TSS window of positive genes. Overlapping genes (3558) and genes whose TSSs were in close proximity (6855) were excluded from the classification. In total, we identified 6435 active genes, 12,968 inactive genes, 1716 PRC repressed and 5082 Polycomb-Active genes.

## RNA-seq data analysis

RNA-seq data from F123 was processed for standard and allele-specific gene expression analysis. The quality of the paired-end RNA sequencing reads was verified using FASTQC (http://www.bioinformatics.babraham.ac.uk/projects/fastqc). No reads needed to be trimmed or removed due to quality concerns. The paired-end reads derived from RNA sequencing were mapped to the most recent mouse reference genome assembly mm10 (GRCm38.p6) using STAR (version 2.7.2c) (Dobin et al, 2012) under consideration of the current mm10 annotation (downloaded from ensemble: ftp://ftp.ensembl.org/pub/current_gtf/mus_musculus/Mus_musculus.GRCm38.98.gtf.gz) and available information of genomic variants in the mm10 F123 genome (described in SNP calling in F123 hybrid). Following recommendations about best practices for data processing in allelic expression analysis (Castel et al, 2015), duplicate reads were removed from the data using Picard MarkDuplicates (version 2.21.1: https://software.broadinstitute.org/gatk/documentation/tooldocs/4.1.3.0/picard_sam_markdupl icates_MarkDuplicates.php). Default options were used with the exception of REMOVE_DUPLICATES = TRUE. To quantify the overall expression of genes, mapped reads overlapping exons and introns were assigned to the respective genes and summarized as gene-specific count values using HTSeq-count (Anders et al, 2014). The use of HTSeq-counts to generate gene-level read count values is recommended by the gold standard tool used for differential gene expression analysis DESeq2 (Love et al, 2014). Options were set to count reads overlapping exons and introns of genes, accounting for the paired-end nature of reads, only considering primary alignments and the default minimal alignment quality of 10. The same annotation file was used as described before in the read mapping step.

Subsequently, TPM values were calculated by normalizing count values for gene length and library size. To differentiate between the expression of genes located on the two parental alleles, reads that overlap heterozygous genomic variants were counted in an allele-specific manner. Reads overlapping those heterozygous variants located within exons and introns of genes were counted using GATK ASEReadCounter (The Genome Analysis Toolkit (GATK) version 4.1.3.0: https://software.broadinstitute.org/gatk/documentation/tooldocs/4.1.3.0/org_broadinstitute_he llbender_-tools_walkers_rnaseq_ASEReadCounter.php). Subsequently, only genomic variants within regions of high mappability and with a minimum total coverage of 20 reads were considered to reduce the risk of introduced biases. In case multiple genomic variants were present within the same gene, the counts were aggregated over the gene in an allele-specific manner using the available haplotype information described above in the read mapping step. Aggregated counts were tested for significant allele-specific expression differences (binomial test vs 0.5), and the false discovery rate was controlled for by correcting resulting $p$ values for multiple testing using the Benjamini and Hochberg method. Genes were defined as differentially expressed by an adjusted $P$ value ≤ 0.05, a fold change (log2) ≥ 1 and TPM ≥ 1 between reads mapping to CAST and S129. The ASE ratio was calculated as the ratio of read counts supporting the CAST haplotype and total read count. The Log2 fold change was defined as the log-scaled ratio of reads supporting the CAST haplotype divided by the read count observed in the S129 haplotype.

## Gene Ontology (GO) enrichment

GO enrichment analysis of genes with allele-specific expression was performed using Web Gestalt (https://www.webgestalt.org/). All expressed genes were used as the background universe. Over-representation analysis was performed selecting Gene Ontology as a Functional database in the website with default settings.

## Insulation scores calculation and TAD border calling

TAD calling was performed by calculating insulation scores in NPMI GAM contact matrices at 50 kb resolution using the insulation square method as previously described (Winick-Ng et al, 2021). The insulation score was computed with insulation square sizes ranging from 100 to 1000 kb for the unphased matrices and each haplotype. TAD borders were called using a 400 kb insulation square size and based on local minima of the insulation score with one genomic bin added on each side.

## Allele quantification with cryo-FISH

We obtained the source cryo-FISH data for the detection of 40 kb genomic regions containing the Hoxb1 or Hoxb13 genes performed in mESCs clone 46C, which reports for each nuclear slice, whether 1 or 2 copies of each locus are present (Barbieri et al, 2017). We counted the number of sections that contained both alleles and divided for the number of sections that contained one or two alleles. We performed this analysis for two channels: the green that corresponded to Hoxb1 locus and the red that corresponded to the Hoxb13 locus.

## Identification of compartments A and B

Compartments were calculated from 100 kb resolution GAM co-segregation matrices as previously described (Beagrie et al, 2017). In brief, each chromosome was represented as a matrix of observed interactions O(i,j) between locus i and locus j. We then calculated the expected interactions E(i,j) matrix, where each pair of genomic windows is the mean number of contacts with the same distance between i and j. A matrix of observed over expected values O/E(i,j) was produced by dividing O by E. A correlation matrix C(i,j) was calculated between column i and column j of the O/E matrix. PCA was performed for the first three components on matrix C. Loci with PC eigenvector values with the same sign that correlate best with GC content were called A compartments, whereas regions with the opposite sign were B compartments. Finally, the first PC was chosen for all chromosomes. Eigenvector values on the same chromosome in compartment A were normalized from 0 to 1, whereas values on the same chromosome in compartment B were normalized from −1 to 0.

## Identification of allele-specific contacts

Allele-specific contacts were identified using a previously developed pipeline for finding differential contacts between two contact maps (Beagrie et al, 2023; Winick-Ng et al, 2021) with some adjustments to adapt for the allelic setting. Following the removal of undersampled regions and setting a maximum contact distance of

50 Mb, each chromosomal contact matrix at 50 kb resolution from CAST and S129 NPMI was transformed into their z-scores equivalent, by adjusting for the mean and variance across all contact distances. Next, the difference between both alleles was computed by subtracting normalized S129 contacts from CAST contacts (delta z-score=CAST-S129). Finally, contacts with delta z-score below −1 or above 1, and NPMI intensities above 0.3 in either of the two maps were selected as S129-specific or CAST-specific contacts, respectively, to focus the subsequent analyses on the strongest contacts.

## Identification of strong allelic contacts

Strong allelic contacts represent the highest values on each chromosome of each allele. In contrast to allele-specific contacts which are specific to one haplotype, strong contacts are not informed by the alternative allele and, in consequence, strong CAST contacts can also be strong in the S129 allele, and vice versa. The strongest contacts in the CAST allele and S129 allele were extracted using an NPMI score >0.3 and a z-score >2.0 in the distance-normalized matrices from each haplotype, respectively (see Identification of differential contacts).

## Distance decay and derivatives calculation

Decay plots and momentum curves (Abdennur et al, 2024) of S129 and CAST contact maps were calculated using the mean contact intensity over distance displayed at logarithmic scale (log10). Momentum curves were obtained from the *ksmooth* R function using a Normal kernel with bandwidth of 0.3. The slope values in CAST or S129 contact decay are based on derivatives obtained from the difference between observed mean intensity scores at equidistant breakpoints, set at log10-scaled distance intervals of 0.1.

## ATAC-seq data mapping, processing, QC, and phasing

ATAC-seq reads were mapped, quality controlled, and split into their respective genomes using SNPsplit. Then, peaks were called with ChromA (https://github.com/marianogabitto/ChromA, Gabitto et al, 2020). D score was calculated for each peak, as a measure of their allelic imbalance in order to assign allele-specific peaks followed by a permutation test to assess their significance (Xu et al, 2017). Finally, a stringent filtering was applied to identify allele-specific peaks, requiring both biological replicates to have a D score between −0.3 and 0.3 (reads ratio score), a minimum of ten reads in the peak, and a *P* value < 0.01 in the permutation test, after FDR correction according to Benjamini and Hochberg.

## Motif calling in ATAC-seq peaks

First, *annotatePeaks.pl* script from the Homer tools suite was run in the CAST-specific, S129-specific, common or unphased ATAC peaks, to classify them depending on their genomic position. Then, for each type, the closest gene was identified, which is the most likely to be the target gene. Finally, for each of these groups, *findMotifsGenome.pl* was run to find the enriched motifs. Q value of ≤0.05 and *P* value of ≤0.001 was used as cutoff for enriched motifs.

## ChIP-seq data collection, QC, mapping, and processing

Chromatin immunoprecipitation experiments were performed as previously described (Brookes et al, 2012; Ferrai et al, 2017). Pol2-S5p was detected with mouse antibodies CTD4H8 clone (BioLegend, Cat# 904001); Pol2-S7p with rat antibodies 4E12 clone (Chapman et al, 2007; kindly provided by Dirk Eick); Polycomb mark H3K27me3 was detected with rabbit antibodies (Millipore, Cat# 07-449). ChIP-seq libraries were prepared from 10 ng of immunoprecipitated DNA using TruSeq ChIP Library Preparation Kit (Illumina, IP-202-1012) according to the manufacturer's instructions with minor modifications. Library size was assessed before high-throughput sequencing by Bioanalyzer (Agilent) using the High Sensitivity DNA analysis kit (Agilent, Cat# 5067-4626). ChIP-seq libraries were sequenced 75 bp single-end using Illumina NextSeq500/550 sequencer, according to the manufacturer's instructions.

## ChIP-seq peak calling and phasing

Raw ChIP-seq reads were mapped to the N-masked genome using default parameters of bowtie2 (version 2.3.4.3; Langmead and Salzberg, 2012). Genome-wide enriched regions for Pol2-S5p, Pol2-S7p, and H3K27me3 were identified with Bayesian Change-point Model (BCP) peak-finder (Xing et al, 2012, default settings). Genome-wide enriched regions for H3K27ac, H3K4me3, CTCF and Rad21 were identified with MACS2 peak-finder (Zhang et al, 2008; broad peaks, default settings). If two biological replicates were available (CTCF, H3K4me3, H3K27ac, Rad21), the peak calling was performed in each dataset separately and then peaks identified in both datasets were used for further analysis. Next, ChIP-seq reads were phased for all datasets using the SNPsplit package and the number of reads in each peak was computed with bedtools coverage (Fig. EV8B; Quinlan and Hall, 2010). To classify the peaks as allele-specific, the ratio between CAST and S129 allele-specific reads was computed for each peak. Peaks that have log2 fold change > 2 were selected as allele-specific. Peaks that had <10 SNP-containing reads were excluded from further analysis. Number of allele-specific reads for all CTCF, Rad21, H3K27ac, H3K4me3, H3K27me3, Pol2-S5p and Pol2-S7p peaks, as well as LFC values were provided in the permanent data repository (Irastorza-Azcarate et al, 2024).

## ASE upregulation upon AID-dependent acute depletion of RING1B protein in a Ring1a knockout ESC line

DeSeq2 differential expression analysis for AID-dependent PRC-acute depletion of RING1B protein in a Ring1a knockout ESC line were taken from GSE159399 (Dobrinić and Klose, 2021b), after 8 h of knockdown (Supplementary file GSE159399_RING1AKO.RING1BAID_spikenormalised_DESeq2_NucRNAseq_IAA_8h_v-s_UNT.txt.gz). The geneSymbol column was used to identify ASE genes. Genes excluded from the promoter state classification (overlapping genes and genes whose TSSs were in close proximity) were also excluded from this analysis.

## CTCF orientation calling

*annotatePeaks.pl* script was used from the *Homer* suite tools to call the orientation of CTCF motifs in CTCF peaks (http://homer.ucsd.edu/homer/ngs/annotation.html).

## Genome-wide feature co-occurrence

First, the genome is binned in 200 kb bins. Second, for each feature, the number of peaks is counted in each bin, creating a list. Finally, these lists are correlated and a Pearson correlation coefficient is calculated for each comparison.

## Differential DNA methylation

Since bisulfite treatment causes C to T transitions, certain SNP positions may not be used for allele-specific reads sorting since they might reflect either an allele-specific difference or a methylation state. To overcome this limitation, a modified N-masked genome was prepared using Bismark software package (Krueger and Andrews, 2011) and analyzed publicly available whole-genome bisulfite sequencing data (Li et al, 2019) with SNPsplit package in whole-genome bisulfite sequencing (WGBS) compatible mode (Krueger and Andrews, 2016). The reads were trimmed using the Trim Galore package using the default settings prior to mapping (Martin, 2011). The methylation calls for every analyzed C were extracted using bismark_methylation_extractor script.

For each allele, CpGs with a methylation percentage higher than 50 were taken for further analysis. Next, the ratio of methylated/unmethylated CpGs in the promoter (± 1000 bp from TSS) of genes longer than 2000 bp was calculated. The ratio in the CAST allele was subtracted to the ratio in the S129 allele, giving the differential percentage of possible methylated CpGs. Finally, differentially methylated promoters were those where this differential percentage exceeded the 5th percentile.

## Proteomics

ESC-ERT2 cells grown in DMEM media for SILAC were used as SILAC reference. Cells were cultured in DMEM media lacking L-lysine and L-arginine amino acids, supplemented with 15% knockout serum replacement (KOSR; Invitrogen, #10828), cytokine leukemia inhibitory factor (LIF, Merck, #ESG1107) and heavy amino acid isotopes (L-lysine, +8 Da; Cambridge Isotope Laboratories, #CNLM291H; L-arginine +10 Da; Cambridge Isotope Laboratories, #CNLM-539H; Bendall et al, 2008). Three biological replicates were collected. Cells were lysed in urea buffer (8 M urea, Tris 100 mM, pH 8.25) and sonicated a Bioruptor sonicator (Diagenode), using 10 cycles of sonication (30 s ON, 30 s OFF). After centrifugation to remove debris, protein concentration was measured by Bradford colorimetric assay and 50 µg protein extract were mixed with an equal amount of reference heavy sample. The disulfide bridges of proteins were then reduced in 2 mM DTT for 30 min at 25 °C and successively free cysteines alkylated in 11 mM iodoacetamide for 20 min at room temperature in the dark. LysC digestion was then performed by adding 2 µg of LysC (Wako) to the sample and incubating for 18 h, under gentle shaking at 30 °C. After LysC digestion, the samples were diluted 3 times with 50 mM ammonium bicarbonate, before addition of 7 µl of immobilized trypsin (Applied Biosystems) and incubation for 4 h under rotation at 30 °C. 18 µg of the resulting peptide mixtures were desalted on STAGE Tips (Rappsilber et al, 2002) and the eluates dried and reconstituted to 20 µl of 0.5% acetic acid in water.

## LC-MS/MS analysis

Five microliters of each sample were injected into the HPLC system (Eksigent) coupled to an Orbitrap Velos mass spectrometer (Thermo); each biological replicate was analyzed in two technical replicates. The chromatographic separation using a 240 min gradient ranging from 5% to 45% of solvent B (80% acetonitrile, 0.1% formic acid; solvent A = 5% acetonitrile, 0.1% formic acid). A 30 cm long capillary column (75 µm inner diameter) was packed with 1.8 µm C18 beads (Reprosil-AQ, Dr. Maisch). A tip was generated on one end of the capillary nanospray using a laser puller, allowing fritless packing. The nanospray source was operated with a spray voltage of 1.9 kV and an ion transfer tube temperature of 260 °C. Data were acquired in data-dependent mode, with one survey MS scan in the Orbitrap mass analyzer (30,000 resolution at 400 $m/z$) followed by up to 10 MS/MS scans in the Orbitrap analyzer (15,000 resolution at 400 $m/z$) on the most intense ions. Once selected for fragmentation, ions were excluded from further selection for 45 s, to increase new sequencing events.

## Proteomics data analysis

Raw data were analyzed using the MaxQuant proteomics pipeline v 2.1.3.0 and the built in the Andromeda search engine (Cox and Mann, 2008; Cox et al, 2011) with the Uniprot mouse protein database. Carbamidomethylation of cysteines was chosen as fixed modification, oxidation of methionine and acetylation of N-terminus were chosen as variable modifications. Two missed cleavage sites were allowed and peptide tolerance was set to 7 ppm. The search engine peptide assignments were filtered at 1% FDR at both the peptide and protein level. The 'match between runs' feature was enabled, 'second peptide' feature was enabled, while other parameters were left as default.

# Data availability

The datasets and computer code produced in this study are available in the following databases: GAM data: Gene Expression Omnibus GSE254717 and 4DN data portal 4DNESRQDNG61 (https://data.4dnucleome.org/); Total RNA-seq data: Gene Expression Omnibus GSE254675; ChIP-seq data: Gene Expression Omnibus GSE254710; Insulation score values of unphased, CAST and S129 alleles: Zenodo 14066696; Coordinates of phased and unphased peaks: Zenodo 14066696; Gene expression levels, epigenetic features and classification of gene transcripts: Zenodo 14066696; Transcription factor motif enrichment at CAST or S129 ATAC peaks at promoters, intergenic or genic regions: Zenodo 14066696; Mass spectrometry proteomics: ProteomeXchange Consortium via the PRIDE (Perez-Riverol et al, 2021) dataset PXD048969; Custom python scripts to generate the plots for the figures: GitHub (https://github.com/pombo-lab/Irastorza-Azcarate_Kukalev_Kempfer_2024); GAM-Phaser pipeline, including the full list of SNPs used for CAST and S129 phasing: GitHub (https://github.com/pombo-lab/GAM_phaser).

The source data of this paper are collected in the following database record: biostudies:S-SCDT-10_1038-S44320-025-00107-3.

# Peer review information

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

## Acknowledgements

The authors thank Lonnie Welch, Yingnan Zhang, Luca Fiorillo and Francesco Musella for exploratory data analysis, all laboratory members and the 4D Nucleome consortium for helpful discussions, Dirk Eick for the kind gift of the Pol2-S7p monoclonal antibodies (4E12 clone), and the Genomics Technology Platform and the Protein Production and Characterization Platform, both at the Max Delbrück Center for Molecular Medicine in the Helmholtz Association (MDC), Berlin. AP, BR, and MN acknowledge support from the National Institutes of Health Common Fund 4D Nucleome Program grants U54DK107977 and 1UM1HG011585-03. AP and RFS acknowledge support from the Helmholtz Association (Germany) and the Deutsche Forschungsgemeinschaft (DFG) Priority Program SPP2202 'Spatial Genome Architecture in Development and Disease', SPP2202 (Project number 422841138). AP and KNN acknowledge support from the European Commission under FP7-Marie Curie Action: Initial Training Network InteGeR, 'Integrative Gene Regulation' (PITN-GA-2007-214902). AP acknowledges support from the Deutsche Forschungsgemeinschaft (DFG; German Research Foundation) under Germany's Excellence Strategy - EXC-2049 - 390688087, and the Medical Research Council (MRC, UK; grant MC_U120061476). II-A was supported by a Long-Term Fellowship from the Federation of European Biochemical Societies (FEBS). RFS is a Professor at the Cancer Research Center Cologne Essen (CCCE) funded by the Ministry of Culture and Science of the State of North Rhine-Westphalia. RFS acknowledges the German Ministry for Education and Research as BIFOLD—Berlin Institute for the Foundations of Learning and Data (ref. 01IS18025A and ref. 01IS18037A). MN acknowledges support from NextGeneration EU M4C2 CN00000041 CUP E63C22000940007, MUR PRIN 2022 CUP E53D23001810006, MUR PRIN PNRR 2022 CUP E53D23018360001 and computer resources from INFN, CINECA, ENEA CRESCO/ENEAGRID and Ibisco at the University of Naples. KNN acknowledges support from Medical Research Council (UK) Centenary grant. AGF and SS acknowledge support from MRC grants MC_U120027516 and MC_UP_1605/12.

## Author contributions

**Ibai Irastorza-Azcarate**: Conceptualization; Data curation; Software; Formal analysis; Supervision; Funding acquisition; Validation; Investigation; Visualization; Methodology; Writing—original draft; Project administration; Writing—review and editing. **Alexander Kukalev**: Conceptualization; Data curation; Software; Formal analysis; Supervision; Validation; Investigation; Visualization; Methodology; Project administration; Writing—review and editing. **Rieke Kempfer**: Conceptualization; Data curation; Software; Formal analysis; Validation; Investigation; Visualization; Writing—original draft; Project administration; Writing—review and editing. **Christoph J Thieme**: Data curation; Software; Formal analysis; Validation; Investigation; Visualization; Methodology; Writing—review and editing. **Guido Mastrobuoni**: Data curation; Software; Formal analysis; Validation; Investigation; Writing—review and editing. **Julia Markowski**: Data curation; Software; Formal analysis; Validation; Methodology; Writing—review and editing. **Gesa Loof**: Investigation.
**Thomas M Sparks**: Data curation; Software; Formal analysis; Validation; Methodology. **Emily Brookes**: Investigation; Methodology; Writing—review and editing. **Kedar Nath Natarajan**: Investigation; Methodology. **Stephan Sauer**: Resources. **Amanda G Fisher**: Resources. **Mario Nicodemi**: Conceptualization; Funding acquisition. **Bing Ren**: Conceptualization; Resources; Funding acquisition; Writing—review and editing. **Roland F Schwarz**: Supervision; Funding acquisition; Methodology. **Stefan Kempa**: Supervision; Funding acquisition; Methodology. **Ana Pombo**: Conceptualization; Supervision; Funding acquisition; Validation; Investigation; Methodology; Writing—original draft; Project administration; Writing—review and editing.

Source data underlying figure panels in this paper may have individual authorship assigned. Where available, figure panel/source data authorship is listed in the following database record: biostudies:S-SCDT-10_1038-S44320-025-00107-3.

## Funding

## Disclosure and competing interests statement

BR owns equity in Arima Genomics Inc. and Epigenome Technologies, Inc. AP and MN hold a patent on 'Genome Architecture Mapping': Pombo A, Edwards PAW, Nicodemi M, Scialdone A, Beagrie RA. Patent PCT/EP2015/079413 (2015). AP is a member of the Advisory Editorial Board of Molecular Systems Biology. This has no bearing on the editorial consideration of this article for publication. The remaining authors declare no competing interests.

# Expanded View Figures

**Figure EV1.  Strategy, methodology and evaluation of GAM-Phasing pipeline for allele-specific contact maps.**

(**A**) Schematic overview of GAM data collection, quality control steps, and merging of replicates R1 and R2. (**B**) GAM-phaser pipeline. (**C**) Percentage of reads that were phased to each allele. Conflicting reads are reads containing SNPs from both alleles. (**D**) Percentage of phased positive windows in the entire segregation table for all F123 3NPs passed quality controls GAM samples. (**E**) Phasing efficiency between GAM and Hi-C. GAM efficiency is measured as phased windows divided by the total number of called windows, while Hi-C efficiency is calculated dividing phased ligation events to unique ligation events; reported phasing efficiency was obtained from (Giorgetti et al, 2016). Below, schematic of a phaseable Hi-C ligation event. (**F**) Number of informative contact entries in the phased F123 GAM dataset in comparison with phased Hi-C data collected for human GM12878 B-lymphoblastoid cells (Rao et al, 2014b), at all intrachromosomal distances and for distances up to 10 Mb. (**G**) Number of nuclear sections that are positive for the presence of two or one *Hoxb1* or *Hoxb13* locus detected by cryo-FISH using 40 kb fosmid probes ($n = 341$ *Hoxb1* loci, $n = 362$ *Hoxb13* loci, $n = 2584$ nuclear sections imaged; data source from Barbieri et al, 2017). (**H**) Percentage of locus pairs detected at least once and Kendall's τ coefficient values for different resolutions and different distances. These metrics were used to decide on optimal resolutions of the maps.

▶

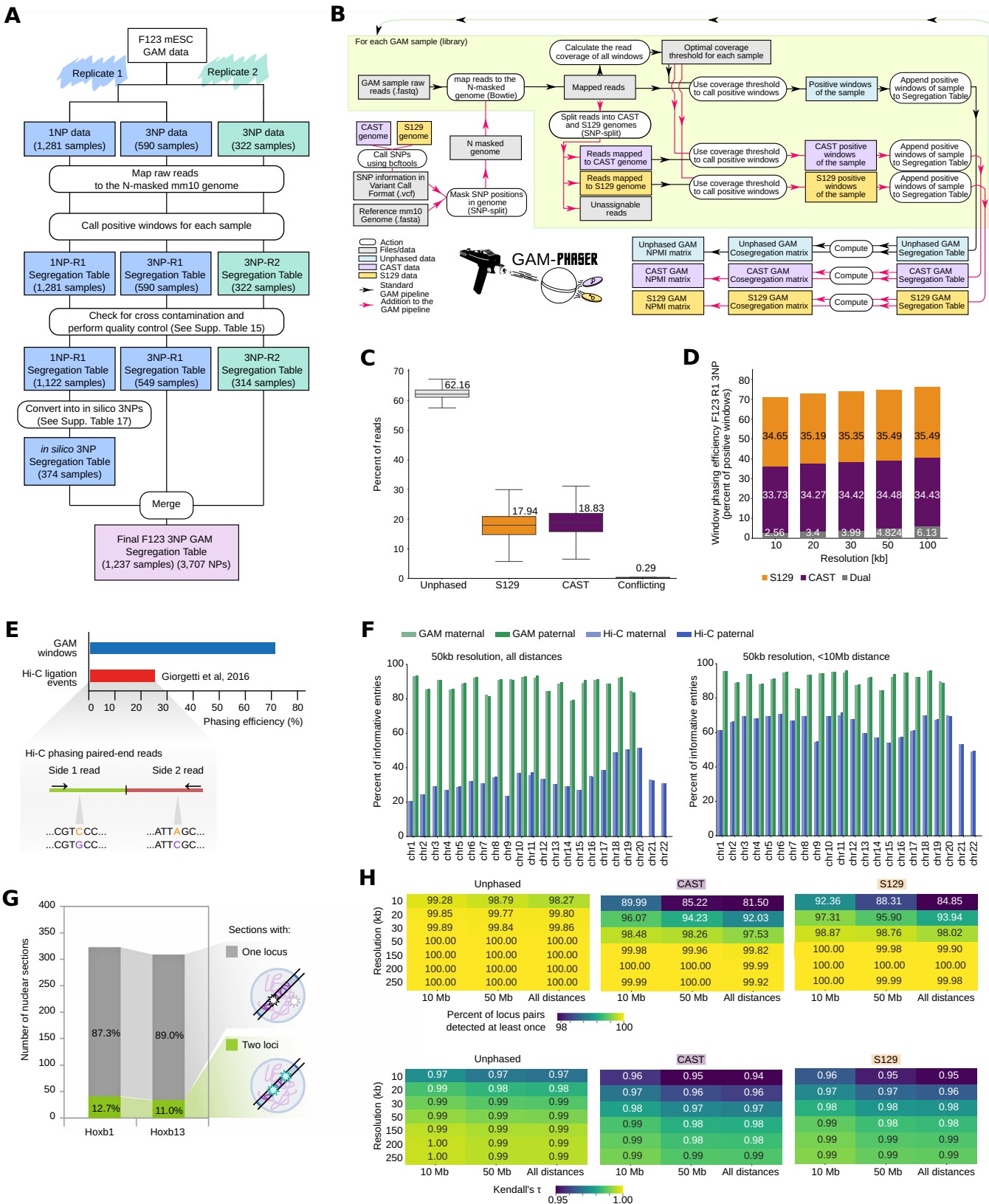

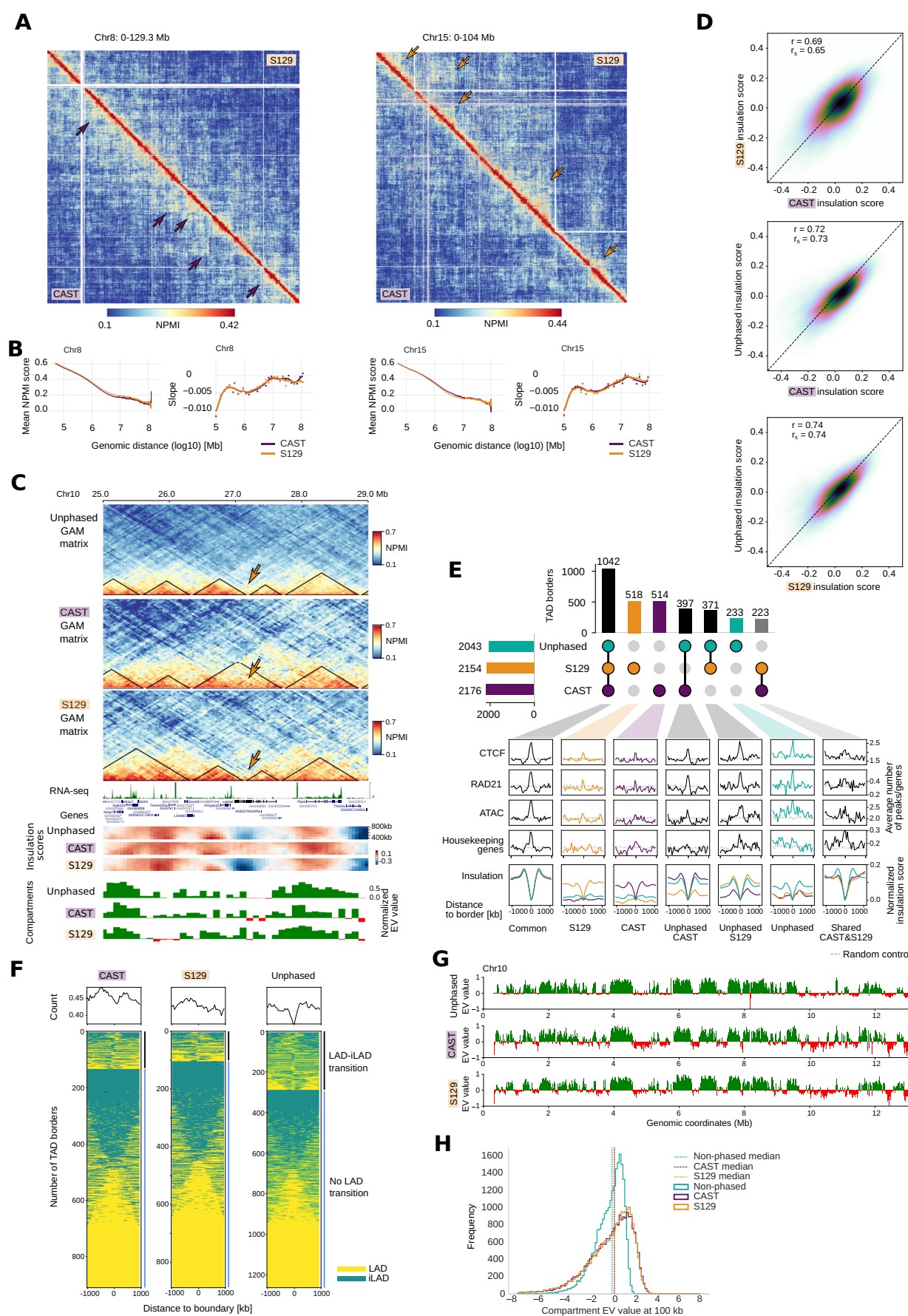

**Figure EV2.  Comparison of insulation, TADs and compartments between S129 and CAST haplotypes.**

(**A**) GAM matrices of chromosomes 8 and 15 showing both alleles at 50 kb resolution. Colored arrows show structural differences between alleles. (**B**) Distance decay curves and momentum curves for contact intensities across all distances in CAST and S129 chromosomes 8 and 15. (**C**) 4 Mb region in chromosome 10 showing an allele-specific TAD border in the S129 allele. Below, RNA-seq track, insulation scores and compartment tracks for all maps. (**D**) Pearson correlation coefficient (r) between combinations of CAST, S129 and unphased insulation scores at 400 kb. (**E**) Upset plot of TAD border combinations between CAST, S129 and the unphased maps. Below, aggregate plots for CTCF, Rad21 and ATAC-seq peaks and housekeeping genes, centered at the TSS ( ± 1 kb). Normalized Insulation score is also shown for each group. (**F**) Overlap of LADs and iLADs with ±1,000 kb around CAST, S129 and common TAD borders, computed from 100 kb resolution GAM matrices to match LAD annotations. Each heatmap is clustered depending on whether the border overlaps with a LAD/iLAD transition or not. (**G**) Compartment tracks for CAST, S129 and the unphased maps for chromosome 10. (**H**) Compartment eigenvector values distribution for CAST, S129 and the unphased datasets. Discontinuous lines show the median for each dataset.

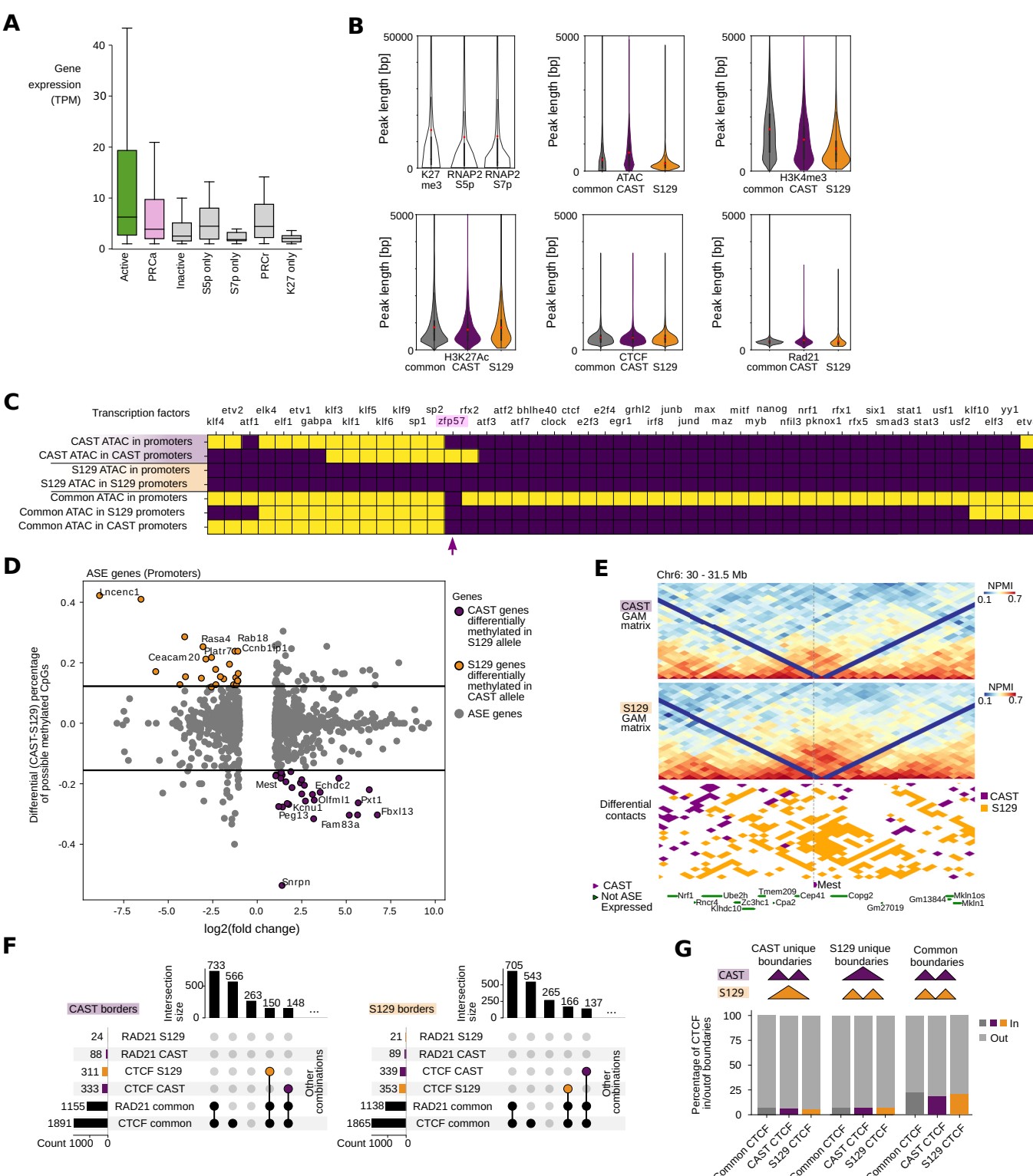

◀ **Figure EV3. Association of allele-specific epigenetic marks and transcription factors with promoters of ASE genes and 3D genome organization.**

(A) Gene expression of the different groups in Fig. 2E. (B) Distribution of Pol2-S5p, Pol2-S7p and H3K27me3 peaks, and phased and unphased ATAC-seq, H3K4me3, CTCF, H3K27ac and RAD21 peak sizes. Red dots indicate the average size for each dataset. (C) Heatmap showing the enriched presence (cutoffs Q value ≤ 0.05 and *P* value of ≤0.001) of different transcription factors that overlap with the peaks of different ATAC-seq groups. ZFP57 is the only transcription factor enriched for an allele-specific group. (D) ASE gene promoters regarding their differential percentage of methylated CpGs. Colored are those genes with a significant amount of methylated CpGs in their promoter (top and bottom 5%) in the allele they are not expressed. (E) CAST and S129 GAM matrices for the Mest locus (Chr6: 30–31.5 Mb). Below, differential contacts and two tracks showing CAST genes and expressed genes. (F) Most borders contain common CTCF and RAD21 or only CTCF and each allele has a similar number of CTCF specific to either of the alleles in its borders. (G) Percentage of CTCF peaks that are inside or outside borders.

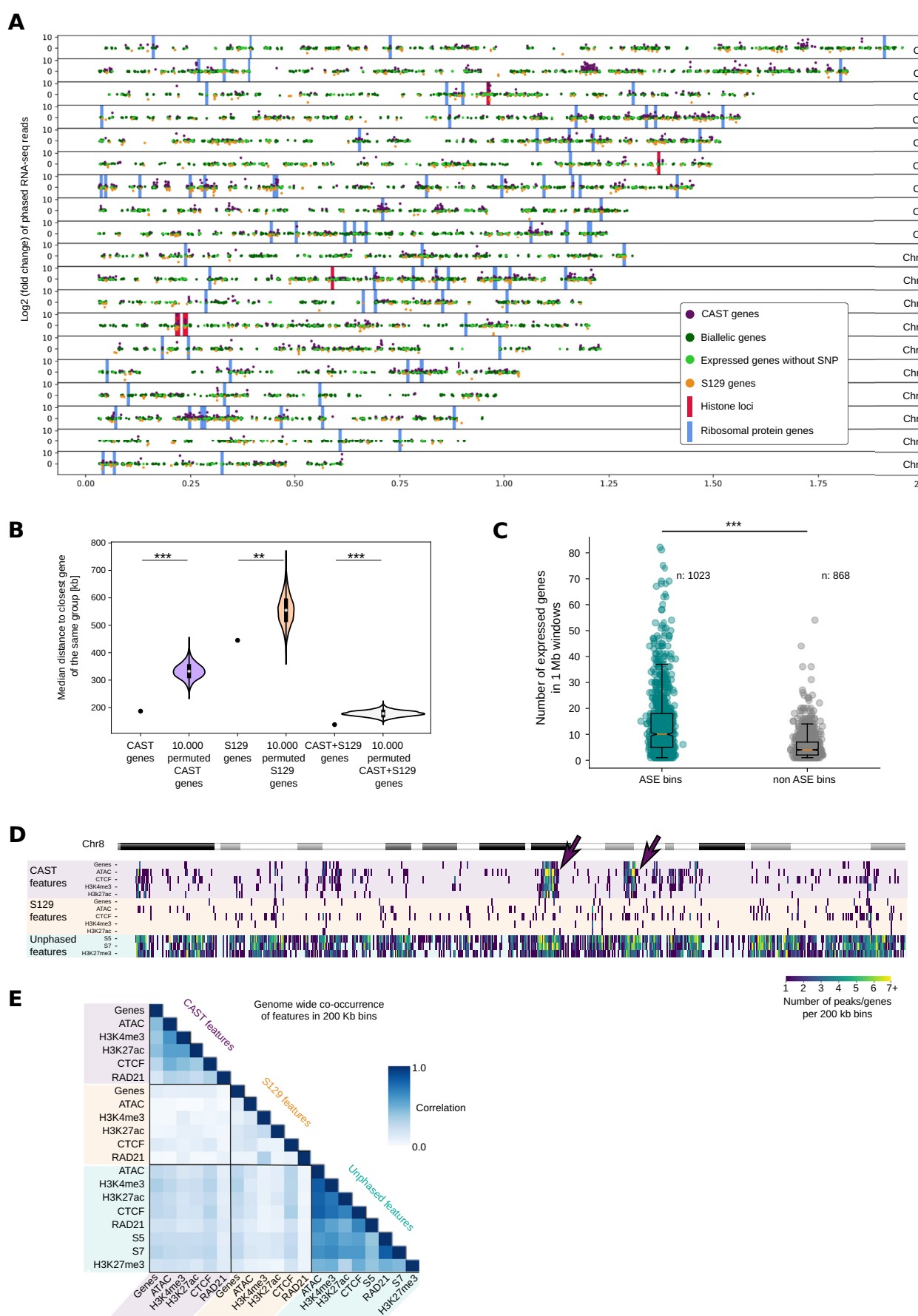

**Figure EV4. Co-presence of ASE genes and chromatin features in the linear genome.**

(**A**) Genomic position of all expressed genes: CAST genes, S129 genes, biallelic genes and genes without SNPs. Red and Blue bars indicate the position of Histone protein genes and Ribosomal protein genes. (**B**) Each of the 3 dots indicate the average distance of all genes of each type (CAST, S129 and CAST or S129) to the closest gene of that type. The violin plot shows the distribution of these averages if we permute the position of the genes 10,000 times. The permutation is carried out by randomly selecting the same number of CAST, S129 or CAST + S129 genes from all expressed genes. *P* values = 0.0001, 0.0145, 0.0001 for CAST, S129 and CAST + S129. (**C**) 1 Mb windows containing at least 1 ASE gene tend to contain more expressed genes than 1 Mb windows that do not contain ASE genes. T test: *P* value = $2.7 \times 10^{-71}$. Number of ASE windows, 1023. Number of non-ASE windows, 868. (**D**) Genomic location of allele-specific features (genes, ATAC-seq, CTCF, H3K4me3 and H3K27ac peaks) and unphased features (Pol2-S5p, Pol2-S7p and H3K27me3 peaks) and their density in bins of 200 kb. Arrows indicate two regions with an enrichment of CAST-specific features. (**E**) Genome-wide Pearson correlation (r) of the co-occurrence of the features in (**D**). CAST features correlate well between each other while S129 features do not.

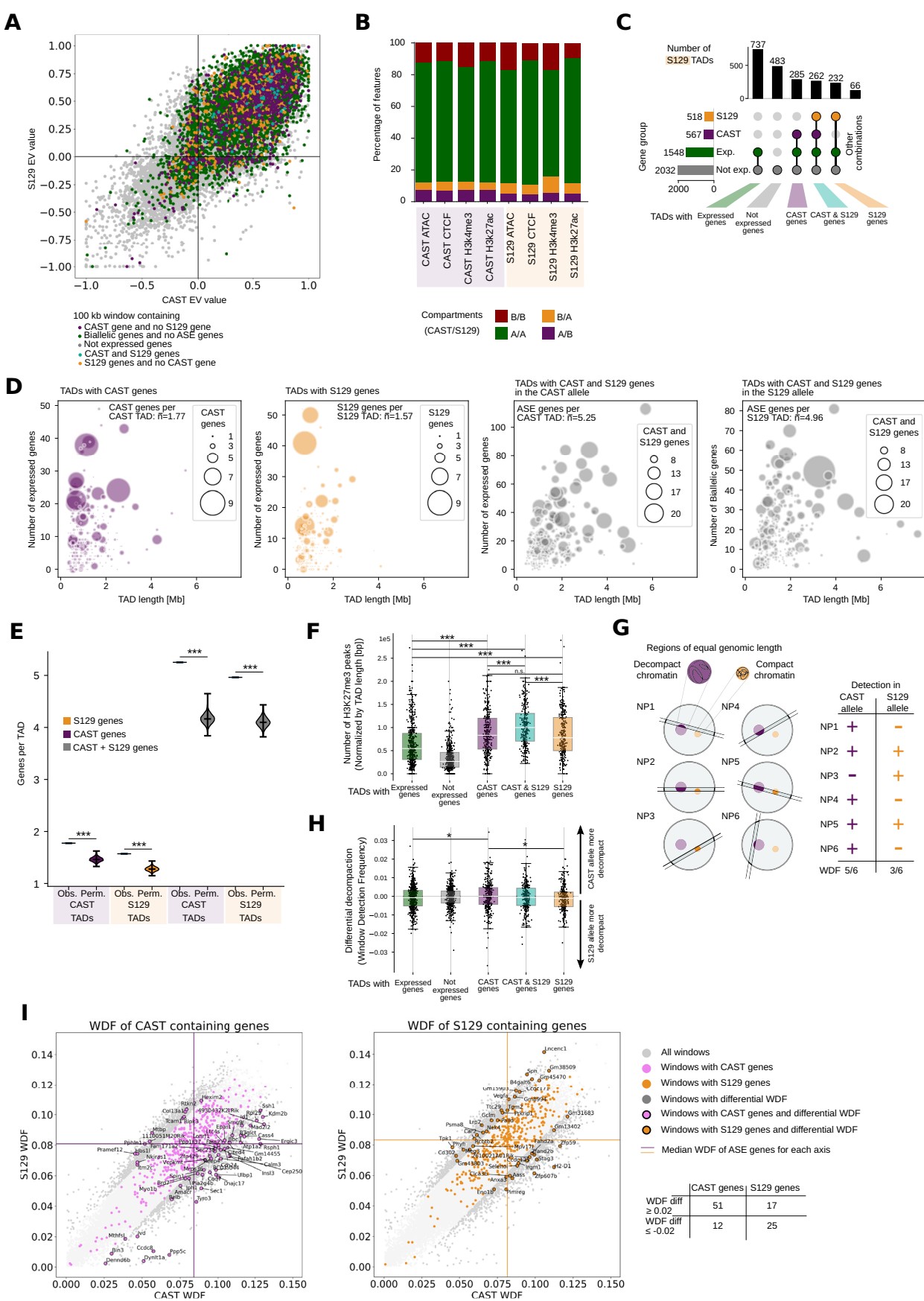

**Figure EV5. Comparative analysis of CAST and S129 alleles in genome compartmentalization and chromatin accessibility.**

(A) Normalized eigenvector (EV) values for the CAST and S129 allele for each 100 kb bin. Color coded are bins containing only not expressed genes, bins containing biallelic genes but not ASE genes, bins containing at least one CAST gene but not S129, bins containing at least one S129 gene but not CAST genes and bins containing at least one CAST gene and one S129 gene. (B) Percentage of ATAC-seq, CTCF, H3K4me3 or H3K27ac peaks in each compartment combination (A and A, B and B, A and B or B and A for CAST and S129 alleles, respectively). CAST-specific features show a tendency to overlap more in A/B (A specific compartment in the CAST allele), S129-specific features tend to overlap more in B/A (A specific compartment in the S129 allele.). (C) UpSet plots showing for the S129 allele, groups of TADs containing different sets of types of genes and their number. (D) Relation between the TAD length, the number of expressed genes in a TAD, and number of genes specific to that allele (dot size) for TADs in CAST and S129. Purple refers to TADs containing CAST genes, orange to TADs containing S129 genes, and gray to TADs containing CAST and S129 genes (for both CAST allele and S129 allele, respectively). (E) Violin plots showing the number of genes per TAD (observed, Obs.) compared to circular permutations of gene positions in the genome (permuted, Perm.). 10,000 permutations were done for each of the 4 examples in (D) and are compared to the number of genes per TAD in the original data (called *real*). All P values are ≤0.0001. Numbers are 911, 889, 1265 and 1265, respectively. (F) Related to (C), number of H3K27me3 peaks normalized by TAD length (two-sided $t$ test: *$P \leq 0.05$, **$P \leq 0.01$, ***$P \leq 0.001$; P values from top to bottom in S129 TADs: $1.9 \times 10^{-14}$, $3.2 \times 10^{-30}$, $1.1 \times 10^{-10}$, 0.00011, $1.8 \times 10^{-5}$, n.s: 0.5092. Number of TADs with: expressed genes, 737; not expressed genes, 483; CAST genes, 285; S129 genes, 232; and with CAST and S129 genes, 262). (G) Loci with the same genomic length can have different volumes due to varying compaction. Decompacted loci with larger volumes are captured more frequently in the collection of GAM cryosections than more compacted chromatin. Window Detection Frequency (WDF) is a GAM-intrinsic measure of relative chromatin compaction, defined by the number of locus detection events in the collection of GAM nuclear slices (Beagrie et al, 2017). From phased window segregation tables, the WDF can be calculated separately for CAST and S129, as a measure of relative compaction between all loci in each haplotype. (H) Related to (C), for each group, the differential (CAST-S129) window detection frequency is represented. Positive values indicate decompaction in the CAST allele, while negative values indicate decompaction in the S129 allele (two-sided $t$ test: *$P \leq 0.05$, **$P \leq 0.01$, ***$P \leq 0.001$; P values from top to bottom for S129 TADs: 0.031, 0.025). Number of TADs analyzed are the same as in (F). (I) Window detection frequency (WDF) values in the CAST and S129 allele for each bin containing genes. Fisher's exact test ($P = 5.3 \times 10^{-5}$) shows the significant tendency of windows with high WDF in the CAST allele containing CAST genes and windows with high WDF in the S129 allele containing S129 genes compared to windows with lower WDF. Numbers for CAST and S129 genes with differential WDF ≥ 0.02 are 51 and 17, respectively. For CAST and S129 genes with differential WDF ≤ −0.02 are 12 and 25, respectively.

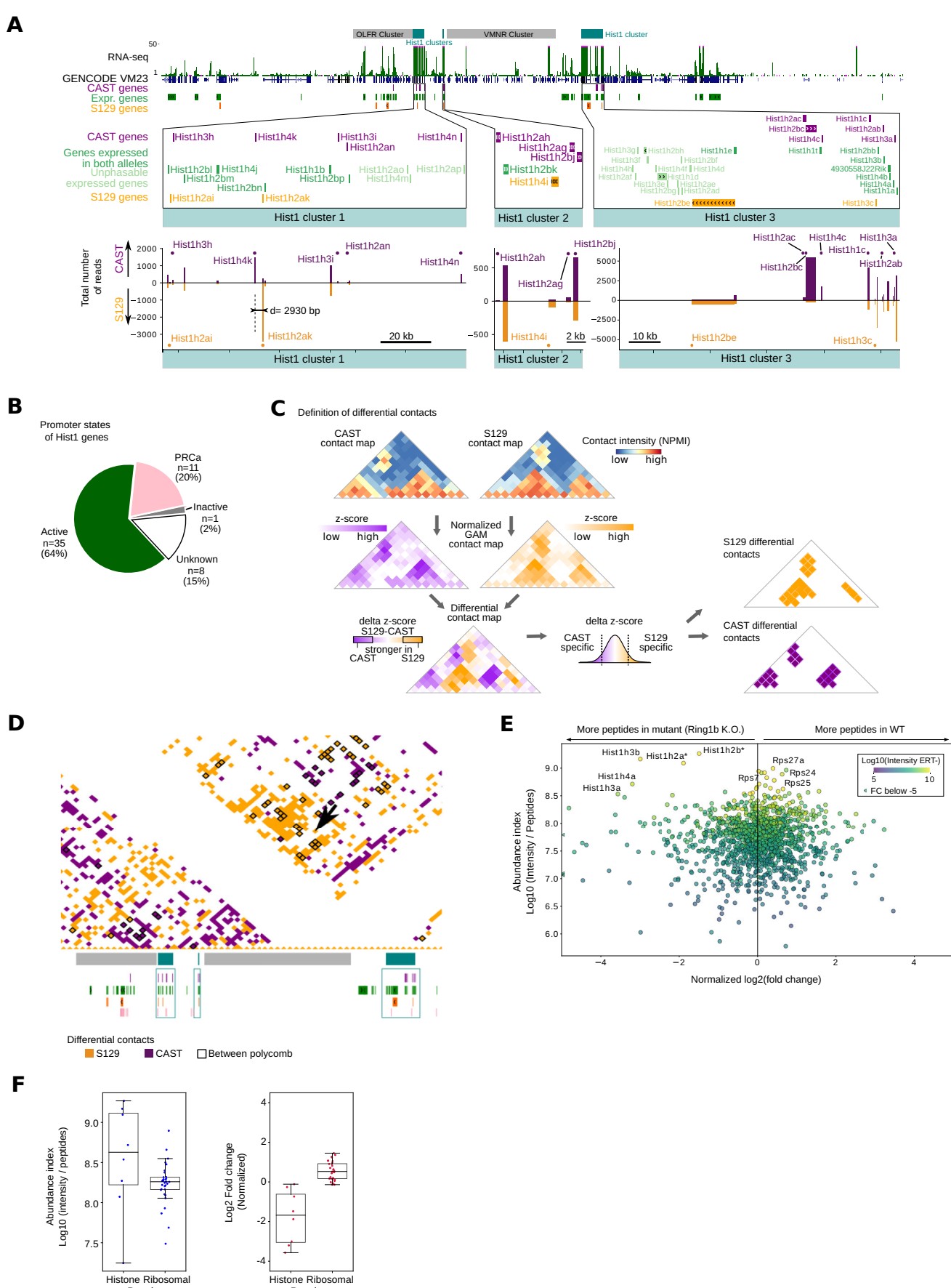

**Figure EV6. Differential gene regulation at the Hist1 gene cluster.**

(A) Hist1 locus region from Fig. 4A with greater detail. Histone genes are depicted. Number of reads phased for the CAST or S129 allele are shown for all genes in the Hist1 clusters. (B) Proportion of Hist1 genes according to promoter state. (C) Schematic showing the pipeline used to compare CAST and S129 contact intensities and to extract CAST-specific and S129-specific contacts. (D) Allele-specific contacts at the Hist1 locus as shown in Fig. 4A. Black squares show the contacts where H3K27me3 peaks are present in both windows of the contact. (E) SILAC experiments were performed in the ESC-ERT2 cells in the presence and absence of tamoxifen to induce knockout of *Ring1b*, in three biological replicates. *Ring1b* knockout results in upregulation of histone proteins. Abundance was estimated by the ratio of intensity and number of peptides. Normalized log2 fold change was calculated applying the z-score normalization to the log2 of heavy/light (H/L) ratio of the untreated experiment divided by the H/L ratio of the conditional knockout. (F) Boxplots showing the abundance index and the log2 fold change for detected histone proteins and ribosomal proteins. Numbers of data points are 8, 32, 8, 32, respectively from left to right.

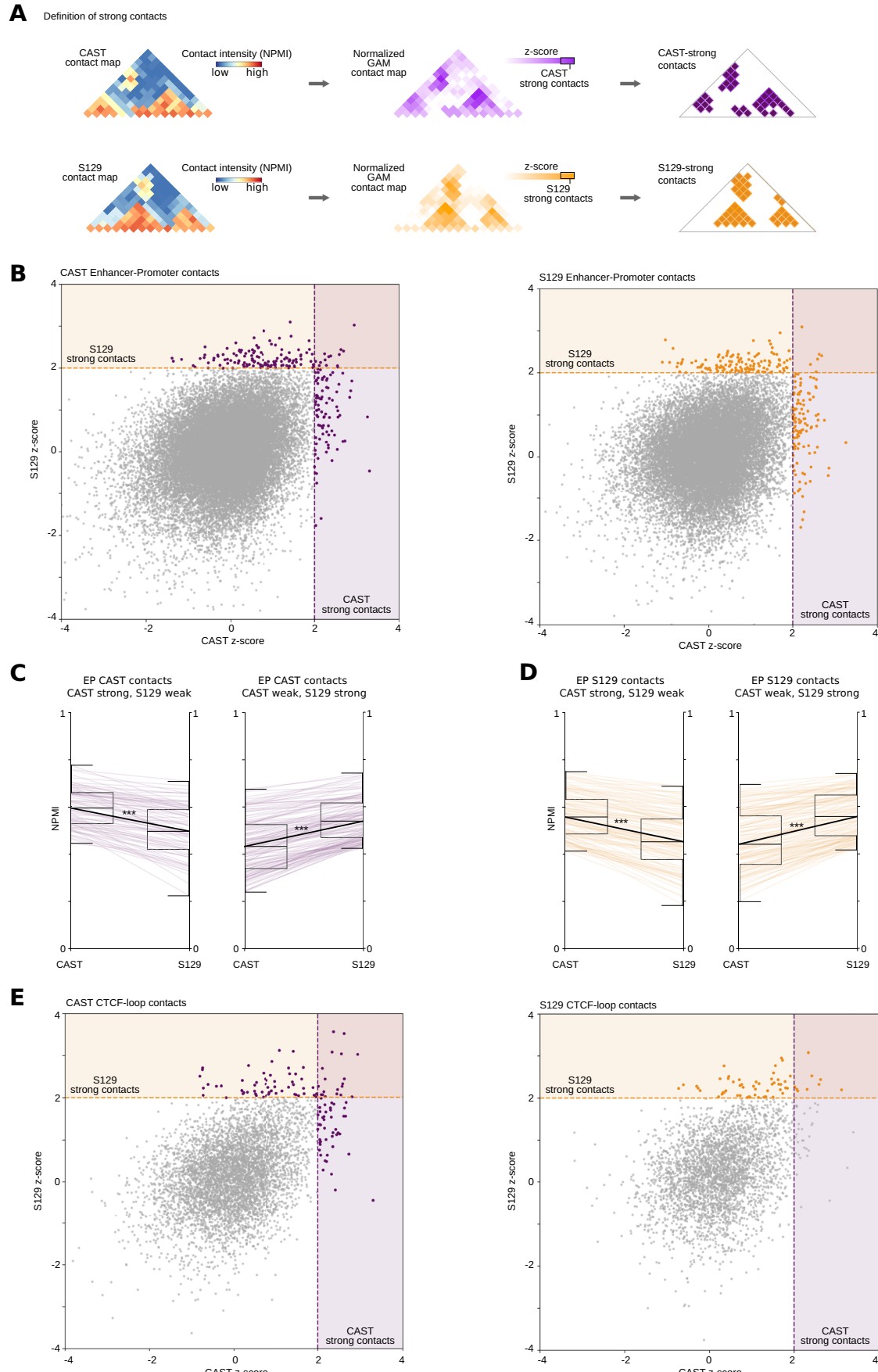

**Figure EV7.   Strong allelic contacts bridge promoters, enhancers, and CTCF sites.**

(A) Schematic showing the strategy to identify strong allelic contacts. (B) All possible contacts involving conditions for CAST-specific enhancer–promoter contacts and S129-specific enhancer–promoter contacts. Lines mark cutoffs for strong and allele-specific contacts in each haplotype. (C) Differences in contact intensities observed in the CAST and S129 haplotypes for allele-specific enhancer–promoter (E–P) elements associated with CAST contacts that were found to be strong in CAST but weak in S129 (on the left), or strong in S129 but weak in CAST (on the right). Two sample *t* test: *P* values are 4.17e-10 and 8.78e-17, respectively. Numbers are 84 and 130 respectively. (E) All possible contacts involving conditions for CAST-specific CTCF loops and S129-specific CTCF loops. Lines mark cutoffs for strong and allele-specific contacts in each haplotype. (D) Differences in contact intensities observed in the CAST and S129 haplotypes for allele-specific enhancer–promoter (E–P) elements associated with S129 contacts that were found to be strong in CAST but weak in S129 (on the left), or strong in S129 but weak in CAST (on the right). Two sample t test: p values are 3.92e-12 and 4.42e-14, respectively. Numbers are 112 and 129, respectively.

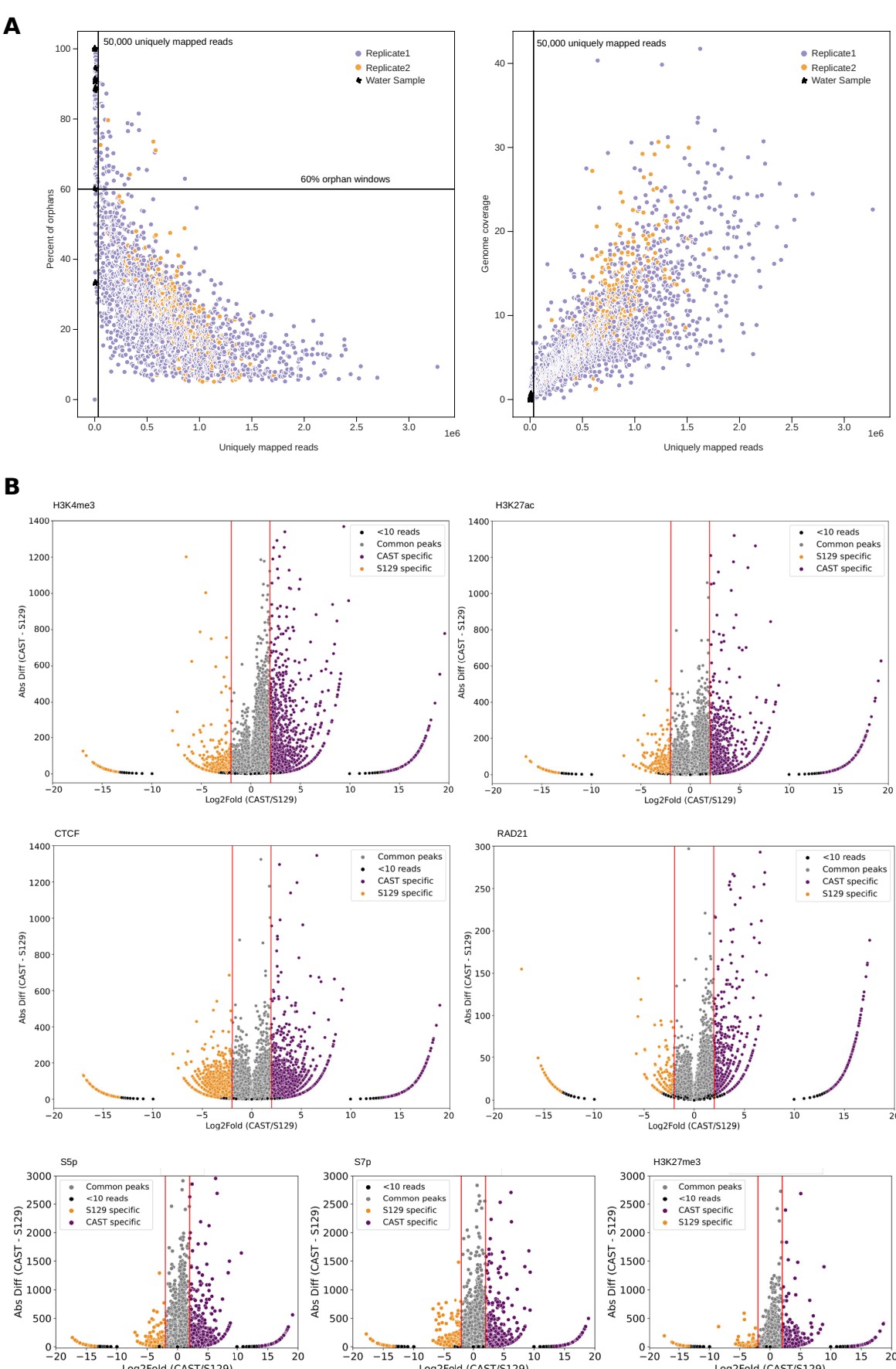

**Figure EV8. Assessment of read-count thresholds in phasing.**

(A) Distribution of percentage of orphan windows, uniquely mapped reads and genome coverage in each GAM sample. Replicate 1, replicate 2 and water (0NP) samples are shown. Thresholds used to remove potentially low from high quality GAM samples are shown in vertical and horizontal black lines. (B) Distribution of phased H3K4me3, H3K27ac, CTCF, RAD21, Pol2-S5p, Pol2-S7p, and H3K27me3 peaks, showing their absolute difference in phased reads (CAST-S129) and their log2 fold change.

