## [Peer Review File · Molecular Systems Biology]

Extensive folding variability between homologous chromosomes in mammalian cells

Ibai Irastorza-Azcarate, Alexander Kukalev, Rieke Kempfer, Christoph Thieme, Guido Mastrobuoni, Julia Markowski, Gesa Loof, Thomas Sparks, Emily Brookes, Kedar Natarajan, Stephan Sauer, Amanda Fisher, Mario Nicodemi, Bing Ren, Roland Schwarz, Stefan Kempa, and Ana Pombo

Corresponding author(s): Ana Pombo (ana.pombo@mdc-berlin.de) , Ibai Irastorza-Azcarate (Ibai.irastorzaazcarate@mdc-berlin.de)

Review Timeline:

Submission Date:	5th May 24
Editorial Decision:	31st May 24
Revision Received:	7th Dec 24
Editorial Decision:	15th Jan 25
Revision Received:	31st Mar 25
Accepted:	10th Apr 25

Editors: Maria Polychronidou and Poonam Bheda

Transaction Report:

31st May 2024

Manuscript Number: MSB-2024-12394

Title: Extensive folding variability between homologous chromosomes in mammalian cells

Dear Ana,

Thank you again for submitting your work to Molecular Systems Biology. We have now heard back from two of the three reviewers who agreed to evaluate your study. In the interest of time, and since the recommendations of the two reviewers are similar, we have decided to proceed with making a decision based on the two available reports. As you will see below the reviewers acknowledge that the presented findings are a relevant contribution to the field. However, they raise several concerns, which we would invite you to address in a revision.

I think that the reviewers' recommendations are rather clear and I therefore see no need to repeat the comments listed below. All issues raised by the reviewers would need to be satisfactorily addressed. As you may already know, our editorial policy allows in principle a single round of major revision. It is therefore essential to provide responses to the reviewers' comments that are as complete as possible. If you have any questions of if would like to discuss your revision plan, please feel free to get in touch.

On a more editorial level, we would ask you to address the following points:

- Please provide a .doc version of the manuscript text (including legends for main Figures and EV Figures) and individual production quality figure files for the main Figures and EV Figures (one file per figure).
- Please include 5 keywords.
- We have replaced Supplementary Information by the Expanded View (EV format). In this case (unless the number of EV figures becomes very large during revision), all additional figures can be provided as EV Figures. Please provide one file per EV Figure. Their legends should be included in the manuscript text. For detailed instructions regarding expanded view please refer to our Author Guidelines: .
- Tables EV1-EV12 should be provided as EV Datasets. Please provide one file per EV Dataset. In each file, a description of the table/dataset should be provided in a separate sheet.
- Please provide a "standfirst text" summarizing the study in one or two sentences (approximately 250 characters), three to four "bullet points" highlighting the main findings and a "synopsis image" (exactly 550px width and max 400px height, jpeg or png format) to highlight the paper on our homepage.
- All Materials and Methods need to be described in the main text. We would encourage you to use 'Structured Methods', our new Materials and Methods format. According to this format, the Material and Methods section should include a Reagents and Tools Table (listing key reagents, experimental models, software and relevant equipment and including their sources and relevant identifiers) followed by a Methods and Protocols section in which we encourage the authors to describe their methods using a step-by-step protocol format with bullet points, to facilitate the adoption of the methodologies across labs. More information on how to adhere to this format as well as downloadable templates (.doc or .xls) for the Reagents and Tools Table can be found in our author guidelines: . An example of a Method paper with Structured Methods can be found here:
- Please make sure that all information related to the Data Availability is included in the manuscript text. (Currently some information is in the cover letter, but I did forward it to the reviewers.)
- Please include a Data availability section describing how the data and code have been made available. This section needs to be formatted according to the example below:
The datasets and computer code produced in this study are available in the following databases:
 - Chip-Seq data: Gene Expression Omnibus GSE46748 (<https://www.ncbi.nlm.nih.gov/geo/query/acc.cgi?acc=GSE46748>)
 - Modeling computer scripts: GitHub (<https://github.com/SysBioChalmers/GECKO/releases/tag/v1.0>)
 - [data type]: [full name of the resource] [accession number/identifier] ([doi or URL or identifiers.org/DATABASE:ACCESSION])
- For data quantification: please specify the name of the statistical test used to generate error bars and P values, the number (n) of independent experiments (specify technical or biological replicates) underlying each data point and the test used to calculate p-values in each figure legend. The figure legends should contain a basic description of n, P and the test applied. Graphs must include a description of the bars and the error bars (s.d., s.e.m.).
- Please include a "Disclosure & Competing Interests Statement" in the main text.

- When you resubmit your manuscript, please download our CHECKLIST (<https://bit.ly/EMBOPressAuthorChecklist>) and include the completed form in your submission.

Please note that the Author Checklist will be published alongside the paper as part of the transparent process (<https://www.embopress.org/page/journal/17444292/authorguide#transparentprocess>).

If you feel you can satisfactorily deal with these points and those listed by the referees, you may wish to submit a revised version of your manuscript. Please attach a covering letter giving details of the way in which you have handled each of the points raised by the referees. A revised manuscript will be once again subject to review and you probably understand that we can give you no guarantee at this stage that the eventual outcome will be favorable.

Kind regards,

Maria

Maria Polychronidou, PhD
Senior Editor
Molecular Systems Biology

We realize that it is difficult to revise to a specific deadline. In the interest of protecting the conceptual advance provided by the work, we recommend a revision within 3 months (29th Aug 2024). Please discuss the revision progress ahead of this time with the editor if you require more time to complete the revisions. Use the link below to submit your revision:

IMPORTANT:

See also figure legend guidelines: <https://www.embopress.org/page/journal/17444292/authorguide#figureformat>

- Please note that corresponding authors are required to supply an ORCID ID for their name upon submission of a revised manuscript (EMBO Press signed a joint statement to encourage ORCID adoption).

(<https://www.embopress.org/page/journal/17444292/authorguide#editorialprocess>)

Currently, our records indicate that the ORCID for your account is 0000-0002-7493-6288.

Link Not Available

***** PLEASE NOTE ***** As part of the EMBO Press transparent editorial process initiative (see our Editorial at <https://dx.doi.org/10.1038/msb.2010.72>), Molecular Systems Biology publishes online a Review Process File with each accepted manuscripts. This file will be published in conjunction with your paper and will include the anonymous referee reports, your point-by-point response and all pertinent correspondence relating to the manuscript. If you do NOT want this File to be published, please inform the editorial office at msb@embo.org within 14 days upon receipt of the present letter.

Reviewer #1:

In this manuscript, Irastorza-Azcarate and colleagues leverage GAM applied to mESCs from a hybrid mouse strain with a high load of haplotype-specific SNPs in order to examine the allele-specific architecture of the genome. Although we have long been aware of such differences existing (the authors cite a number of examples in their Introduction), the true extent of this allele-specific variance has remained unexplored. Leveraging GAM, the long contiguous genomic coverage of which simplifies

phasing, together with a new computational approach in 'GAM-phaser', the authors reveal a truly pervasive variance in 3D chromatin folding between alleles (GAM appears to produce phase contacts in >3-fold more of the genome compared to Hi-C). I find this significant, as most of us in the field have been conditioned to look at average maps from both alleles in which such differences are mostly hidden. Therefore, the advance presented here can be of importance for both genome regulation (e.g., the fact that most TAD borders appear to be haplotype-specific) and genome engineering studies in the future. Overall, the manuscript text is detailed and clear, the analysis is very well conducted and conclusions are well controlled (e.g., Ring1b- and Ezh2-KO data). The two key findings (in my opinion), that most active chromatin features are biallelic and that H3K27me3 clusters produce most allelic regulation, points to 3D genome folding having a pivotal role in gene regulation. I only have few remarks that I hope the authors find useful:

- The authors state that "[c]ontact distance decay and momentum curves showed similar frequency of contacts between haplotypes within < 5Mb of genomic distance, but became visibly distinct at long-range distances with different haplotype preferences depending on the chromosome", but I cannot find a follow-up to this that could provide an explanation. Could they elaborate? How is this explained? Has this got something to do also with the resolution of this GAM dataset?
- Is my interpretation that most allelic regulation comes via H3K27me3-mediated repression correct? If so, since the authors classify about 1/3 of ASEs as H3K27me3+S5p+S7p+, how would the rest be regulated?
- Looking at Fig. 5 and the attempt at dissecting E-P contacts, I have the following questions: What is the effective resolution of phased GAM matrices (as I might have missed this)? If one compares the differences marked by the authors in Fig. 5d and 5e, I would argue that contact changes in panel e are obvious, but not much so in panel d. Is this the magnitude of changes to be expected? Perhaps the authors can produce a boxplot summarizing the strength changes in these AS contacts? Then, they would be able to also provide some statistics on this (which also seem to be missing from the comparisons in panel 5c)?
- Also in respect to Fig. 5 and this whole section, it would be useful to readers if the authors explained how "general decompaction" of the loci is measured and exemplified in GAM matrices.
- The last part of the Results on CTCF loops involved in ASE, does not merit a segment of its own as it does not conclusively rule for or against CTCF contribution to this phenomenon. I would rather see it as a short add-on to the previous subsection.
- In general, although the analysis is very detailed and precise, some findings were either previously known or somewhat trivial (e.g., clustering of ASE in the A-compartment) to merit such long standalone passages. I suggest condensing few of these, which should also improve the flow of the text.
- I find the Discussion to be rather lengthy; some repetition of Results could be condensed/removed.

I always disclose my identity to the authors: A. Papantonis

Reviewer #2:

Parental genomes can differentially contribute to transcriptional activity, resulting in Allele-Specific gene Expression (ASE). The allelic regulation can incorporate different mechanisms, including a still poorly characterized contribution of 3D chromosome organization.

The authors performed allele-specific Genome Architecture Mapping in hybrid mouse embryonic stem cells, followed by intersection with other allele-specific and non-specific data for chromatin features (gene activity, histone modifications, RNA PolIII, accessibility, architectural proteins). Initially focusing on the GAM data, considerable allele-specific differences at different levels of chromosome organization are reported, with a strong enrichment at sites of ASE and (Polycomb) H3K27me3 peaks. Moreover, they reveal that allele-specific bias in histone gene activity coincides with large differences in genome organization, with links to H3K27me3, allelic enhancer loops and CTCF binding.

This study adds an interesting 3D chromosome structure dimension to the emerging view that H3K27me3/Polycomb mediates extensive mono-allelic/allele-specific gene activity in mammalian cells. Although I have some reservations regarding some interpretation and the somewhat complicated presentation of the results, the study reports a number of important and interesting findings that merit publication in Molecular Systems Biology.

Major considerations:

- The introduction and first section of the results create quite some expectations about GAM's capacity to resolve high-resolution chromosome structure. It's somewhat disappointing to see then that the authors settle on a 50kb resolution for downstream analysis (line 163). Particularly taking into account that allele-specific Hi-C matrixes at 25kb resolution and even single-cell Dip-C models at 20kb resolution from human cells (with 5-10 times lower SNP densities!) have been reported (Rao et al, Cell 2014; Tan et al, Science 2018; both cited). The authors should present and discuss this result in its correct context (lines 118 and 559), incorporating the sequencing read numbers required to achieve such resolution (as reviewer I have access to the meta-data of the individual sequencing runs, but not metrics like read numbers).
- The authors identify a strong link between the presence of H3K27me3, genes with ASE and allele-specific GAM structures. The H3K27me3 ChIP-seq data (and Pol2-S5p and S7p) are not allele-specific. Despite circumstantial evidence and the well-established function of H3K27me3/Polycomb in gene repression, a formal validation of allele-specific deposition and functions is lacking. Considering that this data was generated in the F123 mESC line, why did they not analyze this in an allele-specific manner? I don't understand the explanation for this in line 990. Is this due to low read numbers (and the absence of replicates)? This should be clarified. The main conclusions of the manuscript would be strengthened if the authors could perform this

analysis in an allele-specific manner, particularly the H3K27me3 analysis. The authors should consider if this merits adding additional replicates.

- I am not convinced that the large differences in TAD boundaries (Figure 1 and S2) are not in a large part due to technical limitations caused by the allele-specific mapping. Allele-specific GAM maps, obviously, have a lower sequencing depth as compared to the non-phased data, which will reduce signal over noise. Insulation score and enrichment of other chromatin features indicates that the allelic boundaries are quite weak, reducing the robustness of their calling and thus increasing both the number of false positive and false negative identifications. The authors should benchmark the sensitivity of their boundary calling pipeline relative to the sequencing depth (gains and losses of identified peaks) by downsampling the unphased data set to similar read depths as the allele-specific data sets. Alternatively, considering the decoupling between allele-specific TADs and CTCF binding (lines 603-607), could these weaker boundaries be associated with ASE promoters (e.g. Zhang et al, Nature Genetics 2023)?

Minor remarks:

- The use of the female and male signs to distinguish the chromosomes of parental origin can be confusing, as the F123 cells themselves are of male origin. Although their use is not unique to this study, the use of paternal and maternal (origin) should be prioritized (see e.g. Rao et al 2014 for an example of more appropriate mixed use).
- Lines 83, 535 and 545: Tan et al, Science 2018 should be added.
- Lines 240-243, 329-331 and 569-571: I'm not sure how ASE genes and monoallelic genes are different. This should be explained.
- Line 288: I can't find the origin of the ATAC-seq and ChIP-seq data. From the material and methods section, I gather that these were taken from previous publications? If so, these should be mentioned.
- Line 350: the nearly complete absence of co-occurrence for the S129 alleles is surprising. The authors should reflect on this, either here or in the discussion. Do the authors think that this is linked to the different contribution of allelic features between the paternal (CAST, stronger) and maternal (S129, weaker) genomes? Or do they have other explanations?
- Line 356: considering the different types of analyses in Fig. 2 and 3, and the non-allele-specific nature of the H3K27me3/Polycomb data, the quantitative contribution of different features appears difficult to assess. Unless the authors better integrate the two types of analyses, should this statement be rephrased?
- Line 485: Barshad et al, Nature Genetics 2023 should be added.
- Figures 1g and S2e: I'm not sure how the numbers in the Upset plots in the two figures relate to each other. Whereas total CAST boundaries are the same (2176) and 129 boundaries nearly (2158 vs 2154), I do not manage to see how the 3 combinations in Fig. 1g can be composed from the larger number of categories in Fig. S2e. Can the authors check?
- Figure 3d: although I realize the values on the y-axis represent a normalized value, it appears counter-intuitive to see values in the range of 100k peaks / TAD. Should this be normalized to a more realistic value?
- Figure S3a: the values on the x-axis are unreadable. From what I can distinguish, they are not identical, making the information essential for the interpretation.
- Figure S5d: I'm surprised to see many TADs of moderate size (<200 kb) that contain large numbers of active genes (up to 100) in these panels. What gene list was used to obtain these numbers? Is this the non-redundant gene list that is mention on page 27? If yes, can the authors confirm these numbers? If not, why did the authors not use the same list?
- Figures S6c and S7a are identical; the second occurrence can be removed. Am I correct that the CAST and S129 contact maps (but not the z-score map) are inverted? Adding a gradient scale to the panel will help the interpretation as well.

Manuscript Number: MSB-2024-12394

Title: Extensive folding variability between homologous chromosomes in mammalian cells

Response to reviewers

Reviewer #1:

In this manuscript, Irastorza-Azcarate and colleagues leverage GAM applied to mESCs from a hybrid mouse strain with a high load of haplotype-specific SNPs in order to examine the allele-specific architecture of the genome. Although we have long been aware of such differences existing (the authors cite a number of examples in their Introduction), the true extent of this allele-specific variance has remained unexplored. Leveraging GAM, the long contiguous genomic coverage of which simplifies phasing, together with a new computational approach in 'GAM-phaser', the authors reveal a truly pervasive variance in 3D chromatin folding between alleles (GAM appears to produce phase contacts in >3-fold more of the genome compared to Hi-C). I find this significant, as most of us in the field have been conditioned to look at average maps from both alleles in which such differences are mostly hidden. Therefore, the advance presented here can be of importance for both genome regulation (e.g., the fact that most TAD borders appear to be haplotype-specific) and genome engineering studies in the future. Overall, the manuscript text is detailed and clear, the analysis is very well conducted and conclusions are well controlled (e.g., Ring1b- and Ezh2-KO data). The two key findings (in my opinion), that most active chromatin features are biallelic and that H3K27me3 clusters produce most allelic regulation, points to 3D genome folding having a pivotal role in gene regulation. I only have few remarks that I hope the authors find useful:

1a. The authors state that "contact distance decay and momentum curves showed similar frequency of contacts between haplotypes within < 5Mb of genomic distance, but became visibly distinct at long-range distances with different haplotype preferences depending on the chromosome", but I cannot find a follow-up to this that could provide an explanation. Could they elaborate? How is this explained?

We were also intrigued about the differences in decay curves above 5Mb, and had performed extensive analyses, which did not make it to the manuscript due to length limits. To explore the differences in decay, we took the 'most differential contacts' between the two alleles, as well as the contacts that are strongest in both alleles (here called 'common contacts'; $z\text{-score} \geq 2$), at different genomic distances. We investigated whether the anchors of the different sets of contacts were enriched for specific chromatin features. Using all available features, the only enriched feature in allele-specific contacts relative to the control dataset, were contacts connecting genomic windows in LADs, for all genomic distances (**Rebuttal Fig. 1a**). This suggests that GAM captures not only contacts within each LAD but also connections across distant LADs, likely sensing LAD proximity at the nuclear periphery. As an illustration, we plotted the CAST and S129 GAM contact maps and differential contacts up to 50 Mb from chromosome 11 (**Rebuttal Fig. 1b**).

Rebuttal Figure 1. a) Frequency of detecting LADs at both contact anchor points in CAST specific, S129 specific, and common contact sets, compared to expected mean counts derived from 10,000 permutations of each respective set. b) From top to bottom: Contact intensities on chromosome 11 up to 50 Mb distance for CAST and S129; differential z-score contact map for selected contacts having delta z greater than 2.0 (S129-specific, orange) or less than $z=-2.0$ (CAST-specific, purple); LAD-LAD pair combinations (50 kb resolution data) colored by the median value of the delta z-scores of contained differential contacts if greater or less than 1, or grey otherwise; annotated LAD regions (50 kb data); UCSC tracks of protein-coding ASE genes. c) Median of the delta z-scores for all LAD-LAD patches on chromosome 11 sorted by genomic distance and colored when patch contact median is greater or less than 1; and d) expected background distribution after circular shifts of LAD positions. e) Median of the delta z-scores for each LAD-LAD patch genome-wide sorted by genomic distance and colored when this median is greater or less than 1. Quantile indicators for 0.25, 0.5, 0.75. Contrasted by data using randomized LAD positions obtained from circular shifts.

Interestingly, visualization of differential contacts in matrices shows that contacts of the same haplotype are observed in patches connecting LADs, suggesting striking preferences for LAD-LAD contacts in each genotype. To more directly assess whether specific LAD pairs contact each other more often in one haplotype than the other, we calculated the median of the delta Z-scores for every LAD-LAD contact patch on chromosome 11. We found that the LAD-LAD

patches have a strong preference for CAST or S129 contacts (**Rebuttal Fig. 1c**; similar results were observed for the other chromosomes). Moreover, the median z-scores in the LAD-LAD patches tend to be stronger with increased distance, suggesting that at the larger genomic distances, LAD-LAD contacts are strong compared with other genomic regions at the same distance. This pattern is lost when randomly shuffling the delta values (**Rebuttal Fig. 1d**). These haplotype-specific contact patches and their increased Z-score with distance were also seen genome-wide (**Rebuttal Fig. 1e**).

LADs are known to play a significant role in chromosome structure and gene regulation by organizing the genome around the nuclear lamina. Our results suggest that LAD-LAD contacts likely contribute to the distinct haplotype structural preferences observed especially at long-range distances. As the available published LAD datasets are mapped in a different ESC clone than all other datasets (mapped in F123), and we decided against focusing the current manuscript on the investigation of preferred LAD-LAD contacts at the nuclear or nucleolar periphery, or the specific patterns observed. We hope this clarification addresses the reviewer's concerns and provides a more comprehensive understanding of our results.

1b. Has this got something to do also with the resolution of this GAM dataset?

The different decay at longer genomic distances is not due to the resolution of the GAM dataset, as it can be observed at both 50 and 250 kb resolution (**Rebuttal Figure 2**). Visual inspection of whole chromosome maps intuitively agrees with this quantitative observation (manuscript EV Fig. 2a), as along the diagonal there is a similar range of contact intensities, whereas at longer distances the range of stronger contacts is more varied. For example, we see the same number of TAD borders per chromosome (manuscript Fig. 1e), albeit in different positions, but the long range contacts are more haplotype specific.

Rebuttal Figure 2. Distance decay and momentum curves plotted from NPMI contact maps, (a) at 50 kb resolution (from EV Figure 2a) and (b) at 250 kb resolution.

2. Is my interpretation **that most allelic regulation comes via H3K27me3-mediated repression correct?** If so, since the authors classify about 1/3 of ASEs as H3K27me3+S5p+S7p+, **how would the rest be regulated?**

“that most allelic regulation comes via H3K27me3-mediated repression correct?” Out of all the explorations performed to detect specific regulatory mechanisms that could explain all or most ASE genes, H3K27me3 came up as the most prominent, but it only explains about one third of the genes with ASE imbalance. In our revised manuscript, we have now investigated whether Polycomb repression is functionally important to modulate the expression levels of ASE genes, by taking advantage of published data for RING1A^{KO}-RING1B^{AID}-acute protein depletion in E14-tg2A ESCs (Dobrinčić et al. 2021, NSMB; PMID: 34608337), which are unfortunately not hybrid ESCs. These analyses identify 290 ASEs which are (a) marked by H3K27me3 in F123-ESCs and (b) up-regulated in the Ring1B^{AID}-E14-ESCs upon Ring1B depletion (**Rebuttal Fig. 3**). A strong association with Polycomb repression mechanisms is also seen for the remaining 487 ASE genes which are either associated with upregulation after Ring1B depletion (219 genes), or marked by H3K27me3 (268), likely due to the mismatch in ESC lines and culture conditions, or in an imperfect overlap between PRC2 deposition of H3K27me3 and PRC1 deposition of H2Aub1 through Ring1B activity. We also found that 395 ASE genes which are not marked by H3K27me3 nor upregulated upon Ring1B depletion, reinforcing the view that ASE imbalance is also associated with other mechanisms at a subset of ASE genes. These new analyses were added to the revised manuscript, including new manuscript Fig. 2f, new paragraph in section “H3K27me3 occupies a third of ASE gene promoters”, further section in Methods and table in permanent data repository Irastorza-Azcarate et al., 2024.

Rebuttal Figure 3. (as in the revised manuscript) Upset plot shows how many ASE genes are up- or down-regulated after 8 h of acute PRC1-AID knock-down, and whether they are marked by H3K27me3 at their promoters. Color coded are the fraction of genes belonging to ASE genes expressed more in the CAST allele, or the S129 allele. Genes were classified based on DESeq2 LFC value: AID up have LFC ≥ 0.5 , AID down LFC ≤ -0.5 , AID NC $-0.5 < \text{LFC} < 0.5$.

The view that Polycomb repression likely acts in combination with other mechanisms is supported by visual inspection of the genomic neighborhoods of ASE genes, where a combination of candidate mechanisms can be observed, for example, the histone locus.

“If so, since the authors classify about 1/3 of ASEs as H3K27me3+S5p+S7p+, how would the rest be regulated?” To address this question, we summarize the combined occupancy data at each ASE gene promoter, based on their overlap with phased and unphased peaks, which we have made available from the permanent data Zenodo repository (Irastorza-Azcarate

et al, 2024; DOI <https://doi.org/10.5281/zenodo.14066696>; **Rebuttal Figure 4a**). We measured the overlap with ASE gene promoters of allele-specific or biallelic peaks of CTCF, H3K4me3, H3K27ac or ATAC, differential DNA methylation, and presence/absence of Pol2-S7p, -S5p or H3K27me3. Very few allele-specific features occur exactly at ASE gene promoters, suggesting that ASE imbalance is either (a) not regulated locally at promoters, (b) is regulated by other local mechanisms which we have not mapped, or that (c) other more complex effects are at play, namely regulation mediated by inter- and intra-genic chromatin regulation through three-dimensional chromatin structure. The clustering of ASE genes in TADs, which are also enriched for Polycomb occupancy, could itself create structural bystander effects that would influence ASE imbalance upon changes in Polycomb repressor complexes. Although ASE gene promoters are not often directly associated with allelic-specific features, they often overlap with many common biallelic features, including H3K27me3 (Rebuttal Figure 4b). Since most ASE genes contain common active chromatin features and CTCF, we suspect that haplotype-specific chromatin structure could itself contribute to the ASE imbalance. We have not included these analyses in the manuscript, due to manuscript space limits, and welcome recommendations from the editor and/or reviewers if inclusion is preferred.

Rebuttal Figure 4. (a) Upset plot with the group overlaps for ASE genes associated with allele-specific (AS) features, allele-specific DNA methylation and Polycomb at ASE gene promoters, and (b) for ASE genes depleted of allele-specific features sharing common H3K27ac, ATAC or H3K4me3 peaks at their gene promoter.

3a- Looking at Fig. 5 and the attempt at dissecting E-P contacts, I have the following questions: What is the effective resolution of phased GAM matrices (as I might have missed this)?

The driving factor for the effective resolution of GAM datasets is the number of collected nuclear profiles (NPs; see Beagrie et al. 2023 Nat. Methods PMID: 37336949, in particular Extended Fig. 3b,c,f therein). The GAM data analysis pipeline extensively explores sample quality (gamtools; <https://gam.tools/>; Beagrie and Schueler, 2017, bioRxiv doi: <https://doi.org/10.1101/114710>). Based on the co-segregation sampling statistics, the quality of F123 phased GAM datasets produced is robustly compatible with analysis of allele-specific chromatin folding down to 20 kb resolution, for contacts with genomic distances within 10 Mb, or down to 10 kb resolution for contacts within 2 Mb (see manuscript EV Fig. 1g). To help showcase the quality of matrices at higher resolutions, we plotted chr5 regions across multiple resolutions from 10 to 50 kb (rows) in a 5 Mb region (**Rebuttal Fig. 5a**) and a 50 Mb region (**Rebuttal Fig. 5b**; high quality images of these matrices are provided in pdf format, in Appendix file to the Rebuttal Letter). As a future resource for the field, we have uploaded to GEO (accession GSE254717) all the phased chromatin contact maps for CAST and S129 allele at 10, 20, 30, 50 and 250 kb resolutions. For this manuscript, we settled for using 50 kb resolution at all genomic distances, to be able to coherently compare different structural features across different length scales.

Rebuttal Fig 5. Phased GAM maps for (a) Chr5:20-25 Mb, and (b) Chr5:5-55 Mb for CAST and S129 haplotypes at 10, 20, 30 and 50 kb resolution.

3b- If one compares the differences marked by the authors in Fig. 5d and 5e, I would argue that contact changes in panel e are obvious, but not much so in panel d. Is this the magnitude of changes to be expected?

We agree with the reviewer that the contacts in Fig. 5d in the first manuscript submission were less obvious, especially because we had plotted all the strong contacts in the CAST allele, regardless of their strength in S129, although in this section were highlighting the contacts

strong in one allele but weak in the other. For improved clarity, we have revised the figure to only highlight contacts involving CAST-specific E-P contacts which are weak in S129. The genes involved in the contacts remain the same. For consistency, we modified Fig. 5d in the same manner. Both revised panels are shown below (**Rebuttal Fig. 6**).

Rebuttal Figure 6. Revised Figure 5d (left) and 5e (right), highlighting only contacts involving E-P contacts that are stronger in one of the alleles.

3c- Perhaps the authors can produce a boxplot summarizing the strength changes in these AS contacts? Then, they would be able to also provide some statistics on this (which also seem to be missing from the comparisons in panel 5c)?

We thank the reviewer for this valuable suggestion. To enhance the clarity and statistical robustness of the analysis, we have produced box plots summarizing the strength changes in these allele-specific NPMI contacts. Four new plots are included in EV Fig. 7c and 7d of the revised manuscript, displaying the contact intensity for the four different types of allele-specific enhancer-promoter (EP) contacts (**Rebuttal Fig. 7**). We have also performed statistical analysis to provide a more comprehensive comparison, and found highly significant differences in all.

Rebuttal Figure 7. Comparison of genome-wide contact intensities found in the CAST and S129 haplotypes for the four different combinations of allele-specific enhancer-promoter (EP) associated with CAST contacts (a) and S129 contacts (b). Added as EV Fig. 7c and 7d, respectively. Two sample T-test P-values (number of samples; from left to right): $4.17e-10$ ($n=84$), $8.78e-17$ ($n=130$), $3.92e-12$ ($n=112$), and $4.42e-14$ ($n=129$).

4- Also in respect to Fig. 5 and this whole section, it would be useful to readers if the authors explained how "general decompaction" of the loci is measured and exemplified in GAM matrices.

We have revised the manuscript to further explain how the GAM technology inherently detects relative differences in chromatin compaction, and have included new schematics in the manuscript (new EV Fig. 5g; also shown in **Rebuttal Fig. 8**). If we consider two loci with equal genomic length but different compactness, the actual volume of the more compact locus is relatively smaller than the least compact locus, even though they have the same DNA content. Through the slicing process of GAM data collection, loci with larger volume will be captured more frequently than loci with smaller volume. As a measure of relative chromatin compaction, we calculate Window Detection Frequency (WDF), as the number of nuclear slices (NPs) where each window is captured, relative to the total number of nuclear slices collected. In Beagrie et al. 2017 (EV. Fig. 9c therein), we had shown that higher WDF correlates with higher gene expression levels and higher chromatin accessibility, as expected from gene expression being associated with local chromatin decondensation of the transcribed region. In the revised manuscript, we confirm these relationships by comparing WDF in different alleles relative to their ASE imbalance (manuscript Fig. 5d,e,h), and we further expand the differences in WDF in relation to Polycomb occupancy. (Added new text in Results, Methods and new Figure legend EV Fig 5g.)

Rebuttal Figure 8. Concept of detecting compacted and decondensed chromatin querying GAM cryosections and the implications at the level of Window Detection Frequency (WDF). Figure adapted from Beagrie et al. 2017 (EV Fig. 9c. therein) for the representation of allele-specific differences in compaction and added as EV Figure 5g.

5- The last part of the Results on CTCF loops involved in ASE, does not merit a segment of its own as it does not conclusively rule for or against CTCF contribution to this phenomenon. I would rather see it as a short add-on to the previous subsection.

We have merged the CTCF and E-ASE gene contact sections.

6- In general, although the analysis is very detailed and precise, some findings were either previously known or somewhat trivial (e.g., clustering of ASE genes in the A-compartment) to merit such long standalone passages. I suggest condensing few of these, which should also improve the flow of the text.

We have simplified this section of the text, as suggested.

7- I find the Discussion to be rather lengthy; some repetition of Results could be condensed/ removed.

We have revised the discussion to avoid repetition of Results.

Reviewer #2:

Parental genomes can differentially contribute to transcriptional activity, resulting in Allele-Specific gene Expression (ASE). The allelic regulation can incorporate different mechanisms, including a still poorly characterized contribution of 3D chromosome organization.

The authors performed allele-specific Genome Architecture Mapping in hybrid mouse embryonic stem cells, followed by intersection with other allele-specific and non-specific data for chromatin features (gene activity, histone modifications, RNA PolII, accessibility, architectural proteins). Initially focusing on the GAM data, considerable allele-specific differences at different levels of chromosome organization are reported, with a strong enrichment at sites of ASE and (Polycomb) H3K27me3 peaks. Moreover, they reveal that allele-specific bias in histone gene activity coincides with large differences in genome organization, with links to H3K27me3, allelic enhancer loops and CTCF binding.

This study adds an interesting 3D chromosome structure dimension to the emerging view that H3K27me3/Polycomb mediates extensive mono-allelic/allele-specific gene activity in mammalian cells. Although I have some reservations regarding some interpretation and the somewhat complicated presentation of the results, the study reports a number of important and interesting findings that merit publication in Molecular Systems Biology.

Major considerations:

1- The introduction and first section of the results create quite some expectations about GAM's capacity to resolve high-resolution chromosome structure. It's somewhat disappointing to see then that the authors settle on a 50kb resolution for downstream analysis (line 163). Particularly taking into account that allele-specific Hi-C matrices at 25kb resolution and even single-cell Dip-C models at 20kb resolution from human cells (with 5-10 times lower SNP densities!) have been reported (Rao et al, Cell 2014; Tan et al, Science 2018; both cited). The authors should present and discuss this result in its correct context (lines 118 and 559), incorporating the sequencing read numbers required to achieve such resolution (as reviewer I have access to the meta-data of the individual sequencing runs, but not metrics like read numbers).

“incorporating the sequencing read numbers required to achieve such resolution (as reviewer I have access to the meta-data of the individual sequencing runs, but not metrics like read numbers)”

We thank the reviewer for highlighting this issue. We have expanded the Methods section entitled “Determining resolution of pairwise co-segregation matrices” to provide a more explicit explanation about why sequencing depth is not limiting for GAM data (please see also answer to Rev #1 points 3a,b), as follows.

The quality of chromatin contact maps from GAM data can be defined by two main metrics: resolution of genomic bins, and contact detectability (number of entries in the contact matrix which were observed at least once). The effective resolution of GAM datasets generally depends on the number of

NPs collected (Beagrie et al, 2023), as each GAM sample contains only 5-15% of the genome, and enough NPs are necessary to sample the co-segregation of all possible genomic windows in each chromosome. The reads sequenced in each GAM sample are used to identify presence or absence of genomic regions in that sample in a binary fashion that does not directly affect the sensitivity to detect contacts. The chromatin contacts are defined as normalized cosegregation frequencies between genomic bins, and their sensitivity depends on how many events are counted (i.e. how many GAM samples were collected). Since each GAM sample has so little DNA (5-10% of the DNA of a single cell), the sequencing depth required to detect positive windows in each sample is promptly saturated with a low sequencing depth of 2-3 million reads per sample NP in the present data. This is approximately double the depth used in the first GAM manuscript, of 1-2 million (Beagrie et al., 2017; Beagrie et al., 2023).

“(as reviewer I have access to the meta-data of the individual sequencing runs, but not metrics like read numbers)”.

EV Dataset 10 listed all QC metrics for each GAM sample (library), including the number of sequencing reads per sample. We have now added a note in the Methods section to flag that the sequencing depth information is listed in this EV Dataset, to aid readers find this information more easily.

“Particularly taking into account that allele-specific Hi-C matrices at 25kb resolution and even single-cell Dip-C models at 20kb resolution from human cells (with 5-10 times lower SNP densities!) have been reported (Rao et al, Cell 2014; Tan et al, Science 2018; both cited).”

Like in Rao et al. 2014, we could have chosen to analyze the present F123 GAM dataset at the same or even higher resolutions. We believe that the choice of 25kb in the Rao analyses, whereas very interesting and pioneering for the field in 2014, was not sufficient to make genome-wide quantitative statements about the extent of allele-specific structural differences across the whole genome, due to limited information content. In agreement with this view, the Rao et al. manuscript showcases only a few examples of allele-specific differences and steers away from making genome-wide statements. The same considerations are true for another pioneering manuscript Tan et al. 2018. Here, we were not searching for examples, we aimed to identify the global extent of allele-specific differences in chromosome structure.

To directly compare the quality (information content) of the present GAM dataset with the Rao et al. Hi-C contact dataset, we measured the contact detectability (number of informative entries in the contact matrices) across all distances or within 10Mb, at 10 and 50 kb resolutions, in both the human Hi-C data mentioned (Rao et al., 2014, GSE63525), and the F123 GAM dataset. For Hi-C, ‘*informative matrix entries*’ are those with 1 or more intra-chromosomal ligation events between any two genomic windows (1 ligation event originates from a single paired end read). For GAM, ‘*informative matrix entries*’ are those with co-detection of any two genomic windows in at least one GAM sample (requiring at least 78 base pairs of genomic coverage, or at least two or more reads, in each of the contacting windows; ie 4 sequencing reads in total).

Taking all possible intra-chromosomal contact entries, we find that the published human phased Hi-C matrices only have about 1.8-7.0% of informative contact entries at 10 kb resolution, depending on chromosome, whereas the phased F123 GAM matrices contain 56-77% of informative contact entries (**Rebuttal Fig. 9a**). At 50 kb, phased Hi-C matrices have 20-51% informative entries whereas phased F123 GAM matrices contain 79-93% (**Rebuttal Fig. 9b**). As chromatin contacts in Hi-C are often studied closer to the diagonal, we also compared the detectability of matrix entries in phased Hi-C and GAM matrices at shorter contact distances. Taking all possible intra-chromosomal contact entries within a 10 Mb distance, we found that the human phased Hi-C matrices only have about 11-15% of informative entries at 10 kb resolution, whereas the phased GAM matrices contain 68-83% (**Rebuttal Fig 9c**). At 50 kb, phased Hi-C matrices have 49-71% of informative entries, while the phased GAM matrices contain 85-96% (**Rebuttal Fig. 9d**). We have added the comparison of informative matrix windows between GAM and Hi-C at 50 kb resolution as a new panel in EV Fig. 1f, and reported these interesting results in manuscript section “GAM-phaser: a pipeline to phase GAM data”.

Rebuttal Figure 9. Percent of informative window pairs (“contacts”) in published phased Hi-C GM12878 chromatin contact maps (Rao et al., 2014, GSE63525) and GAM phased F123 chromatin contact maps at (a) 10 kb resolution for all genomic distances, (b) 50 kb resolution for all genomic distances, (c) 10 kb resolution for 10 Mb genomic distance, (d) 50 kb resolution for 10 Mb genomic distance. The published Hi-C phased chromatin contact matrices were first converted from binary format to *2bedgraph* format using *hictk* package (<https://www.biorxiv.org/content/10.1101/2023.11.26.568707v2>). Next, we counted the percent of NaN entries per chromosome for all genomic distances and 10 Mb genomic distance. For

GAM data, we counted the percent of NaN entries in raw co-segregation phased chromatin contact matrices per chromosome for all genomic distances and up to 10 Mb genomic distances. Even though the F123 phased GAM datasets could have been analyzed at higher resolution in the present study (e.g. 10 or 20 kb), we preferred to focus on 50 kb resolution, to have higher certainty of identifying haplotype-specific 3D genome conformations and to explore larger genomic distances. The quality of the higher resolution matrices in GAM data can be seen by visual inspection of GAM matrices with different resolutions (from 10 to 50 kb; rows) in a 5 Mb region (**Rebuttal Fig. 10a**) or a 50 Mb region (**Rebuttal Fig. 10b**; higher quality images of the matrices are shared in pdf format as an Appendix file to the Rebuttal Letter. Differences between the CAST and S129 contact maps are clearly visible at 10 kb resolution, both locally and at larger genomic distances, as expected from the detectability of co-segregation events across most matrix entries.

Rebuttal Figure 10. (same as Rebuttal Figure 4) Phased GAM maps are shown for (a) chr5:20-25 Mb and (b) chr5:5-55Mb for each haplotype CAST and S129 at 50 kb resolution, as used in the manuscript, together with higher resolution data at 30, 20, and 10 kb, which is made available as a resource through GEO (accession number GSE254717).

These analyses indicate that even though human phased Hi-C matrices have been explored in the literature, and some allele-specific contacts were identified, the majority of entries in the published phased human H-C matrices are devoid of contact information at the chosen resolutions, making it difficult to quantify the extent of haplotype-specific 3D genome structure from published phased Hi-C data. For the current manuscript, we were especially interested in developing a broad and robust investigation of structural allelic imbalance and how it relates with genome-wide ASE imbalance. We deliberately chose 50 kb to minimize false negatives, avoiding missing contact data present in the 10 kb resolution, even though it was only relatively minor loss of contact information (20-30%). By choosing to focus the present manuscript at 50 kb, we are confident that the structural differences reported are the most robust. As a future resource for the community, we had uploaded phased chromatin contact maps, for CAST and S129 allele at 10, 20, 30, 50 and 250 kb resolutions to GEO (accession number GSE254717).

2. The authors identify a strong link between the presence of H3K27me3, genes with ASE and allele-specific GAM structures. The H3K27me3 ChIP-seq data (and Pol2-S5p and S7p) are not allele-specific. Despite circumstantial evidence and the well-established function of H3K27me3/Polycomb in gene repression, a formal validation of allele-specific deposition and functions is lacking. Considering that this data was generated in the F123 mESC line, why did they not analyze this in an allele-specific manner? I don't understand the explanation for this in line 990. Is this due to low read numbers (and the absence of replicates)? This should be clarified. The main conclusions of the manuscript would be strengthened if the authors could perform this analysis in an allele-specific manner, particularly the H3K27me3 analysis. The authors should consider if this merits adding additional replicates.

We appreciate the reviewer's comments towards more carefully evaluating our claims related to the role of Polycomb repression in ASE imbalance. In brief, we have now included in the manuscript the phasing information of Pol2-S5p, Pol2-S7p and H3K27me3 (see permanent data repository Irastorza-Azcarate et al, 2024; DOI <https://doi.org/10.5281/zenodo.14066696>). We have also used published data for AID-dependent Polycomb depletion in ESCs (Dobrinić et al. 2021, NSMB; PMID: **34608337**) which strengthens the claims that Polycomb is associated with partial repression of ASE genes in ESCs. We have also toned down our claims throughout the manuscript, to report a strong *association* between Polycomb occupancy and ASE imbalance, and avoided claiming that Polycomb *mediates* it. We explore these points in more detail below.

Functional validation

To further explore a functional role of Polycomb in the repression of ASE genes, we have taken advantage of the available published data for AID-mediated acute depletion of Ring1B in a Ring1A constitutive knockout ESCs (RING1A^{KO}RING1B^{AID} in E14-tg2a ESCs; Dobrinić et al. 2021, NSMB; PMID: 34608337). We found 290 ASEs in F123-ESCs which are both marked by H3K27me3 in F123-ESCs and are up-regulated upon Ring1B depletion in the AID-E14-ESCs (**Rebuttal Fig. 12**). Other ASE genes may also be under Polycomb influence, as they are either characterized by (a) having H3K27me3 in F123 ESCs (219 genes) or (b) by being upregulated in AID E14 ESCs. If we consider the two pieces of evidence, up to 777 ASE genes are candidate Polycomb targets (**Rebuttal Fig. 11**). These new analyses are in line with the mass-spectrometry results after conditional Ezh2 or Ring1b depletion, measured at the protein levels. The new results were added to the revised manuscript, including a new panel in manuscript Fig. 2f, a new paragraph in the section "H3K27me3 occupies a third of ASE gene promoters", a new section in Methods ("ASE upregulation upon AID-dependent acute depletion of RING1B protein in a Ring1a knockout ESC line") and an updated Zenodo repository (Irastorza-Azcarate et al., 2024; DOI <https://doi.org/10.5281/zenodo.14066696>).

Rebuttal Figure 11. (same as Rebuttal Fig. 3 in response to Rev #1) Upset plot shows how many ASE genes are up- or down-regulated after 8 h of acute PRC1-AID knockdown, and whether they are marked by H3K27me3 at their promoters. Color coded are the fraction of genes belonging to ASE genes expressed more in the CAST allele, or the S129 allele. Genes were classified based on DESeq2 LFC value: AID up have $LFC \geq 0.5$, AID down $LFC \leq -0.5$, AID NC $-0.5 < LFC < 0.5$.

This analysis also confirms that many ASEs (395) are not marked by H3K27me3 nor sensitive to Ring1B depletion, supporting the view that other mechanisms are likely at play to generate their expression imbalance. In the answers to Reviewer 1, point 2, we further discuss other possible mechanisms and the likely importance of combinations thereof (**Rebuttal Fig. 4**, rebuttal page 5).

Quality of H3K27me3 datasets

For reassurance about the quality of the F123 H3K27me3 ChIP data, we share below comparisons with H3K27me3 occupancy from other datasets (Ferrai et al. 2017, PMID: 29038337; Mikkelsen et al. 2007, PMID: 17603471), in ESC clones 46C and V6.5, respectively (**Rebuttal Fig. 12**).

Rebuttal Figure 12. Examples of single-locus ChIP-seq H3K27me3 profiles in V6.5 mESC (Mikkelsen et al, 2007), 46C mESC (Ferrai et al, 2017) and non-phased, CAST phased and S129 phased reads for F123 mESC. (a) *HoxA* cluster chr6:52019148-52467919, (b) *Nkx2.2* gene locus chr2:147,153,345-147,217,353, (c) *Gata6* gene locus chr18:11,033,839-11,098,454. The y-axis shows sequencing depth.

Our previous experience mapping Polycomb occupancy in various ESC lines (clone E14; Brookes et al, 2012 PMID: 22305566; clone 46C, Ferrai et al, 2017 PMID: 29038337) indicated that the patterns are highly reproducible. We produced one biological replicate of the H3K27me3 ChIP data in mESC F123, which was sequenced with higher depth (9.3×10^7 uniquely mapped reads), compared with 4×10^7 in Ferrai et al. (2017) or 1×10^7 in Mikkelsen et al. (2007). Visual inspection of UCSC genome browser tracks shows excellent similarity in the distribution and enrichment of H3K27me3 marks at *Hox* clusters, the *Nkx2.2* and *Gata6* loci

between the single F123 H3K27me3, with the other datasets (**Rebuttal Figure 12a-c**). After recalling H3K27me3 peaks from Mikkelsen et al. using the same approach (Bayesian Change-point Model peak-finder, as described in Methods), we found that >82 and >89% of all H3K27me3 peaks found in the Mikkelsen or Ferrai datasets, respectively, were also found in F123 H3K27me3 ChIP data.

In **Rebuttal Fig. 12**, we also represent the reads with SNP throughout the regions (purple and orange tracks), showing that the sequencing depth was ample to detect allelic occupancy of Polycomb. Further, this figure also shows the complex distribution of CAST- and S129-phased reads across the characteristic long peaks of H3K27me3 occupancy, with some segments of the same peak often having more CAST reads, and other segments having more S129-specific reads, likely resulting in an underestimation of allele-specific presence of H3K27me3 due to its extended occupancy.

Adding phasing information to H3K27me3 analyses

We have added the phased peaks of H3K27me3, Pol2-S5p and Pol2-S7p to the revised manuscript (permanent data repository Irastorza-Azcarate et al., 2024 <https://doi.org/10.5281/zenodo.14066696>; revised Fig. 2g,h, and EV Fig. 8b).

a Figure 2g

b Figure 2h

Rebuttal Fig. 13. Revised manuscript Fig. 2g (a) and 2h (b), including the classification of Pol2-S5p, Pol2-S7p, and H3K27me3 peaks according to whether they are common to both alleles, CAST- or S129-imbalanced, or do not contain reads with SNP.

To identify allele-specific H3K27me3 peaks, we applied the same strategy used for other epigenetic marks, and found 549 CAST and 272 S129 H3K27me3 peaks with allele-specific imbalance (out of total 17,938 peaks; **Rebuttal Fig. 13** and revised manuscript Fig. 2g and Fig 2h). Only 37 and 65 of the allele-specific H3K27me3 peaks directly overlap with ASE gene promoters or coding regions, respectively (data available in the permanent data repository Irastorza-Azcarate et al., 2024 <https://doi.org/10.5281/zenodo.14066696>). Even though many H3K27me3 peaks show allelic imbalance, in a similar proportion to other chromatin features, allele-specific H3K27me3 peaks are rarely found at gene promoters, supporting the view that allele-specific Polycomb effects may be mediated through long-range and/or structural mechanisms. The volcano plots representing the differences in allelic read numbers relative to fold change is shown below (**Rebuttal Fig. 14**) and also included in manuscript EV Fig. 8.

Rebuttal Figure 14. Updated EV Figure 8b. Distribution of S5p, S7p, and H3K27me3 peaks, showing their absolute difference in phased reads (CAST - S129) and their log2 fold change.

To further support our hesitation in using phased broad peaks of H3K27me3 (and Pol2) occupancy, we share examples of phased peaks in **Rebuttal Figs. 15 and 16**. In **Rebuttal Fig. 15**, we show some of the few examples of ASE genes with allele-specific H3K27me3 peaks on their promoters (panels a, b) or gene bodies (panels c, d), which often have low read depth.

Rebuttal Figure 15. Examples of phased H3K27me3 profiles for single ASE genes (a) *Tcirgi* CAST ASE gene with S129 specific Polycomb peak on the promoter, (b) *Sfn9* S129 ASE gene with CAST specific polycomb peak on the promoter, (c) *Tubg1* CAST ASE gene with S129 specific polycomb peak in the

gene body, (d) *Nfatc3* S129 ASE gene with CAST-specific Polycomb peak in the gene body. The y-axis shows sequencing depth.

The main difficulty we encountered was that the absolute majority of H3K27me3 peaks are extremely long (median 14.4 kb, mean 3.4 kb; manuscript EV Fig. 3a) and their phasing is driven by a relatively small proportion of SNP-containing reads present in discrete regions of the large peak. Examples of long peaks with discrete CAST and S129 read distributions in intergenic and promoter regions are shown in **Rebuttal Fig. 16a, b**. Another complicated example is the *Pcdh* locus, where peaks with lower overall H3K27me3 enrichment end up being called allele specific, some are fused, and within the fused peaks we can visually observe incongruent enrichments in different parts of the fused peak (**Rebuttal Fig. 16c**). We believe that the additional replicates and/or increased sequencing depth, although valuable, would not help to resolve these specific issues, since the datasets have excellent quality compared with other widely accepted published datasets, and they were sequenced at very high depth.

Rebuttal Figure 16. Examples of difficult to phase H3K27me3 long (> 4 kb) peaks. (cont. next page) (a) H3K27me3 peaks that were classified as CAST specific due to LFC >2 and number of reads >10, show a very sparse distribution of both CAST and S129 reads along the peak length. (b) The same for two H3K27me3 peaks classified as S129 specific. Note also the differential enrichments of CAST and S129 phased reads in different parts of the same H3K27me3 peaks, which complicates their haplotype

assignment. (c) Protocadherin gene cluster at chr18:36,923,615-37,032,127. Allele specific peaks are highlighted in blue, unphased peaks are highlighted in red.

Several options could be considered to deepen the analyses of H3K27me3 allele specificity, but we felt that they would require extensive tuning and detail and for this reason are beyond the theme of the present manuscript focused on showcasing in a genome-wide manner the extensive allele-specific structural differences in genome topology. One option would be to focus only on the stronger peaks (e.g. with narrow peak finders such as MACS) and ignore the broader H3K27me3 occupancy. In our previous work (Brookes et al. 2012, Ferrai et al. 2017), we showed the value of broad peak calling for Polycomb and Pol II, and this enabled the discovery of 2,000 extra Polycomb target genes, including the observation that metabolic and signaling genes are partially repressed by Polycomb (classified as PRC active genes), which we confirmed in the present paper. PRC-active genes have previously been explored by single cell RNA-seq and suggested to be regulated through structural proximity to active enhancers (Kar et al., 2017, Nature Comm. PMID 28652613). Therefore, focusing on the tip of the iceberg in allele-specific Polycomb deposition is not an appealing research avenue. Another option would be to break the Polycomb peaks in 50 kb windows and classify the windows in terms of allele specificity, but this choice has no biologic meaning and is difficult to integrate with genes of different length. Finally, it would also be possible to consider read differences to explore the genome-wide imbalance between reads with SNP only, but these analyses are affected by sequencing noise. We dedicated a significant amount of effort to tune the phasing of long peaks for H3K27me3 and Pol2-S5p and -S7p, to a satisfactory manner, but we were not convinced by the value of the different outcomes.

Beyond the present manuscript, future 3D genome structure analyses of allele-specific chromatin conformation upon Polycomb knockout, depletion or inhibition will be essential to pinpoint the stronger contributions of Polycomb repression on ASE imbalance over more local effects driven by genetic variability of regulatory elements and transcription factor activity. Our study uncovers differences in 3D genome structure and begins the exploration of a huge complexity and combination of mechanisms that can contribute to ASE imbalance.

3a- I am not convinced that the large differences in TAD borders (Figure 1 and S2) are not in a large part due to technical limitations caused by the allele-specific mapping. Allele-specific GAM maps, obviously, have a lower sequencing depth as compared to the non-phased data, which will reduce signal over noise.

We understand that the effects of sequencing depth in GAM data are not trivial, and we would like to reassure the reviewer that the allele-specific GAM maps at 50kb resolution do not suffer from lower sequencing depth. Typically, a given genomic window which is found present in a single GAM sample (i.e. an independent sequencing library) originates from only one allele. To this end, we had shown in the manuscript, using DNA-FISH, that ~90% of nuclear slices containing a given window have only one of the two alleles (EV Fig. 1f). The sampling of only one allele per GAM sample occurs because the nuclear slices (NPs) are very thin (~200 nm thick, or approx. 5-10% of the nuclear volume). The number of actual reads required to detect the presence of each window is therefore the same in a phased or non-phased dataset; in other words, the same number of reads contributes to detecting the presence of a given window in a

phased or non-phased dataset, because the window is only present from one allele in the vast majority of cases. By phasing the GAM data, we are simply allocating windows to different alleles based on the detection of SNP-containing reads. Importantly, the positive detection of genomic windows in GAM data is based on the sequencing of many reads, and the datasets are sequenced to achieve saturation of sequencing reads per window.

Rebuttal Figure 17. Schematics illustrating that most phased windows are called in each GAM sample only from one allele and have abundant unphased reads. Teal, unphased reads (top) or reads without SNP (below in schematics), purple/orange, CAST- or S129-SNP containing reads.

To further support that the allele-specific maps do not have lower sequencing depth, we state here in more detail, specific aspects of the phasing pipeline. GAM-Phaser was developed to take advantage of the fact that, in GAM data, most windows in each nuclear slice originate from a single allele. In GAM-Phaser, we use all our sequencing depth to determine whether each given window is present or absent in a given GAM sample (**Rebuttal Fig. 17**, both teal and purple/orange reads are considered to call positive windows per dataset). First, we calculate the minimum number of reads to call positive genomic windows using all reads, in each GAM sample, and at each genomic resolution (since NPs vary in genomic complexity depending on whether they are more or less equatorial, the sequencing depth per window varies with NP). After defining the positive window detection threshold, GAM-Phaser takes this same minimum threshold to decide which windows can be phased in each given sample, by considering only SNP-containing reads (only purple or orange; **Rebuttal Fig. 18**). In other words, we require that enough SNP-containing reads are present in each positive phased window with the same threshold used when considering all reads. In our F123 GAM dataset, we found that an average of 70% of all positive windows in the dataset passed this threshold, further supporting that the depth of sequencing is amply sufficient to work at 50kb resolution (EV Fig. 1d). We believe that this is a *most conservative approach to define allele-specific contacts*, in this first effort to define allele-specific chromatin contacts in GAM data, but there is ample scope to relax this threshold in future studies, which would facilitate the exploration of allele-specific contacts across more genomic regions and higher resolutions.

Rebuttal Figure 18. Schematics showing how calling of positive windows depends only on the total number of reads. To phase CAST- or S129-specific windows, the number of SNP containing reads must reach the minimum number of total reads required to call a window positive; this is a most stringent cutoff for phasing GAM data. Teal represents unphased reads (reads without SNP) and purple represents SNP-containing reads mapped to the CAST allele.

3b. Insulation score and enrichment of other chromatin features indicates that the allelic borders are quite weak, reducing the robustness of their calling and thus increasing both the number of false positive and false negative identifications. The authors should benchmark the sensitivity of their border calling pipeline relative to the sequencing depth (gains and losses of identified peaks) by downsampling the unphased data set to similar read depths as the allele-specific data sets.

To test the robustness of allelic specific borders, we first downsampled the GAM dataset by randomly taking only 40, 60, or 80% of all NPs (i.e. independent GAM samples), and re-mapped phased and unphased matrices. We recalculated TAD borders from all eroded datasets, and compared the number, genomic positions, insulation and CTCF, etc, enrichments at TAD borders insulations (**Rebuttal Fig. 19**).

We find that common, CAST and S129 TAD borders range in the same percentages for 40, 60 or 80% of NPs (Common: 37-40%, CAST: 29-32% and S129: 28-31%; **Rebuttal Fig. 19a**). A similar number of allele-specific borders is detected across all erosion levels, with approximately 1300 being consistently common (**Rebuttal Fig. 19b**). We also explored the enrichments of CTCF, Rad21, ATAC peaks, and housekeeping genes in the different sets of TAD borders, and found that the enrichments observed in the full (100%-NP) dataset are fully conserved in the eroded datasets, except for ATAC peak enrichment which is lost at 40% erosion.

Rebuttal figure 19. (a) Number of TAD borders in downsampled GAM datasets. (b) Upset plots show the number of common and unique TADs in each haplotype in GAM datasets randomly selecting only 40, 60 or 80% of the full collection of NPs (left to right). (c) Enrichment for CTCF, Rad21, ATAC-seq and housekeeping borders is maintained in the eroded datasets, except for ATAC-seq peak enrichment in 40% eroded datasets. The normalized insulation score in TADs categorized according to (b).

Second, we considered the suggestion of the reviewer and downsampled only the unphased GAM dataset by taking 50% of NPs, followed by recomputing unphased matrices and TAD borders. Comparison between the TAD borders obtained from the 50%-eroded unphased GAM dataset with the total ('uneroded') S129-specific and CAST-specific borders shows that a similar

number of TAD borders is detected in the halved GAM dataset, and that many borders are found in both alleles and the downsampled unphased dataset (**Rebuttal Fig. 20**). The border enrichments are also generally maintained.

Rebuttal Figure 20. Upset plots show the number of common and unique TADs in nonphased and each haplotype after downsampling nonphased GAM datasets by removing 50% of GAM samples (NPs) (a) and feature enrichment in these subgroups (b). Same intersection and enrichment analysis in the full GAM dataset (c; also revised EV Fig. 2e).

3c. Alternatively, considering the decoupling between allele-specific TADs and CTCF binding (lines 603-607), could these weaker borders be associated with ASE promoters (e.g. Zhang et al, Nature Genetics 2023)?

To assess whether the allele-specific weaker borders could be associated with ASE promoters, we counted the number of ASE gene promoters and also, biallelic gene promoters, common CTCF and allele-specific CTCF peaks in borders (**Rebuttal Fig. 21**). Even though there are relatively few ASE genes and allele-specific CTCF peaks in the genome, they are equally enriched in the allele-specific borders, as the common CTCF peaks and biallelic genes. These results show that the allele-specific borders are not more associated with ASE gene promoters or allele-specific CTCF peaks than common promoters and common CTCF peaks. Whereas it is possible that some allele-specific TAD borders may be regulated by allele-specific features, meaningful genome-wide trends are not identified.

Rebuttal Figure 21. Number of CAST borders (a) and S129 borders (b) overlapping with common, CAST and S129 CTCF peaks, expressed genes, CAST genes or S129 genes. Asterisks indicate statistical significance of finding feature-positive borders using 10000 circular permutation tests (***) represents $p \leq 0.0001$, ** represents $p \leq 0.001$). The smaller graphs show the percentage of each feature overlapping with CAST or S129 borders. Statistical significance was calculated with 10,000 circular permutations..

Minor remarks:

4- The use of the female and male signs to distinguish the chromosomes of parental origin can be confusing, as the F123 cells themselves are of male origin. Although their use is not unique to this study, the use of paternal and maternal (origin) should be prioritized (see e.g. Rao et al 2014 for an example of more appropriate mixed use).

We have corrected all figures to represent on CAST and S129 and removed the female and male signs.

5- Lines 83, 535 and 545: Tan et al, Science 2018 should be added.

We have added the reference.

6- Lines 240-243, 329-331 and 569-571: I'm not sure how ASE genes and monoallelic genes are different. This should be explained.

Both mono-allelic and ASE genes are expressed genes ($TPM \geq 1$). Monoallelic genes are only expressed from one allele, and are represented in the sequencing data with SNP-containing reads from only one allele (i.e. there are zero SNP-containing reads from the other allele). ASE genes refer to all genes, including monoallelic, which are transcribed with an imbalance between both alleles (i.e. we find $\log_2FC > 1$ or < -1 , $TPM \geq 1$, and $p\text{-value} \leq 0.05$).

We have revised the manuscript text as follows: ASE genes are all genes exhibiting allelic expression imbalance, while monoallelic genes are a subgroup of ASE genes which contain only reads from one allele.

6- Line 288: I can't find the origin of the ATAC-seq and ChIP-seq data. From the material and methods section, I gather that these were taken from previous publications? If so, these should be mentioned.

We had indicated the origin of the datasets by citing the references when the datasets were first mentioned in the main text (line 128-134, in revised manuscript). The origin and GEO of the datasets was also listed in EV Dataset 1. The ATAC-seq was publicly available from F123 mESCs (Juric et al., 2019). Some ChIP-seq datasets were also publicly available from F123 mESCs: for CTCF, cohesin (RAD21), H3K4me3 and H3K27ac (Huang et al., 2021). We produced H3K27me3, Pol2-S5p and Pol2-S7p for F123 mESCs, for the first time to our knowledge, and shared the datasets in GEO, as indicated in the manuscript.

7- Line 350: the nearly complete absence of co-occurrence for the S129 alleles is surprising. The authors should reflect on this, either here or in the discussion. Do the authors think that this is linked to the different contribution of allelic features between the paternal (CAST, stronger) and maternal (S129, weaker) genomes? Or do they have other explanations?

We agree that the absence of co-occurrence is likely linked with less abundant S129 features, but we cannot exclude that specific biological mechanisms contribute to this effect. Previous studies of allele-specific imbalance in gene expression and/or chromatin regulation have found a tendency for the paternal allele expression to be more abundant in different tissues (liver, brain, lung and kidney) or cells, including in F123-derived fibroblasts (Crowley et al, 2015; Savol et al, 2017); a new statement to this effect has been added in the “CAST-specific ASE genes and chromatin features co-occur in the linear genome” section. This makes us believe that the differences between CAST and S129 are likely due to parental effects, possibly with the different kinetics of DNA methylation and deposition of Polycomb marks during early development. Throughout the manuscript, we considered possible technical effects to the best of our knowledge, and were always reassured that the tendencies observed were not technical and are likely biological. For example, in manuscript Fig. 2g, we see that several features are more abundant in the paternal genome (sometimes 2-3x as abundant), but others such as CTCF (produced in the same laboratory as for example the H3K4me3) have similar amounts in CAST and S129 peaks (2540 and 2258).

8- Line 356: considering the different types of analyses in Fig. 2 and 3, and the non-allele-specific nature of the H3K27me3/Polycomb data, the quantitative contribution of different features appears difficult to assess. Unless the authors better integrate the two types of analyses, should this statement be rephrased?

In the revised manuscript, we have added the allele-specific classification of Polycomb peaks (in Fig. 2g; Fig. 2h; see also **Rebuttal Fig. 13**). We have also made more clear that ASE imbalance is associated both with Polycomb presence at the ASE gene promoters and is functionally linked with Polycomb repression, but that the Polycomb repression mechanisms may not act directly on the promoters of the least expressed allele, and likely have more complex mechanisms, including through allele specific effects on 3D chromatin structure. We revised the

text to more clearly report an association with Polycomb occupancy and repression mechanisms.

9- Line 485: Barshad et al, Nature Genetics 2023 should be added.

We have added the reference.

10a. Figures 1g and S2e: I'm not sure how the numbers in the Upset plots in the two figures relate to each other. Whereas total CAST borders are the same (2176) and 129 borders nearly (2158 vs 2154),

We are grateful for the reviewer's comment highlighting the slight difference in the numbers of allele-specific TAD borders between manuscript Fig. 1g and Fig. S2e. This effect comes from the order of TAD border intersection (Unphased - S129 - CAST or S129 - CAST - Unphased). This was an oversight in the first manuscript submission and we have now updated Fig. S2e (EV Fig. 2e) using the same order as in Figure 1g, for consistency (**Rebuttal Fig. 22**).

Rebuttal Figure 22. Revised plots in EV Fig. 2e after reordering the sets in the three-way comparison.

10b. I do not manage to see how the 3 combinations in Fig. 1g can be composed from the larger number of categories in Fig. S2e. Can the authors check?

The reason this effect happens is because of the increased overlapping of borders in the comparisons of a larger number of sets, as shown schematically in **Rebuttal Fig. 23**.

**Rebuttal Figure 23.** (a) In a two-way comparison, borders A and B are found distinct. (b) In a three-way comparison, borders A and B are found overlapping due to transitivity, because A/C are overlapping and B/C are overlapping.

*11 - Figure 3d: although I realize the values on the y-axis represent a normalized value, it appears counter-intuitive to see values in the range of 100k peaks / TAD. Should this be normalized to a more realistic value?*

We thank the reviewer for noticing this lapse. We accidentally mislabelled the graph 0 to 400K (=1e5), and it should have been 0 to 4 multiplied by 1e5 instead, and we have now corrected the y-axis.

*12 - Figure S3a: the values on the x-axis are unreadable. From what I can distinguish, they are not identical, making the information essential for the interpretation.*

We are grateful for the comment and have now replaced the items in this panel by a violin plot representation (**Rebuttal Fig. 24).**

**Rebuttal Figure 24.** Revised plots in EV Fig. 3b using violin representations. In particular, note the difference in y-axis scales.

*13- Figure S5d: I'm surprised to see many TADs of moderate size (<200 kb) that contain large numbers of active genes (up to 100) in these panels. What gene list was used to obtain these numbers? Is this the non-redundant gene list that is mentioned on page 27? If yes, can the authors confirm these numbers? If not, why did the authors not use the same list?*

We apologize for unclear scaling of the x axes in manuscript EV Fig. 5d. We have relabelled them to make the scaling more clear. We have now removed 1e6 at the right end of the x-axis and added (Mb) to the x axis title, ie “TAD length (Mb)”

14- Figures S6c and S7a are identical; the second occurrence can be removed. Am I correct that the CAST and S129 contact maps (but not the z-score map) are inverted? Adding a gradient scale to the panel will help the interpretation as well.

We are grateful to the reviewer for pointing out that these two panels were too visually similar to follow the concept. In fact, EV Fig. 6c (**Rebuttal Fig. 25a**) describes our procedure to extract ‘**differential contacts**’ between both haplotypes, while EV Fig. 7a (**Rebuttal Fig. 25b**) explains how we define ‘**strong contacts**, regardless of the other haplotype’. We have revised both panels to make the two types of analysis visually more distinct. We are also grateful for the reviewer noticing that the color indicators for S129 and CAST haplotypes were mismatched between the maps and the labels. Following the advice, we corrected the haplotype mismatch and have also added gradient scales.

a EV Figure 6c

b EV Figure 7a

Rebuttal Figure 25. Revised illustrations to introduce the concept of significant contacts (EV Figure 6c) and strong contacts (EV Fig. 7a).

The reasons we use two definitions are the following. **Differential contacts** were chosen to investigate the strongest structural differences between alleles and understand whether these differences relate to changes in specific chromatin features (e.g. expression, ATAC or histone marks). One such example is the Hist1 locus, where we saw that the topology in the S129 allele is strikingly different to the CAST allele and that there are more genes expressed in the CAST

allele than in the S129 allele. The analysis of ***strong contacts*** detected in a given allele are agnostic to whether they are equal, stronger or weaker in the other allele. We chose 'strong contacts' to investigate enhancer-promoter contacts and CTCF loops because these events are interesting irrespectively of whether they occur in only one allele or both. This is so, as the literature already states that E-P contacts can occur before or during gene activation. We were therefore curious to see whether ASE genes established contacts with putative regulatory regions when more active, more repressed, or in both states. Hence, we collected all strong E-P contacts, irrespectively of whether they were differential, and asked how they related to ASE imbalance. CTCF loops can also happen in one allele regardless of the other allele, and we were curious to see if the allele-specific CTCF loops could be driven by allele-specific CTCF and cohesin.

15th Jan 2025

Manuscript Number: MSB-2024-12394R

Title: Extensive folding variability between homologous chromosomes in mammalian cells

Dear Prof Pombo,

Thank you for the submission of your revised manuscript to Molecular Systems Biology. We have now received the enclosed reports from the referees that were asked to re-assess it. As you will see the reviewers are now globally supportive and I am pleased to inform you that we will be able to accept your manuscript pending the following final amendments and appropriate response to reviewers:

1) Please format the Data availability section according to the example below:

"The datasets and computer code produced in this study are available in the following databases:

- Chip-Seq data: Gene Expression Omnibus GSE46748 (<https://www.ncbi.nlm.nih.gov/geo/query/acc.cgi?acc=GSE46748>)

- Modeling computer scripts: GitHub (<https://github.com/SysBioChalmers/GECKO/releases/tag/v1.0>)

- [data type]: [full name of the resource] [accession number/identifier] ([doi or URL or identifiers.org/DATABASE:ACCESSION])"

2) Please ensure that all sequencing and proteomics data are now publicly released (and remove reviewer tokens from the Data Availability statement).

3) Please add the following statement to the Disclosure and competing interests statement: "Ana Pombo is a member of the Advisory Editorial Board of Molecular Systems Biology. This has no bearing on the editorial consideration of this article for publication." We updated our journal's competing interests policy in January 2022 and request authors to consider both actual and perceived competing interests. Please review the policy <https://www.embopress.org/competing-interests> and update your competing interests if necessary.

4) Our journal encourages inclusion of *data citations in the reference list* to directly cite datasets that were re-used and obtained from public databases. Data citations in the article text are distinct from normal bibliographical citations and should directly link to the database records from which the data can be accessed. In the main text, data citations are formatted as follows: "Data ref: Smith et al, 2001" or "Data ref: NCBI Sequence Read Archive PRJNA342805, 2017". In the Reference list, data citations must be labeled with "[DATASET]". A data reference must provide the database name, accession number/identifiers and a resolvable link to the landing page from which the data can be accessed at the end of the reference. Further instructions are available at .

5) In the Methods, please take care of the following:

- The Material and Methods section should be renamed to "Methods".

- Cell lines: Please be sure to include a sentence in the Methods as to whether or not the cell lines were recently authenticated and tested for mycoplasma contamination. Please also be sure to update the Author Checklist with this information and where it can be found in the manuscript.

6) All Materials and Methods need to be described in the main text using our 'Structured Methods' format. According to this format, the Methods section includes a Reagents and Tools Table (listing key reagents, experimental models, software and relevant equipment and including their sources and relevant identifiers) followed by a Methods and Protocols section describing the methods, ideally using a step-by-step protocol format. The aim is to facilitate adoption of the methodologies across labs. Please download and fill our Reagents and Tools Table template (.docx), which you can find in our author guidelines:

<https://www.embopress.org/doi/10.15252/msb.20178071>. "

7) Please place individual sections of the manuscript in the following order: Title page - Abstract & Keywords - Introduction - Results - Discussion - Methods - Data Availability - Acknowledgements - Disclosure and Competing Interests Statement - References - Figure Legends - Expanded View Figure Legends.

8) For the figures and figure legends, please take care of the following:

- Please make sure to update the callouts of all figures in the main manuscript text so that they appear sequentially.

- The legends and callouts for the 8 EV figures should be renamed to Figure EVx.

- Please note that the exact p values are not provided in the legends of figures 1H, 3D, E

- Please indicate the statistical test used for data analysis in the legends of figures 2B, 4E

- Please note that in figures 3D, E there is a mismatch between the annotated p values in the figure legend and the annotated p values in the figure file that should be corrected.

- Please note that the box plots need to be defined in terms of minima, maxima, centre, bounds of box and whiskers, and percentile in the legends of figures 3D, E; 4F, 5C.

- Please note that information related to n is missing in the legends of figures 2B, 3E, 4F, 5C.

9) Tables: Please rename Tables EV1-12 to Dataset EV1-12. Each dataset will need its legend removed from the manuscript and added to the corresponding file in a separate tab. Please update their callouts in main manuscript text. In addition, the "List of EV Datasets" should be removed from the manuscript file.

10) Funding: the funding information from the Comments box need to be included in More Funders list.

11) Synopsis:

- Synopsis image: Please provide the synopsis image as a high-resolution jpeg file (not a PDF) and ensure that the size is 550 pixels wide x (300-600) pixels high.

12) As part of the EMBO Publications transparent editorial process initiative (see our policy here:

https://www.embopress.org/transparent-process#Review_Process), Molecular Systems Biology will publish online a Peer Review File (PRF) to accompany accepted manuscripts. This file will be published in conjunction with your paper and will include the anonymous referee reports, your point-by-point response and all pertinent correspondence relating to the manuscript. Let us know whether you agree with the publication of the PRF and as here, if you want to remove or not any figures from it prior to publication. Please note that the Authors checklist will be published at the end of the PRF.

13) Please provide a point-by-point letter INCLUDING my comments as well as the reviewer's reports and your detailed responses (as Word file).

I look forward to reading a new revised version of your manuscript as soon as possible.

Yours sincerely,

Poonam Bheda, PhD
Scientific Editor
Molecular Systems Biology

Please click on the link below to submit the revision online

Reviewer #1:

The authors did a very thorough job and left none of my comments unaddressed. As a result, I would recommend that the manuscript is published in its current form, and I would like to see Rebuttal Figures 4 and 7 also included in the final published figures as I think they make the data presentation more complete.

Reviewer #2:

I appreciate the many modifications and additions that were made by the authors. Although the manuscript remains dense, this has considerably improved the analyses and their ease of interpretation. Except for the minor comment below, I consider the manuscript ready for publication in Molecular and Systems Biology.

Minor comment:

The addition of the paragraph about informative contact entries provides useful information how GAM and Hi-C compare for allele-specific analysis. From what I gather though, the comparison does not take into account the large difference in SNP-density between the mouse F123 and the human G12878 cells. The authors should at minimum mention this difference, and ideally should estimate/analyze how the two datasets compare when a similar SNP density was present (for instance by computationally masking excess SNPs in the GAM data set).-

- Please format the Data availability section according to the example below:

"The datasets and computer code produced in this study are available in the following databases:

- Chip-Seq data: Gene Expression Omnibus GSE46748
(<https://www.ncbi.nlm.nih.gov/geo/query/acc.cgi?acc=GSE46748>)

We modified the Data availability section accordingly. Please see lines 1140-1142.

- Please ensure that all sequencing and proteomics data are now publicly released (and remove reviewer tokens from the Data Availability statement).

All sequencing and proteomics data are now publicly available. The reviewer tokens were removed from the Data Availability section.

- Please add the following statement to the Disclosure and competing interests statement: "Ana Pombo is a member of the Advisory Editorial Board of Molecular Systems Biology. This has no bearing on the editorial consideration of this article for publication." We updated our journal's competing interests policy in January 2022 and request authors to consider both actual and perceived competing interests.

We added the statement to the 'Disclosure and competing interests' section. Please see the lines 1189-1191.

- Our journal encourages inclusion of *data citations in the reference list* to directly cite datasets that were re-used and obtained from public databases. Data citations in the article text are distinct from normal bibliographical citations and should directly link to the database records from which the data can be accessed. In the main text, data citations are formatted as follows: "Data ref: Smith et al, 2001" or "Data ref: NCBI Sequence Read Archive PRJNA342805, 2017". In the Reference list, data citations must be labeled with "[DATASET]". A data reference must provide the database name, accession number/identifiers and a resolvable link to the landing page from which the data can be accessed at the end of the reference. Further instructions are available at <https://www.embopress.org/page/journal/17574684/authorguide#referencesformat>.

We have added the requested formats.

- In the Methods, please take care of the following:
 - - The Material and Methods section should be renamed to "Methods".

The Materials and Methods section was renamed into Methods

- - Cell lines: Please be sure to include a sentence in the Methods as to whether or not the cell lines were recently authenticated and tested for mycoplasma contamination. Please also be sure to update the Author Checklist with this information and where it can be found in the manuscript.

We added statement "F123 mESC batches all tested negative for Mycoplasma infection, performed according to the manufacturer's instructions (AppliChem, Cat#A3744,0020)." Please see the lines 701-702.

- All Materials and Methods need to be described in the main text using our 'Structured Methods' format. According to this format, the Methods section includes a Reagents and Tools Table (listing key reagents, experimental models, software and relevant equipment and including their sources and relevant identifiers) followed by a Methods and Protocols section describing the methods, ideally using a step-by-step protocol format. The aim is to facilitate adoption of the methodologies across labs.

We prepared Reagents and Tools Table, following the example and provided it as a new file Irastorza-Azcarate_Reagents_Tools_Table.docx

- Please place individual sections of the manuscript in the following order: **Title page - Abstract & Keywords - Introduction - Results - Discussion - Methods - Data Availability - Acknowledgements - Disclosure and Competing Interests Statement - References - Figure Legends - Expanded View Figure Legends.**

We confirm the correct order of the individual sections of the manuscript

- For the figures and figure legends, please take care of the following:
 - - Please make sure to update the callouts of all figures in the main manuscript text so that they appear sequentially.
 - - The legends and callouts for the 8 EV figures should be renamed to Figure EVx.
 - - Please note that the exact p values are not provided in the legends of figures 1H, 3D, E
 - - Please indicate the statistical test used for data analysis in the legends of figures 2B, 4E
 - - Please note that in figures 3D, E there is a mismatch between the annotated p values in the figure legend and the annotated p values in the figure file that should be corrected.
 - - Please note that the box plots need to be defined in terms of minima, maxima, centre, bounds of box and whiskers, and percentile in the legends of figures 3D, E; 4F, 5C.
 - - Please note that information related to n is missing in the legends of figures 2B, 3E, 4F, 5C.

All figures and figure legends were modified/reviewed accordingly. In the case of 4E, it is not a statistical test, we simply report the Gene Ontology for the top 5% genes, as an illustration.

- Tables: Please rename Tables EV1-12 to Dataset EV1-12. Each dataset will need its legend removed from the manuscript and added to the corresponding file in a separate tab. Please update their callouts in main manuscript text. In addition, the "List of EV Datasets" should be removed from the manuscript file.

We renamed all EV Datasets accordingly, both the actual files and the manuscript references. The list of EV datasets was removed from the manuscript.

- Funding: the funding information from the Comments box need to be included in More Funders list.

We have entered all funders possible, and left some remainder funders in the Comments. The reason being that some funder names are not accepted by the portal – all left in comments were tested.

- Synopsis:
 - - Synopsis image: Please provide the synopsis image as a high-resolution jpeg file (not a PDF) and ensure that the size is 550 pixels wide x (300-600) pixels high.

We now include a copy of the synopsis image named Irastorza-Azcarate_Synopsis_image_550pxi.jpg which is rescaled according to the given size constraints.

We confirm that the current versions of the synopsis image and text reflect the latest version of the manuscript.

- As part of the EMBO Publications transparent editorial process initiative (see our policy here: https://www.embopress.org/transparent-process#Review_Process), Molecular Systems Biology will publish online a Peer Review File (PRF) to accompany accepted manuscripts. This file will be published in conjunction with your paper and will include the anonymous referee reports, your point-by-point response and all pertinent correspondence relating to the manuscript. Let us know whether you agree with the publication of the PRF and as here, if you want to remove or not any figures from it prior to publication. Please note that the Authors checklist will be published at the end of the PRF.

Yes, we agree with online publication of the PRF file together with the manuscript. All results can be included.

- Please provide a point-by-point letter INCLUDING my comments as well as the reviewer's reports and your detailed responses (as Word file).

Additional author changes:

- We have removed the reference Shevchenko et al. 2006, as it was not cited in the current or previous versions of the manuscript

- We have added Rao et al 2014 (and details of dataset utilized in the manuscript) to Dataset EV1, as the latest revised version of the paper included our reanalysis of the published data in Fig EV1f.
- We have removed the Kubo HiC dataset, which was not used in the manuscript. As previously stated in the legend of Fig. EV1e, the 26% proportion of ligation events was directly obtained from Giorgetti et al. 2016, and not recalculated.
- We also remove reference Gusman et al. 2014, which was not applicable to the sentence it was mentioned (line 168)

Reviewer #1:

The authors did a very thorough job and left none of my comments unaddressed. As a result, I would recommend that the manuscript is published in its current form, and I would like to see Rebuttal Figures 4 and 7 also included in the final published figures as I think they make the data presentation more complete.

Answer:

The revised version of the manuscript has already included the content of Rebuttal Fig. 7 as Fig. EV7c and EV7d. We would prefer to keep Rebuttal Figure 4 in the Peer Review file only, as it does not fit well with the storytelling of the manuscript, and it will be made available in the Peer Review File, which makes its content open to the community.

Reviewer #2:

I appreciate the many modifications and additions that were made by the authors. Although the manuscript remains dense, this has considerably improved the analyses and their ease of interpretation. Except for the minor comment below, I consider the manuscript ready for publication in Molecular and Systems Biology.

Minor comment:

The addition of the paragraph about informative contact entries provides useful information how GAM and Hi-C compare for allele-specific analysis. From what I gather though, the comparison does not take into account the large difference in SNP-density between the mouse F123 and the human G12878 cells. The authors should at minimum mention this difference, and ideally should estimate/analyze how the two datasets compare when a similar SNP density was present (for instance by computationally masking excess SNPs in the GAM data set).

Answer:

We thank the reviewer for this suggestion. We modified the sentence lines 169-173 as follows: "Comparison of informative contact entries in phased GAM and phased Hi-C matrices from **human** GM12878 B-lymphoblastoid cells **of lower SNP-density**

(data from Rao *et al*, 2014) shows the detection of 79-93% of all possible intra-chromosomal contacts in GAM data, at 50 kb for all genomic distances, compared with only 20-51% in Hi-C data at the same resolution".

10th Apr 2025

Manuscript number: MSB-2024-12394RR

Title: Extensive folding variability between homologous chromosomes in mammalian cells

Dear Prof Pombo,

Congratulations on an excellent manuscript, I am pleased to inform you that your manuscript has been accepted for publication in Molecular Systems Biology. Thank you for your comprehensive response to referee concerns. It has been a pleasure to work with you to get this to the acceptance stage.

Yours sincerely,

Poonam Bheda, PhD
Scientific Editor
Molecular Systems Biology
